# Near-Optimal Reinforcement Learning for Linear Distributionally Robust Markov Decision Processes

## Abstract

We study off-dynamics reinforcement learning (RL), where the policy training and deployment environments are different. To deal with this environmental perturbation, we focus on learning policies robust to uncertainties in transition dynamics under the framework of distributionally robust Markov decision processes (DR-MDPs), where the nominal and perturbed dynamics are linear Markov Decision Processes. We propose a novel algorithm We-DRIVE-U that enjoys an average sub-optimality $\widetilde{\mathcal{O}}\big(dH \cdot \min\{1/\rho, H\}/\sqrt{K}\big)$, where $K$ is the number of episodes, $H$ is the horizon length, $d$ is the feature dimension and $\rho$ is the uncertainty level. This result improves the state-of-the-art by $\mathcal{O}(dH/\min\{1/\rho, H\})$. We also construct a novel hard instance and derive the first information-theoretic lower bound in this setting. In stark contrast with standard linear MDPs, our lower bound depends on the uncertainty level $\rho$, revealing the unique feature of DRMDPs. Our algorithm also enjoys a 'rare-switching' design, and thus only requires $\mathcal{O}(dH \log(1 + H^2 K))$ policy switches and $\mathcal{O}(d^2 H \log(1 + H^2 K))$ calls for oracle to solve dual optimization problems, which significantly improves the computational efficiency of existing algorithms, whose policy switch and oracle complexities are both $\mathcal{O}(K)$.

## 1 Introduction

In dynamic decision-making and reinforcement learning (RL), Markov decision processes (MDPs) offer a well-established framework for understanding complex systems and guiding agent behavior [37]. However, MDPs encounter significant challenges in practical applications due to incomplete knowledge of model parameters, especially transition probabilities. This sim-to-real gap, representing the difference between training and testing environments, can lead to failures in fields like infectious disease control and robotics [8, 60, 21, 22, 33]. To address these challenges, off-dynamics RL provides a framework where policies are trained on a source domain and deployed to a distinct target domain, promoting robust performance across varying environments [7, 17, 42]. Within this framework, distributionally robust Markov decision processes (DRMDPs) have emerged as a promising way to model transition uncertainty. DRMDPs focus on learning robust policies that perform well under worst-case scenarios [29, 16]. Prior works [56, 52, 32, 35, 53, 34] have proposed algorithms mainly for tabular DRMDPs with finite number of states and actions, which are infeasible when facing large state and action spaces.

In environments characterized by large state and action spaces, function approximation techniques are crucial to overcome the computational burden posed by high dimensionality. Linear function approximation methods, based on relatively simple function classes, have shown significant theoretical and practical successes in standard MDP environments [19, 11, 10, 50, 12]. However, their application in DRMDPs introduces additional complexities. These complexities arise from the nonlinearity caused by the dual formulation in the worst-case analysis, even when the transition dynamics in the

source domain are modeled as linear. Recently, [23] provided the first theoretical results in the *online* setting of $d$-rectangular linear DRMDPs, a specific type of DRMDPs where the nominal model is a linear MDP [19] and the uncertainty set is defined based on the linear structure of the nominal transition kernel. Apart from this, online DRMDP with linear function approximation is largely underexplored and it is not clear how far existing algorithms are from optimal. Consequently, two natural questions arise:

*Can we improve the current results for online DRMDPs with linear function approximation?*
*What is the fundamental limit in this setting?*

In this paper, we provide an affirmative answer to the first question and answer the second question by providing an information theoretic lower bound for $d$-rectangular linear DRMDPs. In particular, motivated by the adoption of variance-weighted ridge regression to achieve nearly optimal result in standard linear MDPs [62, 61, 58, 20, 59, 10, 13], we propose a variance-aware distributionally robust algorithm to solve the off-dynamics RL problem. Due to the nonlinearity caused by the dual optimization of DRMDPs, the adoption of variance information in linear DRMDPs is highly nontrivial. Existing algorithms that incorporate variance information in learning linear DRMDPs requires coverage assumptions on the offline dataset [24, 41], which is infeasible in the online setting where the algorithm needs to interact with the environment to collect data. To be specific, for online DRMDPs, the adoption of variance-weighted ridge regression causes the following **unique challenges** for both algorithm design and theoretical analysis:

- *(Fundamental non-linearity induced by the uncertainty)* The consideration of uncertainty set renders that the (robust) Bellman equations are not linear with respect to the nominal kernel, a key feature in standard MDP. A direct consequence is that though the Q-function remains a linear representation, its parameters must be estimated element-wisely from $d$ (i.e., parameter dimension) variance-weighted ridge regressions, instead of one in standard linear MDP. This poses challenges for algorithm design, as it requires properly incorporating variance information into the estimation process and quantitatively controlling the estimation uncertainty.
- *(Precise control on variance estimation.)* Existing theoretical analyses of online linear MDPs rely heavily on the *Elliptical Potential Lemma*, showing that the estimation error shrinks rapidly enough to guarantee the near-optimality of the learned policy within a small number of rounds. However, this lemma is not applicable in our setting due to the element-wise parameter estimation procedure mentioned above. Instead, we adopt a large-$k$ regime to control the estimation error, based upon the intuition that when the sample size is large, the variance estimation should be close to the true variance (see Lemma D.7). Finally, we leverage the 'Range Shrinkage' property (see Lemma H.10) for linear DRMDPs to bound the true variance, and thus obtain an improved bound.

Our work poses a distinct algorithm design and calls for different theoretical analysis techniques. Our **main contributions** are summarized as follows:

- We propose a novel algorithm, We-DRIVE-U, for $d$-rectangular linear DRMDPs with total-variation (TV) divergence uncertainty sets. We-DRIVE-U is designed based on the optimistic principle [18, 19, 10] to trade off the exploration and exploitation during interacting with the source environment to learn a robust policy. The key novelty of We-DRIVE-U lies in incorporating the variance information into the policy learning, by a carefully designed optimistic estimator of the variance of the optimal robust value function.
- We prove that We-DRIVE-U achieves an average suboptimality of $\widetilde{\mathcal{O}}(dH \cdot \min\{1/\rho, H\}/\sqrt{K})$ when the number of episode $K$ is large, which improves the state-of-the-art result [23] by $\widetilde{\mathcal{O}}(dH/\min\{1/\rho, H\})$, We highlight that the average suboptimality of We-DRIVE-U demonstrates the 'Range Shrinkage' property (refer to Lemma H.10) through the term $\min\{1/\rho, H\}$. We further established an information-theoretic lower bound $\Omega(dH^{1/2} \cdot \min\{1/\rho, H\}/\sqrt{K})$, which shows that We-DRIVE-U is *near-optimal* up to $\mathcal{O}(\sqrt{H})$ for any uncertainty level $\rho \in (0, 1]$.
- We-DRIVE-U is favorable in applications where policy switching is risky or costly, since We-DRIVE-U achieves $\mathcal{O}(dH \log(1 + H^2 K))$ global policy switch (refer to Definition 4.5). Moreover, we note that calls for oracle to solve dual optimizations (3.3) are one of the main sources of computation complexity in DRMDP with linear function approximation. Thanks to the specifically designed 'rare-switching' regime, We-DRIVE-U achieves $\mathcal{O}(d^2 H \log(1 + H^2 K))$ oracle complexity (refer to Definition 4.6). Both results improve exiting online DRMDP algorithms by a factor of $K$. Thus, We-DRIVE-U enjoys low switching cost and low computation cost.

**Notations.** For any positive integer $H \in \mathbb{Z}_+$, we denote $[H] = \{1, 2, \cdots, H\}$. For any set $\mathcal{S}$, define $\Delta(\mathcal{S})$ as the set of probability distributions over $\mathcal{S}$. For any function $V : \mathcal{S} \to \mathbb{R}$, define $[\mathbb{P}_h V](s, a) = \mathbb{E}_{s' \sim P_h(\cdot|s, a)}[V(s')]$, and $[V(s)]_\alpha = \min\{V(s), \alpha\}$, where $\alpha > 0$ is a constant. For a vector $\boldsymbol{x}$, define $x_j$ as its $j$-th entry. Moreover, denote $[x_i]_{i \in [d]}$ as a vector with the $i$-th entry being $x_i$. For a matrix $A$, denote $\lambda_i(A)$ as the $i$-th eigenvalue of $A$. For two matrices $A$ and $B$, denote $A \preceq B$ as the fact that $B - A$ is a positive semi-definite matrix. For any $P, Q \in \Delta(\mathcal{S})$, the total variation divergence of $P$ and $Q$ is defined as $D(P||Q) = 1/2 \int_{\mathcal{S}} |P(s) - Q(s)| ds$.

## 2 Preliminary

In this section, we introduce the mathematical framework of our setting. We use a tuple $\mathrm{DRMDP}(\mathcal{S}, \mathcal{A}, H, \mathcal{U}^\rho(P^0), r)$ to denote a finite horizon DRMDP, where $\mathcal{S}$ and $\mathcal{A}$ are the state and action spaces, $H \in \mathbb{Z}_+$ is the horizon length, $P^0 = \{P_h^0\}_{h=1}^H$ is the nominal transition kernel, $\mathcal{U}^\rho(P^0) = \bigotimes_{h \in [H]} \mathcal{U}_h^\rho(P_h^0)$ denotes an uncertainty set centered around the nominal transition kernel with an uncertainty level $\rho \geq 0$, $r = \{r_h\}_{h=1}^H$ is the reward function. A policy $\pi = \{\pi_h\}_{h=1}^H$ is a sequence of decision rules. For a policy $\pi$, we define the robust value function and Q-function for any $(h, s, a) \in [H] \times \mathcal{S} \times \mathcal{A}$ as

$$V_h^{\pi, \rho}(s) = \inf_{P \in \mathcal{U}^\rho(P^0)} \mathbb{E}^P \left[ \sum_{t=h}^H r_t(s_t, a_t) \Big| s_h = s, \pi \right],$$

$$Q_h^{\pi, \rho}(s, a) = \inf_{P \in \mathcal{U}^\rho(P^0)} \mathbb{E}^P \left[ \sum_{t=h}^H r_t(s_t, a_t) \Big| s_h = s, a_h = a, \pi \right].$$

Moreover, we define the optimal robust value function and optimal robust state-action value function: for any $(h, s, a) \in [H] \times \mathcal{S} \times \mathcal{A}$, $V_h^{\star, \rho}(s) = \sup_{\pi \in \Pi} V_h^{\pi, \rho}(s)$, $Q_h^{\star, \rho}(s, a) = \sup_{\pi \in \Pi} Q_h^{\pi, \rho}(s, a)$, where $\Pi$ is the set of all policies. Correspondingly, the optimal robust policy is the policy that achieves the optimal robust value function $\pi^\star = \mathrm{argsup}_{\pi \in \Pi} V_h^{\pi, \rho}(s)$.

In this paper, we focus on the $d$-rectangular linear DRMDP [26, 4, 23, 24], where the nominal environment is a linear MDP [19] with a simplex state space, defined as follows.

**Assumption 2.1.** Given a known feature mapping $\boldsymbol{\phi} : \mathcal{S} \times \mathcal{A} \to \mathbb{R}^d$ satisfying $\sum_{i=1}^d \phi_i(s, a) = 1$, $\phi_i(s, a) \geq 0$, for any $(i, s, a) \in [d] \times \mathcal{S} \times \mathcal{A}$, we assume the reward functions $\{r_h\}_{h=1}^H$ and nominal transition kernels $\{P_h^0\}_{h=1}^H$ are linearly parameterized. Specifically, for any $(h, s, a) \in [H] \times \mathcal{S} \times \mathcal{A}$, $r_h(s, a) = \langle \boldsymbol{\phi}(s, a), \boldsymbol{\theta}_h \rangle, P_h^0(\cdot|s, a) = \langle \boldsymbol{\phi}(s, a), \boldsymbol{\mu}_h^0(\cdot) \rangle$, where $\{\boldsymbol{\theta}_h\}_{h=1}^H$ are known vectors with bounded norm $\|\boldsymbol{\theta}_h\|_2 \leq \sqrt{d}$ and $\{\boldsymbol{\mu}_h\}_{h=1}^H$ are unknown probability measures over $\mathcal{S}$.

In $d$-rectangular linear DRMDPs, an uncertainty set of transition dynamics $\mathcal{U}_h^\rho(P_h^0)$ is defined based on the linear structure of $P_h^0$ satisfying Assumption 2.1. In particular, for any $(h, i) \in [H] \times [d]$, we first define the factor uncertainty set $\mathcal{U}_{h,i}^\rho(\mu_{h,i}^0) = \{\mu : \mu \in \Delta(\mathcal{S}), D(\mu||\mu_{h,i}^0) \leq \rho\}$, where $D(\cdot||\cdot)$ is a probability divergence which we choose as the total variation (TV) divergence in this paper. Then the uncertainty set of transitions for state $s$ and action $a$ is defined as $\mathcal{U}_h^\rho(s, a; \boldsymbol{\mu}_h^0) = \{\sum_{i=1}^d \phi_i(s, a) \mu_{h,i}(\cdot) : \mu_{h,i}(\cdot) \in \mathcal{U}_{h,i}^\rho(\mu_{h,i}^0), \forall i \in [d]\}$. We also denote $\mathcal{U}_h^\rho(P_h^0) = \bigotimes_{(s,a) \in \mathcal{S} \times \mathcal{A}} \mathcal{U}_h^\rho(s, a; \boldsymbol{\mu}_h^0)$ as the collection of uncertainty sets on the whole state and action spaces. Built on these definitions, [23] showed that the following robust Bellman equations hold for any policy $\pi$

$$Q_h^{\pi, \rho}(s, a) = r_h(s, a) + \inf_{P_h(\cdot|s, a) \in \mathcal{U}_h^\rho(s, a; \boldsymbol{\mu}_h^0)} [\mathbb{P}_h V_{h+1}^{\pi, \rho}](s, a), \tag{2.1a}$$

$$V_h^{\pi, \rho}(s) = \mathbb{E}_{a \sim \pi_h(\cdot|s)} \left[ Q_h^{\pi, \rho}(s, a) \right], \tag{2.1b}$$

Similarly, we have the robust Bellman optimality equations

$$Q_h^{\star, \rho}(s, a) = r_h(s, a) + \inf_{P_h(\cdot|s, a) \in \mathcal{U}_h^\rho(s, a; \boldsymbol{\mu}_h^0)} [\mathbb{P}_h V_{h+1}^{\star, \rho}](s, a), \tag{2.2a}$$

$$V_h^{\star, \rho}(s) = \max_{a \in \mathcal{A}} Q_h^\star(s, a). \tag{2.2b}$$

In the context of online DRMDPs, an agent actively interacts with the nominal environment within $K$ episodes to learn the optimal robust policy. Specifically, at the start of episode $k$, an agent chooses a policy $\pi^k$ based on the history information and receives the initial state $s_1^k$. Then the agent interacts with the nominal environment by executing $\pi^k$ until the end of episode $k$, and collects a new trajectory.

The goal of the agent is to minimize the average suboptimality[1] after $K$ episodes, which is defined as
$$\text{AveSubopt}(K) = 1/K \sum_{k=1}^{K} \left[ V_1^{\star,\rho}(s_1^k) - V_1^{\pi^k,\rho}(s_1^k) \right].$$

[25] recently show that sample efficient learning in online tabular DRMDPs is impossible in the presence of support shift, i.e., the nominal kernel and target kernel do not share the same support. Built on the hard instance constructed in their work, we carefully design feature mappings for the transition kernel to extend their lower bound to the following one for online linear DRMDPs.

**Proposition 2.2.** (Hardness result) There exists two $d$-rectangular linear DRMDPs $\{\mathcal{M}_\theta\}_{\theta \in \{0,1\}}$, such that $\inf_{\mathcal{ALG}} \sup_{\theta \in \{0,1\}} \mathbb{E}[\text{AveSubopt}^{\mathcal{M}_\theta, \mathcal{ALG}}(K)] \geq \Omega(\rho \cdot H)$, where $\text{AveSubopt}^{\mathcal{M}_\theta, \mathcal{ALG}}(K)$ is the average suboptimality of algorithm $\mathcal{ALG}$ in the $d$-rectangular linear DRMDP $\mathcal{M}_\theta$.

Note that the lower bound in Proposition 2.2 does not converge to zero as $K$ increases, which means that in general no algorithm can guarantee to learn the optimal robust policy approximately. To circumvent this problem, in the rest of paper we focus on a tractable subclass of $d$-rectangular linear DRMDP following [23, 25], which is formally defined in the following assumption.

**Assumption 2.3** (Fail-state). Assume there exists a 'fail state' $s_f$ in the $d$-rectangular linear DRMDP, such that for all $(h,a) \in [H] \times \mathcal{A}$, $r_h(s_f, a) = 0$, $\mathbb{P}_h^0(s_f | s_f, a) = 1$.

With Assumption 2.3, we follow the framework in [23], where we have the following results on robust value functions that are helpful in solving the optimization in (2.2).

**Proposition 2.4** (Remark 4.2 of [23]). Under Assumption 2.3, we have $Q_h^{\pi,\rho}(s_f, a) = V_h^{\pi,\rho}(s_f) = 0$, $\forall (\pi, h, a) \in \Pi \times [H] \times \mathcal{A}$. Moreover, for any function $V : \mathcal{S} \to [0, H]$ that satisfies $\min_{s \in \mathcal{S}} V(s) = V(s_f) = 0$, we have $\inf_{\mu \in \mathcal{U}^\rho(\mu^0)} \mathbb{E}_{s \sim \mu} V(s) = \max_{\alpha \in [0,H]} \{ \mathbb{E}_{s \sim \mu^0} [V(s)]_\alpha - \rho\alpha \}$.

# 3 Algorithm design

One prominent property of the $d$-rectangular DRMDP is that the robust Q-functions possess linear representations with respect to the feature mapping $\phi$. In particular, under Assumptions 2.1 and 2.3, [23] show that for any $(\pi, s, a, h) \in \Pi \times \mathcal{S} \times \mathcal{A} \times [H]$, the robust Q-function $Q_h^{\pi,\rho}(s,a)$ has a linear form as follows $Q_h^{\pi,\rho}(s,a) = \left( r_h(s,a) + \phi(s,a)^\top \boldsymbol{\nu}_h^{\pi,\rho} \right) \mathbb{1}\{s \neq s_f\}$, where $\boldsymbol{\nu}_h^{\pi,\rho} = \left( \nu_{h,1}^{\pi,\rho}, \ldots, \nu_{h,d}^{\pi,\rho} \right)^\top$, $\nu_{h,i}^{\pi,\rho} = \max_{\alpha \in [0,H]} \left\{ z_{h,i}^\pi(\alpha) - \rho\alpha \right\}$, $z_{h,i}^\pi(\alpha) = \mathbb{E}^{\mu_{h,i}^0} \left[ V_{h+1}^{\pi,\rho}(s') \right]_\alpha$ and $\alpha \in [0, H]$ is the dual variable derived from the dual formulation (see Proposition H.1 for more details). Moreover, the robust Bellman optimality equation (2.2) shows that the greedy policy with respect to the optimal robust Q-function is exactly the optimal robust policy $\pi^\star$. Therefore, the core idea behind the algorithm design is to estimate the optimal robust Q-function using linear function approximation, and then find $\pi^\star$ by the greedy policy derived from the estimated optimal robust Q-function. We present our algorithm in Algorithm 1.

## 3.1 Variance-weighted ridge regression for online DRMDPs

Algorithm 1 is a value-iteration based algorithm that iteratively estimates the robust Q-function through variance-weighted ridge regression. Different from the Q-function estimation for standard linear MDP, we element-wisely estimate the parameters of robust Q-functions. This is a distinct feature for linear DRMDPs. We next interpret the details of our algorithm design.

From Line 6 to 13 of Algorithm 1, we adopt the backward induction procedure to update the robust Q-function estimation. In particular, for any $(k,h) \in [K] \times [H]$, suppose we have an estimated robust value function $\hat{V}_{k,h+1}^\rho$. By the robust Bellman optimality equation (2.2) and Proposition 2.4, conducting one step backward induction on $\hat{V}_{k,h+1}^\rho$ leads to the following linear form [23]:

$$r_h(s,a) + \inf_{P_h \in \mathcal{U}_h^\rho(s,a;\boldsymbol{\mu}^0)} \mathbb{P}_h[\hat{V}_{k,h+1}](s,a) = \phi(s,a)^\top (\boldsymbol{\theta}_h + \boldsymbol{\nu}_h^{\rho,k}) \mathbb{1}\{s \neq s_f\}, \quad (3.1)$$

where $\nu_{h,i}^{\rho,k} := \max_{\alpha \in [0,H]} \{z_{h,i}^k(\alpha) - \rho\alpha\}$ and $z_{h,i}^k(\alpha) := \mathbb{E}^{\mu_{h,i}^0} [\hat{V}_{k,h+1}(s')]_\alpha$, for any $i \in [d]$. Note that under Assumption 2.1, for any $\alpha \in [0, H]$, $z_{h,i}^k(\alpha)$ is the $i$-th element of the parameter of the

---

[1]Our 'average sub-optimality' differs from standard 'regret' as it measures the gap to the optimal policy in the worst-case target environment, not the nominal one.

following linear formulation, $[\mathbb{P}_h^0[\hat{V}_{k,h+1}]_\alpha](s,a) = \langle \phi(s,a), \mathbf{z}_h^k(\alpha)\rangle$. Thus, we can estimate $\mathbf{z}_h^k(\alpha)$ from data to get estimations of $z_{h,i}^k(\alpha), \forall i \in [d]$. To this end, we introduce the variance-weighted ridge regression regime to estimate $\mathbf{z}_h^k(\alpha)$ as follows

$$\min_{\mathbf{z} \in \mathbb{R}^d} \sum_{\tau=1}^{k-1} \bar{\sigma}_{\tau,h}^{-2} \Big( \mathbf{z}^\top \phi(s_h^\tau, a_h^\tau) - \big[\hat{V}_{k,h+1}^\rho(s_{h+1}^\tau)\big]_\alpha \Big)^2 + \lambda \|\mathbf{z}\|_2^2,$$

which leads to the following closed-form estimation

$$\hat{\mathbf{z}}_h^k(\alpha) = \mathbf{\Sigma}_{k,h}^{-1} \sum_{\tau=1}^{k-1} \bar{\sigma}_{\tau,h}^{-2} \phi(s_h^\tau, a_h^\tau) \big[\hat{V}_{k,h+1}^\rho(s_{h+1}^\tau)\big]_\alpha, \tag{3.2}$$

where $\mathbf{\Sigma}_{k,h} = \lambda\mathbf{I} + \sum_{\tau=1}^{k-1} \bar{\sigma}_{\tau,h}^{-2} \phi(s_h^\tau, a_h^\tau) \phi(s_h^\tau, a_h^\tau)^\top$ and $\bar{\sigma}_{\tau,h}$ is a variance estimator that will be formally introduced in Section 3.2. We then approximate $\boldsymbol{\nu}_h^{\rho,k}$ by solving the optimization problem element-wisely

$$\hat{\nu}_{h,i}^{\rho,k} = \max_{\alpha \in [0,H]} \big\{ \hat{z}_{h,i}^k(\alpha) - \rho\alpha \big\}, \quad i \in [d]. \tag{3.3}$$

Then we can estimate the Q-function via (3.1). Due to the nature of online RL, the estimation might be highly uncertain due to the lack of exploration, and thus we incorporate a bonus term $\hat{\Gamma}_{k,h}(s,a) = \beta \sum_{i=1}^d \phi_i(s,a) \sqrt{\mathbf{1}_i^\top \mathbf{\Sigma}_{k,h}^{-1} \mathbf{1}_i}$ into the robust Q-function estimation, where $\beta = \widetilde{\mathcal{O}}(H\sqrt{d\lambda} + \sqrt{d})$. The final estimator is given on Line 8 of Algorithm 1. We will show in later analysis that the estimated Q-function is an optimistic estimator for the optimal robust Q-function.

Inspired by [10], we also establish estimators for the lower bound of robust Q-functions by the same backward induction procedure, which will be helpful in constructing the variance estimator $\bar{\sigma}_{\tau,h}$ as shown in the next section. In particular, given $\check{V}_{k,h}^\rho$, We obtain the variance weighted regression estimator $\check{\mathbf{z}}_h^k(\alpha) = \mathbf{\Sigma}_{k,h}^{-1} \sum_{\tau=1}^{k-1} \bar{\sigma}_{\tau,h}^{-2} \phi(s_h^\tau, a_h^\tau) \big[\check{V}_{k,h+1}^\rho(s_{h+1}^\tau)\big]_\alpha$. Then we get the estimation

$$\check{\nu}_{h,i}^{\rho,k} = \max_{\alpha \in [0,H]} \big\{ \check{z}_{h,i}^k(\alpha) - \rho\alpha \big\}, \quad i \in [d]. \tag{3.4}$$

By (3.1) and (3.4), we get another estimation of the robust Q-function, which we aim to show is a pessimistic estimation of the optimal robust Q-function. Similarly, to quantify the uncertainty caused by online exploration, we introduce a penalty term $\check{\Gamma}_{k,h}(s,a) = \bar{\beta} \sum_{i=1}^d \phi_i(s,a) \sqrt{\mathbf{1}_i^\top \mathbf{\Sigma}_{k,h}^{-1} \mathbf{1}_i}$, where $\bar{\beta} = \widetilde{\mathcal{O}}(H\sqrt{d\lambda} + \sqrt{d^3 H^3})$. The final pessimistic estimator $\check{Q}_{k,h}^\rho$ is shown on Line 9 of Algorithm 1. Though [24] also constructed pessimistic robust Q-function estimations for DRMDPs, our methods are very different due to 1) they do not update the estimation episodically, 2) their estimators are used to get the optimal robust policy estimation, while ours are used to construct the variance estimator, as shown in the next section.

## 3.2 Estimating the variance of the value function

In this section, we construct the variance weight $\bar{\sigma}_{\tau,h}$ used in (3.2) and aim to get an optimistic estimator for the variance of the optimal robust value function, $\mathbb{V}_h V_{h+1}^{*,\rho}$. Due to the distinct element-wise estimation procedure introduced in the previous section, the coarse variance estimation design in [10] for standard linear MDPs does not apply. Instead, we need to carefully design the variance estimators used in weighted ridge-regressions based upon the unique characteristics of linear DRMDPs.

We desire to design the variance estimator at episode $k$ to be a uniform variance upper bound for all subsequent episodes. To obtain the optimistic estimator for $\mathbb{V}_h V_{h+1}^{*,\rho}$, we first solve regression problems to obtain the estimator for $\mathbb{V}_h \hat{V}_{k,h+1}^\rho$, which is denoted as $\bar{\mathbb{V}}_h \hat{V}_{k,h+1}^\rho$. Then we analyze the error between $\mathbb{V}_h V_{h+1}^{*,\rho}$ and $\bar{\mathbb{V}}_h \hat{V}_{k,h+1}^\rho$ to finish the construction. Different from (5.2) in [24], the variance estimator here is not trivially constructed from subtracting a specific penalty term because we should guarantee the monotonicity of estimated variance for the online exploration. The variance of estimated optimistic value function $\hat{V}_{k,h+1}^\rho$ can be decomposed into

$$\big[\mathbb{V}_h \hat{V}_{k,h+1}^\rho\big](s,a) = \big[\mathbb{P}_h^0 \big(\hat{V}_{k,h+1}^\rho\big)^2\big](s,a) - \big(\big[\mathbb{P}_h^0 \hat{V}_{k,h+1}^\rho\big](s,a)\big)^2. \tag{3.5}$$

---

**Algorithm 1** Weighted Distributionally Robust Iterative Value Estimation with UCB (We-DRIVE-U)

---

1: **Initialization:** confidence parameters $\beta, \bar{\beta}, \widetilde{\beta} > 0$ and regularization $\lambda > 0$. $k_{\text{last}} = 0$. For each stage $h \in [H]$, initialize $\boldsymbol{\Sigma}_{0,h} = \boldsymbol{\Sigma}_{1,h} = \boldsymbol{\Lambda}_{1,h} = \lambda \mathbf{I}$ and the upper and lower estimation $\hat{Q}^\rho_{0,h}(\cdot, \cdot) = H, \check{Q}^\rho_{0,h}(\cdot, \cdot) = 0$

2: **for** episode $k = 1, \cdots, K$ **do**

3:     Receive the initial state $s_1^k$

4:     Set $\hat{V}^\rho_{k,H+1}(\cdot) \leftarrow 0, \check{V}^\rho_{k,H+1}(\cdot) \leftarrow 0$

5:     **if** there exists a stage $h' \in [H]$ such that $\det(\boldsymbol{\Sigma}_{k,h'}) \geq 2\det(\boldsymbol{\Sigma}_{k_{\text{last}},h'})$ **then**

6:         **for** stage $h = H, \cdots, 1$ **do**

7:             For $h = H, \hat{\boldsymbol{\nu}}^{\rho,k}_h \leftarrow 0, \check{\boldsymbol{\nu}}^{\rho,k}_h \leftarrow 0$; otherwise compute $\hat{\nu}^{\rho,k}_{h,i}, \forall i \in [d]$ according to (3.3) and $\check{\nu}^{\rho,k}_{h,i}, \forall i \in [d]$ according to (3.4).

8:             $\hat{Q}^\rho_{k,h}(s,a) \leftarrow \min\{r_h(s,a) + \boldsymbol{\phi}(s,a)^\top \hat{\boldsymbol{\nu}}^{\rho,k}_h + \hat{\Gamma}_{k,h}(s,a), \hat{Q}^\rho_{k-1,h}(s,a), H-h+1\} \mathbb{1}\{s \neq s_f\}$

9:             $\check{Q}^\rho_{k,h}(s,a) \leftarrow \max\{r_h(s,a) + \boldsymbol{\phi}(s,a)^\top \check{\boldsymbol{\nu}}^{\rho,k}_h - \check{\Gamma}_{k,h}(s,a), \check{Q}^\rho_{k-1,h}(s,a), 0\} \mathbb{1}\{s \neq s_f\}$

10:             Set the last updating episode $k_{\text{last}} \leftarrow k$

11:             $\hat{V}^\rho_{k,h}(s) \leftarrow \max_a \hat{Q}^\rho_{k,h}(s,a), \quad \check{V}^\rho_{k,h}(s) \leftarrow \max_a \check{Q}^\rho_{k,h}(s,a)$

12:             $\pi^k_h(s) \leftarrow \text{argmax}_{a \in \mathcal{A}} \hat{Q}^\rho_{k,h}(s,a)$

13:         **end for**

14:     **else**

15:         $\hat{V}^\rho_{k,h}, \check{V}^\rho_{k,h}, \pi^k_h \leftarrow \hat{V}^\rho_{k-1,h}, \check{V}^\rho_{k-1,h}, \pi^{k-1}_h, \forall h \in [H]$

16:     **end if**

17:     **for** stage $h = 1, \cdots, H$ **do**

18:         Take $a^k_h \leftarrow \pi^k_h(s^k_h)$ and receive $s^k_{h+1}$

19:         Calculate the estimated variance $\sigma_{k,h}$ according to (3.6) and $\bar{\sigma}_{k,h}$ according to (3.7)

20:         $\boldsymbol{\Sigma}_{k+1,h} \leftarrow \boldsymbol{\Sigma}_{k,h} + \bar{\sigma}^{-2}_{k,h} \boldsymbol{\phi}(s^k_h, a^k_h) \boldsymbol{\phi}(s^k_h, a^k_h)^\top, \boldsymbol{\Lambda}_{k+1,h} \leftarrow \boldsymbol{\Lambda}_{k,h} + \boldsymbol{\phi}(s^k_h, a^k_h) \boldsymbol{\phi}(s^k_h, a^k_h)^\top$

21:     **end for**

22: **end for**

---

Under Assumption 2.1, $\mathbb{P}^0_h(\hat{V}^\rho_{k,h+1})^2$ and $\mathbb{P}^0_h \hat{V}^\rho_{k,h+1}$ on the RHS of (3.5) are linear in $\boldsymbol{\phi}(s,a)$. Thus we can approximate the variance as $[\mathbb{V}_h \hat{V}^\rho_{k,h+1}](s,a) \approx [\bar{\mathbb{V}}_h \hat{V}^\rho_{k,h+1}](s,a) = [\boldsymbol{\phi}(s,a)^\top \widetilde{\boldsymbol{w}}^k_{h,2}]_{[0,H^2]} - [\boldsymbol{\phi}(s,a)^\top \hat{\boldsymbol{w}}^k_{h,1}]^2_{[0,H]}$, where $\hat{\boldsymbol{w}}^k_{h,1} = \min_{\boldsymbol{w} \in \mathbb{R}^d} \sum_{\tau=1}^{k-1} (\boldsymbol{w}^\top \boldsymbol{\phi}(s^\tau_h, a^\tau_h) - \hat{V}^\rho_{k,h+1}(s^\tau_{h+1}))^2 + \lambda \|\boldsymbol{w}\|^2_2$ and $\widetilde{\boldsymbol{w}}^k_{h,2} = \min_{\boldsymbol{w} \in \mathbb{R}^d} \sum_{\tau=1}^{k-1} (\boldsymbol{w}^\top \boldsymbol{\phi}(s^\tau_h, a^\tau_h) - (\hat{V}^\rho_{k,h+1}(s^\tau_{h+1}))^2)^2 + \lambda \|\boldsymbol{w}\|^2_2$. Different from the variance estimation in standard MDPs [10], we construct both $\widetilde{\boldsymbol{w}}^k_{h,2}$ and $\hat{\boldsymbol{w}}^k_{h,1}$ by solving vanilla ridge regressions, instead of variance-weighted ridge regressions. This specific choice of parameter estimation will simplify our analysis of the variance estimation error, while fully capture the variance information. Now we can construct $\sigma_{k,h}$, which is the estimated variance of the optimal robust value function $V^{*,\rho}_h$ in episode $k$, as follows

$$\sigma_{k,h} = \sqrt{[\bar{\mathbb{V}}_h \hat{V}^\rho_{k,h+1}](s^k_h, a^k_h) + E_{k,h} + d^3 H D_{k,h} + 1/2}, \tag{3.6}$$

where $E_{k,h}$ represents the error between the estimated variance and the true variance of $\hat{V}^\rho_{k,h+1}$, and $D_{k,h}$ represents the error between the true variance of $\hat{V}^\rho_{k,h+1}$ and the true variance of $V^{*,\rho}_{h+1}$. Formally, we define

$$E_{k,h} = \min\{\widetilde{\beta} \|\boldsymbol{\phi}(s^k_h, a^k_h)\|_{\boldsymbol{\Lambda}^{-1}_{k,h}}, H^2\} + \min\{2H\bar{\beta} \|\boldsymbol{\phi}(s^k_h, a^k_h)\|_{\boldsymbol{\Lambda}^{-1}_{k,h}}, H^2\},$$

$$D_{k,h} = \min\{4H(\boldsymbol{\phi}(s^k_h, a^k_h)^\top \hat{\boldsymbol{w}}^k_{h,1} - \boldsymbol{\phi}(s^k_h, a^k_h)^\top \check{\boldsymbol{w}}^k_{h,1} + 2\bar{\beta} \|\boldsymbol{\phi}(s^k_h, a^k_h)\|_{\boldsymbol{\Lambda}^{-1}_{k,h}}), H^2\},$$

where $\boldsymbol{\Lambda}_{k,h} = \lambda \mathbf{I} + \sum_{\tau=1}^{k-1} \boldsymbol{\phi}(s^\tau_h, a^\tau_h) \boldsymbol{\phi}(s^\tau_h, a^\tau_h)^\top$, $\bar{\beta} = \widetilde{\mathcal{O}}(H\sqrt{d\lambda} + \sqrt{d^3 H^3})$, $\widetilde{\beta} = \widetilde{\mathcal{O}}(H^2\sqrt{d\lambda} + \sqrt{d^3 H^6})$, and $\check{\boldsymbol{w}}^k_{h,1} = \min_{\boldsymbol{w} \in \mathbb{R}^d} \sum_{\tau=1}^{k-1} (\boldsymbol{w}^\top \boldsymbol{\phi}(s^\tau_h, a^\tau_h) - \check{V}^\rho_{k,h+1}(s^\tau_{h+1}))^2 + \lambda \|\boldsymbol{w}\|^2_2$. Finally, we construct weights for the variance-weighted ridge regression problem (3.2): $\forall k, h \in [K] \times [H]$,

$$\bar{\sigma}_{k,h} = \max\{\sigma_{k,h}, 1, \sqrt{2d^3 H^2} \|\boldsymbol{\phi}(s^k_h, a^k_h)\|^{1/2}_{\boldsymbol{\Sigma}^{-1}_{k,h}}\}. \tag{3.7}$$

Compared with the variance estimation for the value function in standard MDPs [10] which is $\bar{\sigma}_{k,h} = \max\left\{\sigma_{k,h}, H, 2d^3 H^2 \|\phi(s_h^k, a_h^k)\|_{\Sigma_{k,h}^{-1}}^{1/2}\right\}$, our variance estimation is tighter in both the second and the third terms. The second term in (3.7) is 1, instead of $H$. This is important in achieving a tighter dependence on $H$. The intuition is that, when $k$ is large, $\bar{\sigma}_{k,h}$ should be close to the variance of the optimal robust value function. This design is motivated by the 'Range Shrinkage' phenomenon unique to DRMDPs (see the discussion after Theorem 4.1 for details), which observes that the true variance is in the order of $\mathcal{O}(1)$ when $\rho = \mathcal{O}(1)$. To get a precise variance estimation, $\bar{\sigma}_{k,h}$ should be in the same order of the true variance. Moreover, a constant order lower bound on $\bar{\sigma}_{k,h}$ will also ensure the weight will not cause any inflation in the weighted regression (3.2). The third term in (3.7) to be also tighter than that of [10], while maintaining the same theoretical property.

### 3.3 Algorithm interpretation

Now we provide some discussions to interpret Algorithm 1.

**Remark 3.1.** We highlight that Algorithm 1 is the first algorithm adopting a 'rare-switching' update strategy for distributionally robust RL. Different from [10], the 'rare-switching' condition on Line 5 is set at the beginning of each episode. This is achieved by our variance estimator design, which is independent of the parameter update for $z_h^k(\alpha)$. The update rule on Line 5 determines whether to update robust Q-function estimations and switch to a new policy for the current episode, and leads to two advantages, 1) the number of times solving the ridge regression (3.2) and dual optimization (3.3) significantly decreases, which constitute the main computation cost of Algorithm 1, and 2) in real application scenarios where policy switching is costly or risky, Algorithm 1 possesses low policy switching property. We refer the readers to Proposition 4.7 and Remark 4.8 for more details.

**Remark 3.2.** On Line 7, we estimate $\nu_h^{\rho,k}$ element-wisely, and thus the estimator $\hat{\nu}_h^{\rho,k}$ is derived from $d$ separate variance-weighted ridge regressions (3.2) and dual optimizations (3.3). This leads to the specific form of bonus term $\hat{\Gamma}_{k,h}(s,a) = \beta \sum_{i=1}^d \phi_i(s,a)\sqrt{\mathbf{1}_i^\top \Sigma_{k,h}^{-1} \mathbf{1}_i}$, which is actually an upper bound of the robust estimation error (see Lemma D.4 and its proof) at episode $k$. Though the bonus term resembles that in [23], we highlight that the sampling covariance matrix $\Sigma_{k,h}$ in $\hat{\Gamma}_{k,h}(s,a)$ is indeed a variance-weighted one. The specific form of the bonus term leads to the new variance-weighted $d$-rectangular robust estimation error defined in (4.1).

**Remark 3.3.** On Line 8 and 9, we adopt a monotonic Q-function update strategy, such that the estimated optimistic (pessimistic) robust value function is monotonically decreasing (increasing) to the optimal robust value function. This strategy is to make sure that the variance estimator $\sigma_{k,h}$ at any episode $k \in [H]$ is a uniform upper bound for those in the subsequent episodes, which would be helpful in bounding the estimation error arising from the variance-weighted ridge regression (3.2). This idea is first introduced by [2] for standard tabular MDPs and then utilized by [10] for standard linear MDPs. This is the first time it is utilized in the online linear DRMDP setting, where the episodic estimation regime proposes additional requirement on the variance estimator construction compared to the offline setting studied by [24].

## 4 Suboptimality upper bound analysis

We now provide theoretical results on the upper bound on the suboptimality of Algorithm 1.

**Theorem 4.1.** Under Assumptions 2.1 and 2.3, set $\lambda = 1/H^2$, then for any fixed $\delta \in (0,1)$ and $\rho \in (0,1]$, with probability at least $1 - \delta$, the average suboptimality of We-DRIVE-U satisfies

$$\text{AveSubopt}(K) \le 2\sqrt{2H^3 \log(6/\delta)/K} + \frac{4\beta}{K} \underbrace{\sum_{k=1}^K \sum_{h=1}^H \sum_{i=1}^d \phi_{h,i}^k \sqrt{\mathbf{1}_i^\top \Sigma_{k,h}^{-1} \mathbf{1}_i}}_{\text{variance-weighted } d\text{-rectangular estimation error}}, \qquad (4.1)$$

where $\beta = \widetilde{\mathcal{O}}(\sqrt{d})$, $\phi_{h,i}^k$ is the $i$-th element of $\phi(s_h^k, a_h^k)$ and $\mathbf{1}_i$ is the $i$-th standard basis vector.

Recall from Remark 3.2, the quantity $\sum_{i=1}^d \phi_{h,i}^k \sqrt{\mathbf{1}_i^\top \Sigma_{k,h}^{-1} \mathbf{1}_i}$ in (4.1) comes from solving $d$ separate variance-weighted ridge regressions at step $h$ in episode $k$. A similar term also appears in the Theorem 5.1 of [23]. Differently, the the quantity $\sum_{i=1}^d \phi_{h,i}^k \sqrt{\mathbf{1}_i^\top \Sigma_{k,h}^{-1} \mathbf{1}_i}$ is based on the variance-weighted sampling covariance matrix $\Sigma_{k,h}$, rather than the vanilla sampling covariance matrix

$\mathbf{\Lambda}_{k,h}$ as in [23]. In order to further bound (4.1), we need to take a closer examination of the variance estimator. Intuitively, when episode $k$ is large, the variance estimator should be close to the variance of the optimal robust value function. Recent study [24] shows a 'Range shrinkage' phenomenon in the $d$-rectangular linear DRMDP (refer to Lemma H.10), stating that the range of any robust value function satisfies $\max_{s \in \mathcal{S}} V_h^{\pi,\rho}(s) - \min_{s \in \mathcal{S}} V_h^{\pi,\rho}(s) \leq \min\{1/\rho, H\}, \forall(\pi, h, \rho) \in \Pi \times [H] \times (0, 1]$. This implies that the variance of the optimal robust value function is upper bounded by $\min\{1/\rho, H\}$. Thus, when $k$ is large, we can expect $\bar{\sigma}_{k,h} \lesssim \widetilde{\mathcal{O}}(\min\{1/\rho, H\})$ and hence $\mathbf{\Sigma}_{k,h}^{-1} \preceq \widetilde{\mathcal{O}}(\min\{1/\rho^2, H^2\})\mathbf{\Lambda}_{k,h}^{-1}$. To this end, next we rigorously bound (4.1) under the same setting as the Corollary 5.3 of [23], and formally show that the variance information leads to a tighter dependence on $H$ compared to [23].

**Theorem 4.2.** Assume that there exists an absolute constant $c > 0$, such that for all $(\pi, h) \in \Pi \times [H]$

$$\mathbb{E}_\pi^{P^0}\left[\phi(s_h, a_h)\phi(s_h, a_h)^\top\right] \geq c/d \cdot \mathbf{I}. \tag{4.2}$$

Then under the same setting in Theorem 4.1 and the assumption in (4.2), for any fixed $\delta \in (0, 1)$, with probability at least $1 - \delta$, the average suboptimality of We-DRIVE-U is upper bounded by $\widetilde{\mathcal{O}}\left((dH \cdot \min\{1/\rho, H\} + H^{3/2})/\sqrt{K} + d^{15}H^{13}/K\right)$.

**Remark 4.3.** When $d \geq H$ and the total number of episodes $K$ is sufficiently large, the average suboptimality can be simplified as $\widetilde{\mathcal{O}}\{dH\min\{1/\rho, H\}/\sqrt{K}\}$. Note that under the same assumption in (4.2), [23] prove that the average suboptimality of their algorithm DR-LSVI-UCB is of the order $\widetilde{\mathcal{O}}(d^2H^2/\sqrt{K})$. Thus, We-DRIVE-U improves the state-of-the-art result by $\mathcal{O}(dH/\min\{1/\rho, H\})$. Moreover, we highlight that the upper bound in Theorem 4.2 depends on the uncertainty level $\rho$, which arises from the 'Range Shrinkage' phenomenon. When $\rho$ increases from 0 to 1, the suboptimality decreases up to a factor of $\mathcal{O}(H)$.

**Remark 4.4.** The assumption (4.2) is actually imposed on the DRMDP, requiring that the environment we encounter is exploratory enough. We would like to note that this assumption is necessary in deriving our upper bound, since the elliptical potential lemma [1, Lemma 11], which is critical in deriving upper bounds in linear bandits and linear MDPs, does not apply in the analysis of linear DRMDPs. We note that the previous work [23] also used this assumption to get the final upper bound for their algorithm. Moreover, the assumption (4.2) can be deemed as an online version of the well-known full-type coverage assumption on the offline dataset in offline (non-) robust RL. Specifically, in the context of standard offline RL, [5, 44, 47] assume the offline dataset should cover the distribution measure induced by any policy under the nominal environment. In the context of offline robust RL, [32, 31, 57] assume that the offline dataset should cover the distribution measure induced by any policy under any transition kernel in the uncertainty set. It would be an interesting future research direction to study if assumption (4.2) can be relaxed.

Next, we study the deployment complexity of Algorithm 1, which constitutes two sources of cost. The first source is the policy switching cost, say, the total number of changes in the exploration policy. This might be the main bottleneck in applications where changing the exploration policy is costly or risky [3, 45]. The second source is the computation cost in solving the dual optimization in (3.3). Recall in Remark 4.8 we discuss that Algorithm 1 adopts the 'rare-switching' update strategy, which significantly reduces the two sources of cost. Next, we formally define them as follows.

**Definition 4.5** (Global Switching Cost). We define the *global switching cost* of an algorithm that runs for $K$ episodes as $N_{\text{switch}}^{gl} := \sum_{k=1}^K \mathbb{1}\{\pi_k \neq \pi_{k+1}\}$.

**Definition 4.6** (Dual Oracle). We assume access to a maximization oracle, which takes a function $z : [0, H] \to \mathbb{R}$ and a fixed constant $\rho > 0$ as input, and outputs the maximum value $z_{\max}$ and the maximizer $\alpha_{\max}$ defined as $z_{\max} = \max_{\alpha \in [0,H]}\{z(\alpha) - \rho\alpha\}$ and $\alpha_{\max} = \text{argmax}_{\alpha \in [0,H]}\{z(\alpha) - \rho\alpha\}$. For an algorithm, we define the *oracle complexity* as the number of calls of the dual oracle. Finally, we show that We-DRIVE-U admits low switching cost and low oracle complexity.

Next, we formally present theoretical results on the deployment complexity of Algorithm 1.

**Proposition 4.7.** Under the same setting as Theorem 4.1, the switching cost of We-DRIVE-U is upper bounded by $dH \log(1 + H^2K)$, and the oracle complexity of We-DRIVE-U is upper bounded by $2d^2H \log(1 + H^2K)$.

**Remark 4.8.** The switching cost of the state-of-the-art algorithm DR-LSVI-UCB [23] is $K$ and the oracle complexity is $dK$. Thus, We-DRIVE-U improves both the switching and oracle cost by a factor

of $K$. Different from standard linear MDPs, where the main computation complexity only comes from the policy update [10], in the linear DRMDP setting, the calls of dual oracle, besides policy updates, are also a main source of computational burden. The update rule on Line 5 guarantees that We-DRIVE-U calls the dual oracle and updates the policy only when the criterion is met. Actually, Algorithm 1 is the first DRMDP algorithm that admits low deployment complexity.

# 5 Information-theoretic lower bound analysis

According to Theorem 4.2, when $\rho = \mathcal{O}(1)$, the suboptimality of We-DRIVE-U is of order $\mathcal{O}(dH/\sqrt{K})$. After multiplying $K$ to recover the cumulative suboptimality, it is smaller than the minimax lower bound for standard linear MDP, $\Omega(d\sqrt{H^3 K})$ [62]. To assess the optimality of We-DRIVE-U, next we present an information-theoretic lower bound for online linear DRMDPs.

**Theorem 5.1.** Let uncertainty level $\rho \in (0, 3/4]$, $H \geq 6$, and $K \geq 9d^2 H/32$. Then for any algorithm, there exists a $d$-rectangular linear DRMDP parameterized by $\boldsymbol{\xi} = (\boldsymbol{\xi}_1, \cdots, \boldsymbol{\xi}_{H-1})$ such that the expected average suboptimality is lower bounded as follows:

$$\mathbb{E}_{\boldsymbol{\xi}}[\text{AveSubopt}(M_{\boldsymbol{\xi}}, K)] \geq \Omega\big((dH^{1/2} \cdot \min\{1/\rho, H\})/\sqrt{K}\big), \tag{5.1}$$

where $\mathbb{E}_{\boldsymbol{\xi}}$ denotes the expectation over the randomness of the algorithm and the nominal environment.

**Remark 5.2.** We highlight that the lower bound (5.1) depends on the uncertainty level $\rho$, which is a distinctive characteristic for DRMDPs. Theorem 5.1 implies We-DRIVE-U is near-optimal up to a factor of $\tilde{\mathcal{O}}(\sqrt{H})$. Moreover, when $\rho \to 0$, the linear DRMDP degrades to a standard linear MDP, and (5.1) matches the information-theoretic lower bound, $\Omega(d\sqrt{H^3 K})$, for standard linear MDPs [62] after multiplying $K$ to recover the cumulative regret. When $\rho = \mathcal{O}(1)$, (5.1) reduces to $\Omega(dH^{1/2}/\sqrt{K})$, which is $\mathcal{O}(H)$ smaller than the lower bound for standard linear MDPs.

Next, we investigate the $\widetilde{\mathcal{O}}(\sqrt{H})$ gap between the upper and lower bounds and propose a conjecture on its origin. In the analysis of non-robust MDPs [2, 18, 10] and tabular DRMDPs [25], a tight dependence on $H$ is often achieved by exploiting the total variance law of the value function at each episode. Currently, we bound each term in the variance-weighted $d$-rectangular estimation error in (4.1) separately. A tight upper bound might be achieved by first bounding the variance-weighted $d$-rectangular estimation error as a whole by the square root of the total variance and then invoking the total variance law. In particular, inspired by the total variance law in Lemma C.6 of [25], the total variance should be in the order of $\mathcal{O}(H \min\{1/\rho, H\})$. Together with an additional $\sqrt{H}$ arising in the suboptimality analysis, we conjecture the dependence of the upper bound on $H$ could be improved to $\mathcal{O}(\sqrt{H^2 \min\{1/\rho, H\}})$.

When the uncertainty level is small, i.e., $\rho = \mathcal{O}(1/H)$, the conjectured result leads to an upper bound on the suboptimality that depends on $\mathcal{O}(H^{3/2})$, matching the current lower bound we present in (5.1). This suggests that our current lower bound is tight and the total variance analysis could improve our upper bound. When the uncertainty level is relatively large, i.e., $\rho = \mathcal{O}(1)$, the conjectured upper bound is $\mathcal{O}(H)$, which matches the current upper bound in Theorem 4.2. This means the total variance analysis does not further improve the upper bound, and we suspect that a tighter lower bound is instead necessary. This leaves an interesting open problem for future study.

# 6 Conclusion

This paper advanced the study of online $d$-rectangular linear DRMDPs by establishing a tighter regret upper bound and the first lower regret bound under this setting. We introduced We-DRIVE-U, a novel variance-aware algorithm that leverages variance-weighted ridge regression and low policy-switching techniques. Under standard MDP structure assumptions, we proved We-DRIVE-U achieves an average suboptimality of $\widetilde{\mathcal{O}}(dH \min\{1/\rho, H\}/\sqrt{K})$, improving the state-of-the-art by $\widetilde{\mathcal{O}}(dH/\min\{1/\rho, H\})$. We also established an information-theoretic lower bound of $\Omega(dH^{1/2} \min\{1/\rho, H\}/\sqrt{K})$, which implies We-DRIVE-U's near-optimality up to $\mathcal{O}(\sqrt{H})$. Furthermore, We-DRIVE-U reduces computational complexity with $\mathcal{O}(dH \log(1 + H^2 K))$ policy switches and $\mathcal{O}(d^2 H \log(1 + H^2 K))$ oracle complexity, which outperforms existing methods by a factor of $K$. We also conducted numerical experiments to validate the robustness and improved performance over existing algorithms, which is presented in Appendix B.

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

## A  Related work

**Distributionally Robust MDPs**    There has been a large body of works studying DRMDPs under various settings, for instance, the setting of planning and control [48, 46, 54, 27, 9] where the exact transition model is known, the setting with a generative model [63, 52, 30, 49, 36, 51], the offline setting [32, 35, 4, 38] and the online setting [6, 23, 25]. Among tabular DRMDPs, the most relevant studies to ours are [36, 25]. In particular, [36] studies tabular DRMDPs with TV uncertainty sets. They provide an information-theoretic lower bound, as well as a matching upper bound on the sample complexity. The key message is that the sample complexity bounds depend on the uncertainty level, and when the uncertainty level is of constant order, policy learning in a DRMDP requires less samples than in a standard MDP. Further, [25] studies the online tabular DRMDPs with TV uncertainty sets, they provide an algorithm that achieves the near-optimal sample complexity under a vanishing minimal value assumption to circumvent the curse of support shift.

**Online Linear MDPs and Linear DRMDPs**    The nominal model studied in our paper is assumed to be a linear MDP with a simplex feature space. There is a line of works studying online linear MDPs [50, 19, 28, 55, 43, 11, 40, 15], and the minimax optimality of this setting is studied in the recent work of [10]. In particular, they adopt the variance-weighted ridge regression scheme and the 'rare-switching' policy update strategy in their algorithm design. The setting of online linear DRMDP is relatively understudied, with both the lower bound and the near-optimal upper bound remain elusive. Specifically, the only work studies the online linear DRMDP setting is [23]. Under the TV uncertainty set, their algorithm, DR-LSVI-UCB, achieves an average suboptimality of the order $\tilde{O}(d^2 H^2/\sqrt{K})$. However, recent evidence from studies [24, 41] on offline linear DRMDPs suggests that this rate is far from optimality. In particular, [24] proves that their algorithm, VA-DRPVI, achieves an upper bound on the suboptimality in the order of $\tilde{O}(dH \min\{1/\rho, H\}/\sqrt{K})$. Nonetheless, their algorithm and analysis are based on a pre-collected offline dataset which satisfies some coverage assumption, and thus cannot be utilized in the online setting, where a strategy on data collection is required to deal with the challenge of exploration and exploitation trade-off.

## B  Experiments on simulated linear DRMDPs

### B.1  Simulated linear DRMDPs

We conduct numerical experiments to illustrate the performances of our proposed algorithm, We-DRIVE-U, and compare it with the state-of-the-art algorithm for $d$-rectangular linear DRMDPs, DR-LSVI-UCB [23], as well as their non-robust counterpart, LSVI-UCB [19]. All numerical experiments were conducted on a MacBook Pro with a 2.6 GHz 6-Core Intel CPU.

We leverage the simulated linear MDP setting proposed by [23]. For completeness, we recall the experiment setting as follows. The source and target linear MDP environment are shown in Figure 1(a) and Figure 1(b). The state space is $\mathcal{S} = \{x_1, \cdots, x_5\}$ and action space $\mathcal{A} = \{-1, 1\}^4 \subset \mathbb{R}^4$. At each episode, the initial state is always $x_1$, and it can transit to $x_2, x_4, x_5$ with probability defined in the figures. $x_2$ is an intermediate state from which the next state can be $x_3, x_4, x_5$. $x_4$ is the fail state with reward 0 and $x_5$ is an absorbing state with reward 1. For the reward functions and transition probabilities, they are designed to depend on $\langle \boldsymbol{\xi}, a \rangle$, where $\boldsymbol{\xi} \in \mathbb{R}^4$ is a hyperparameter controls the MDP instances. The target environment is constructed by only perturbing the transition probability at $x_1$ of the source domain, and the extend of perturbation is controlled by a hyperparameter $q \in (0, 1)$. We refer more details on the construction of the linear DRMDP to the Supplementary A.1 of [23].

We set $\boldsymbol{\xi} = (1/\|\boldsymbol{\xi}\|_1, 1/\|\boldsymbol{\xi}\|_1, 1/\|\boldsymbol{\xi}\|_1, 1/\|\boldsymbol{\xi}\|_1)^\top$ and consider different choices of $\|\boldsymbol{\xi}\|_1$ from the set $\{0.1, 0.2, 0.3\}$. Following the implementation in [23], we use heterogeneous uncertainty level and set $\rho_{1,4} = 0.5$ and $\rho_{h,i} = 0$ for all other cases. We set the number of interactions with the nominal environment to 200. We evaluate policies learned by We-DRIVE-U, DR-LSVI-UCB [23] and LSVI-UCB [19] by the accumulative rewards achieved in the target domain, which are illustrated in Figure 2. Figure 2 shows that: 1) policies learned by We-DRIVE-U are robust to environmental perturbation, and the extent of the robustness depends on the pre-specified parameter $\rho$; 2) In most cases, We-DRIVE-U outperforms DR-LSVI-UCB, meaning it being more robust to environment perturbation. Moreover, Table 1 demonstrates the low-switching property of We-DRIVE-U. During 200 interactions of the training process, We-DRIVE-U switches policies only around 24 times,

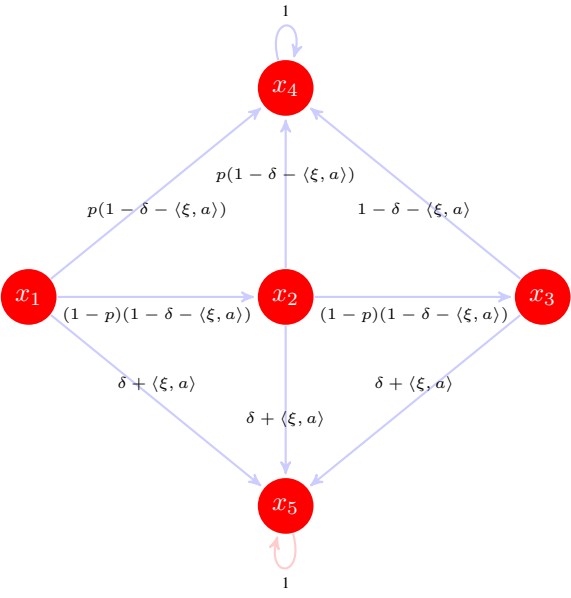

(a) The source environment.

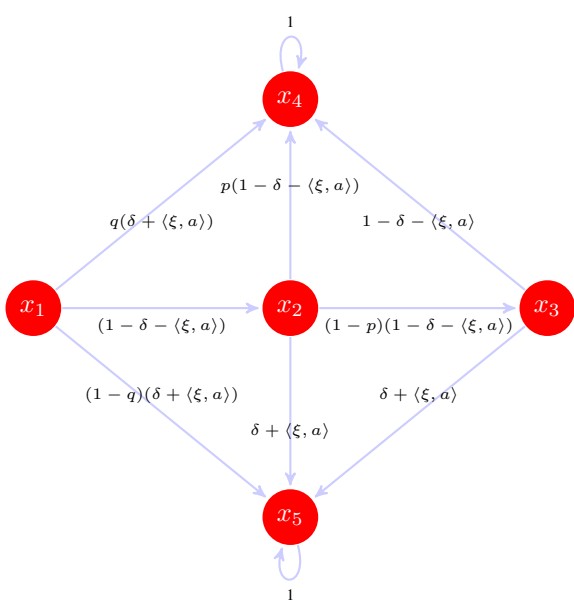

(b) The target environment.

Figure 1: The source and the target linear MDP environments. The value on each arrow represents the transition probability. For the source MDP, there are five states and three steps, with the initial state being $x_1$, the fail state being $x_4$, and $x_5$ being an absorbing state with reward 1. The target MDP on the right is obtained by perturbing the transition probability at the first step of the source MDP, with others remaining the same.

which stands in stark contrast to the 200 policy switches by LSVI-UCB and DR-LSVI-UCB. These numerical results prove the superiority of our proposed algorithm We-DRIVE-U and align well with our theoretical findings. All numerical experiments were conducted on a MacBook Pro with a 2.6 GHz 6-Core Intel CPU.

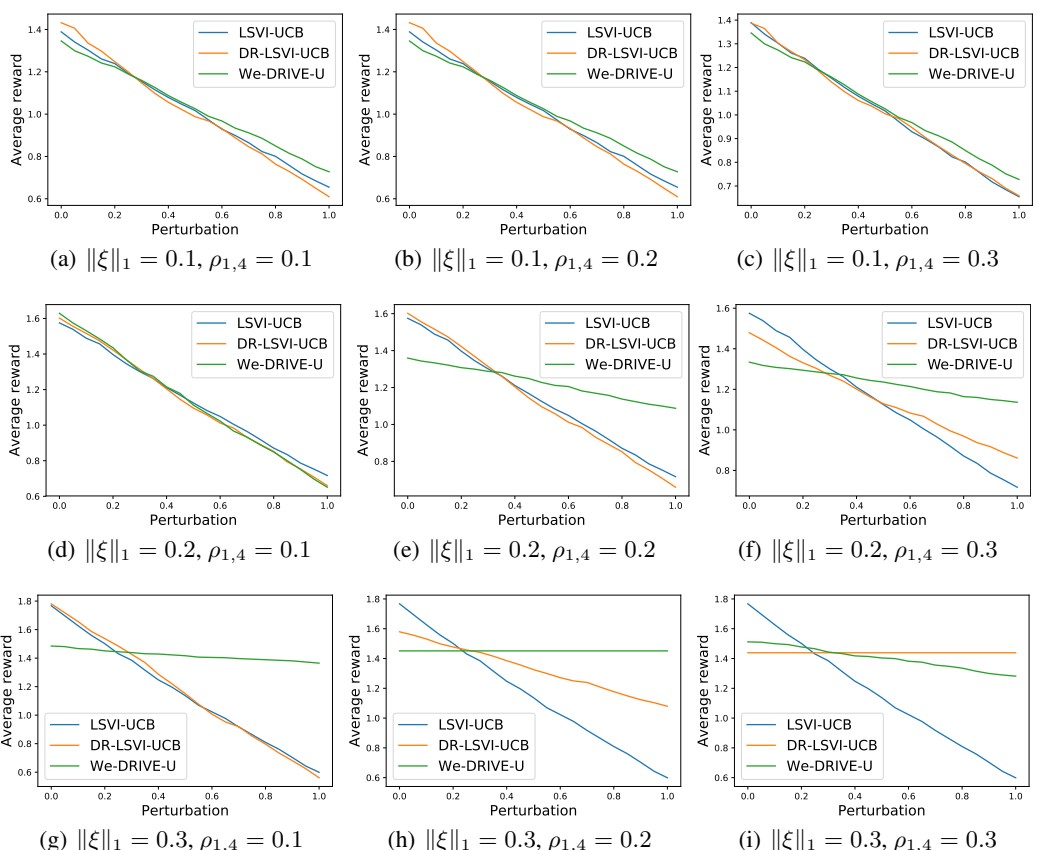

Figure 2: Simulation results under different source domains. The $x$-axis represents the perturbation level corresponding to different target environments. $\rho_{1,4}$ is the input uncertainty level for our We-DRIVE-U algorithm. $\|\xi\|_1$ is the hyperparameter of the linear DRMDP environment.

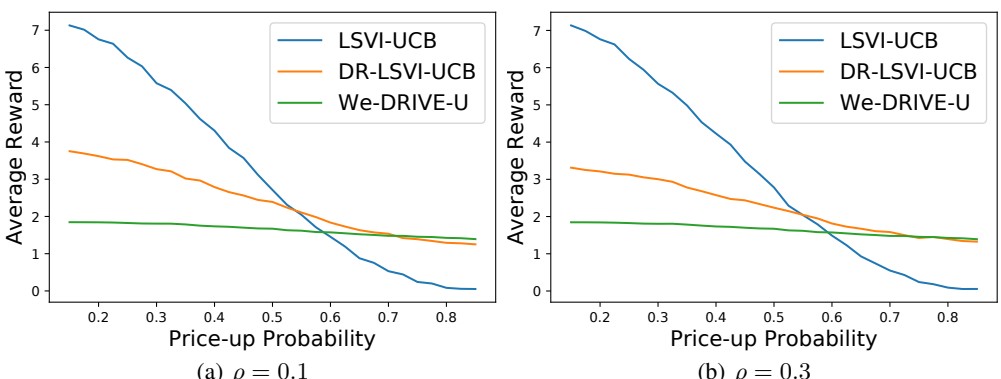

Figure 3: Results for the simulated American put option problem. $\rho$ is the uncertainty level in We-DRIVE-U.

Table 1: Simulation results of the switch complexity of We-DRIVE-U. We present the average policy switch times of We-DRIVE-U during 200 interactions with the nominal environment, averaged over 10 replications. As a comparison, the policy switch times for LSVI-UCB and DR-LSVI-UCB are both **200** under each setting.

|  | $\rho$=0.1 | $\rho$=0.2 | $\rho$=0.3 |
|---|---|---|---|
| $\|\xi\|_1$=0.1 | 23.8 | 24.0 | 23.8 |
| $\|\xi\|_1$=0.2 | 24.2 | 24.4 | 24.0 |
| $\|\xi\|_1$=0.3 | 24.3 | 23.6 | 24.8 |

## B.2  Simulated American put option

We additionally conduct a simulation study in the American put option environment (see more details in Section 6.2 of [1]), which under the hood is not a linear MDP. By manually constructing $\phi$, we show in Figure 3 that our algorithm still achieves some degree of robustness.

## C  Proof of Proposition 2.2

*Proof.* We instantiate the hard example in Example 3.1 of [25] in terms of the formulation of $d$-rectangular linear DRMDP satisfying Assumption 2.1. Consider two $d$-rectangular linear DRMDPs , $\mathcal{M}_0$ and $\mathcal{M}_1$. The state space $\mathcal{S} = \{s_{\text{good}}, s_{\text{bad}}\}$, and the action space is $\mathcal{A} = \{0, 1\}$. We define the feature mapping as

$$\phi^\varrho(s_{\text{good}}, a) = \begin{pmatrix} 1 \\ 0 \\ 0 \\ 0 \\ 0 \end{pmatrix}, \forall a \in \mathcal{A}, \ \phi^\varrho(s_{\text{bad}}, 0) = \begin{pmatrix} 0 \\ p(1-\varrho) \\ q\varrho \\ (1-p)(1-\varrho) \\ (1-q)\varrho \end{pmatrix}, \ \phi^\varrho(s_{\text{bad}}, 1) = \begin{pmatrix} 0 \\ p\varrho \\ q(1-\varrho) \\ (1-p)\varrho \\ (1-q)(1-\varrho) \end{pmatrix},$$

where $\varrho \in \{0, 1\}$ is the index of the $d$-rectangular linear DRMDP instance. Define the factor distributions $\boldsymbol{\mu} = (\delta_{s_{\text{good}}}, \delta_{s_{\text{good}}}, \delta_{s_{\text{good}}}, \delta_{s_{\text{bad}}}, \delta_{s_{\text{bad}}})^\top$ and the reward parameter $\boldsymbol{\theta} = (1, 0, 0, 0, 0)^\top$. Then it is trivial to check that equipped with the $d$-rectangular TV divergence uncertainty set, this example recover the hard example in Example 3.1 of [25]. $\square$

## D  Proof of the Upper Bound on the Suboptimality of We-DRIVE-U

In this section, we present the proofs of our main theoretical results Theorems 4.1 and 4.2. We start with presenting the technical lemmas in Appendix D.1, and then we derive the upper bound on the suboptimality of We-DRIVE-U in Appendices D.2 and D.3.

### D.1  Technical Lemmas

**Definition D.1** (Good event). Under Assumptions 2.1 and 2.3, then for any fixed $\delta \in (0, 1)$, $\alpha' \in [0, H]$ and $\rho \in (0, 1]$, we define $\mathcal{E}_h$ be the event that for all episode $k \in [K]$, stage $h \leq h' \leq H$,

$$\left\| \sum_{\tau=1}^{k-1} \bar{\sigma}_{\tau,h'}^{-2} \phi_{h'}^\tau \Big[ \big[\hat{V}_{k,h'+1}^\rho(s_{h'+1}^\tau)\big]_{\alpha'} - \big[\mathbb{P}_{h'}^0 \big[\hat{V}_{k,h'+1}^\rho\big]_{\alpha'}\big](s_{h'}^\tau, a_{h'}^\tau) \Big] \right\|_{\boldsymbol{\Sigma}_{k,h'}^{-1}} \leq \gamma, \qquad \text{(D.1)}$$

where $\gamma = \widetilde{\mathcal{O}}(\sqrt{d})$.

**Lemma D.2.** We define $\bar{\mathcal{E}}$ as the event that the following inequalities hold for all $(s, a) \in \mathcal{S} \times \mathcal{A}$, $k \in [K]$, $h \in [H]$,

$$\big|\phi(s, a)^\top \hat{\boldsymbol{w}}_{h,1}^k - \big[\mathbb{P}_h^0 \hat{V}_{k,h+1}^\rho\big](s, a)\big| \leq \bar{\beta}\sqrt{\phi(s, a)^\top \boldsymbol{\Lambda}_{k,h}^{-1} \phi(s, a)},$$

$$\big|\phi(s, a)^\top \check{\boldsymbol{w}}_{h,1}^k - \big[\mathbb{P}_h^0 \check{V}_{k,h+1}^\rho\big](s, a)\big| \leq \bar{\beta}\sqrt{\phi(s, a)^\top \boldsymbol{\Lambda}_{k,h}^{-1} \phi(s, a)},$$

$$\left|\phi(s,a)^\top \widetilde{\boldsymbol{w}}_{h,2}^k - \left[\mathbb{P}_h^0 \big(\hat{V}_{k,h+1}^\rho\big)^2\right](s,a)\right| \leq \widetilde{\beta}\sqrt{\phi(s,a)^\top \boldsymbol{\Lambda}_{k,h}^{-1}\phi(s,a)},$$

where $\bar{\beta} = \widetilde{\mathcal{O}}\big(H\sqrt{d\lambda} + \sqrt{d^3 H^3}\big)$ and $\widetilde{\beta} = \widetilde{\mathcal{O}}\big(H^2\sqrt{d\lambda} + \sqrt{d^3 H^6}\big)$. Then event $\bar{\mathcal{E}}$ holds with probability at least $1 - \delta$.

**Lemma D.3** (Variance error). On the event $\mathcal{E}_{h+1}$ and $\bar{\mathcal{E}}$, for all episode $k \in [K]$, the estimated variance satisfies

$$\left|\big[\bar{\mathbb{V}}_h \hat{V}_{k,h+1}^\rho\big]\big(s_h^k, a_h^k\big) - [\mathbb{V}_h \hat{V}_{k,h+1}^\rho]\big(s_h^k, a_h^k\big)\right| \leq E_{k,h},$$

$$\left|\big[\bar{\mathbb{V}}_h \hat{V}_{k,h+1}^\rho\big]\big(s_h^k, a_h^k\big) - [\mathbb{V}_h V_{h+1}^{*,\rho}]\big(s_h^k, a_h^k\big)\right| \leq E_{k,h} + D_{k,h}.$$

Thus we also have

$$\bar{\sigma}_{k,h}^2 \geq \big[\bar{\mathbb{V}}_h \hat{V}_{k,h+1}^\rho\big]\big(s_h^k, a_h^k\big) + E_{k,h} + D_{k,h} \geq \big[\mathbb{V}_h V_{h+1}^{*,\rho}\big]\big(s_h^k, a_h^k\big).$$

**Lemma D.4.** For any fixed policy $\pi$, on the event $\mathcal{E}_h$ and $\bar{\mathcal{E}}$, for all $(s,a,k) \in \mathcal{S}/\{s_f\} \times \mathcal{A} \times [K]$, for stage $h \leq h' \leq H$, we have

$$\big(r_{h'}(s,a) + \phi(s,a)^\top \hat{\boldsymbol{\nu}}_{h'}^{\rho,k}\big) - Q_{h'}^{\pi,\rho}(s,a) = \inf_{P_{h'}(\cdot|s,a)\in\mathcal{U}_{h'}^\rho(s,a;\boldsymbol{\mu}_{h'}^0)} \big[\mathbb{P}_{h'} \hat{V}_{k,h'+1}^\rho\big](s,a)$$
$$- \inf_{P_{h'}(\cdot|s,a)\in\mathcal{U}_{h'}^\rho(s,a;\boldsymbol{\mu}_{h'}^0)} \big[\mathbb{P}_{h'} V_{h'+1}^{\pi,\rho}\big](s,a) + \Delta_{h'}^k(s,a),$$

where $\Delta_{h'}^k(s,a)$ that satisfies $|\Delta_{h'}^k(s,a)| \leq \hat{\Gamma}_{k,h'}(s,a) = \beta \sum_{i=1}^d \phi_i(s,a)\sqrt{\boldsymbol{1}_i^\top \boldsymbol{\Sigma}_{k,h'}^{-1}\boldsymbol{1}_i}$ where $\beta = \widetilde{\mathcal{O}}\big(\sqrt{\lambda d}H + \sqrt{d}\big)$.

**Lemma D.5** (Optimism and pessimism). On the event $\mathcal{E}_h$ and $\bar{\mathcal{E}}$, for all episode $k \in [K]$ and stage $h \leq h' \leq H$, for all $(s,a) \in \mathcal{S} \times \mathcal{A}$, we have $\hat{Q}_{k,h'}^\rho(s,a) \geq Q_{h'}^{*,\rho}(s,a) \geq \check{Q}_{k,h'}^\rho(s,a)$. In addition, we have $\hat{V}_{k,h'}^\rho(s) \geq V_{h'}^{*,\rho}(s) \geq \check{V}_{k,h'}^\rho(s)$.

**Lemma D.6.** On the event $\bar{\mathcal{E}}$, event $\mathcal{E} = \mathcal{E}_1$ holds with probability at least $1 - \delta$.

**Lemma D.7.** Under the assumption (4.2) and events $\mathcal{E}$ and $\bar{\mathcal{E}}$, for any $h \in [H]$, set $\lambda = 1/H^2$ and $\rho \in (0,1]$. Then when $k \geq \widetilde{K}$ where $\widetilde{K} = \widetilde{\mathcal{O}}\big(d^{15} H^{12}\big)$, with probability at least $1 - \delta$, then we have

$$\bar{\sigma}_{k,h}^2 \leq \mathcal{O}\Big(\min\Big\{\frac{1}{\rho^2}, H^2\Big\}\Big).$$

## D.2 Proof of Theorem 4.1

*Proof of Theorem 4.1.* Conditioned on the event $\mathcal{E}$ and $\bar{\mathcal{E}}$, we first do the following decomposition

$$\hat{V}_{k,h}^\rho\big(s_h^k\big) - V_h^{\pi^k,\rho}\big(s_h^k\big)$$
$$= \hat{Q}_{k,h}^\rho\big(s_h^k, a_h^k\big) - Q_h^{\pi^k,\rho}\big(s_h^k, a_h^k\big)$$
$$\leq r_h\big(s_h^k, a_h^k\big) + \phi\big(s_h^k, a_h^k\big)^\top \hat{\boldsymbol{\nu}}_h^{\rho,k_{\text{last}}} + \hat{\Gamma}_{k_{\text{last}},h}\big(s_h^k, a_h^k\big) - Q_h^{\pi^k,\rho}\big(s_h^k, a_h^k\big)$$
$$\leq \inf_{P_h(\cdot|s,a)\in\mathcal{U}_h^\rho(s,a;\boldsymbol{\mu}_h^0)} \big[\mathbb{P}_h \hat{V}_{k,h+1}^\rho\big]\big(s_h^k, a_h^k\big) - \inf_{P_h(\cdot|s,a)\in\mathcal{U}_h^\rho(s,a;\boldsymbol{\mu}_h^0)} \big[\mathbb{P}_h V_{h+1}^{\pi^k,\rho}\big]\big(s_h^k, a_h^k\big) + 2\hat{\Gamma}_{k_{\text{last}},h}\big(s_h^k, a_h^k\big)$$
$$\leq \inf_{P_h(\cdot|s,a)\in\mathcal{U}_h^\rho(s,a;\boldsymbol{\mu}_h^0)} \big[\mathbb{P}_h \hat{V}_{k,h+1}^\rho\big]\big(s_h^k, a_h^k\big) - \inf_{P_h(\cdot|s,a)\in\mathcal{U}_h^\rho(s,a;\boldsymbol{\mu}_h^0)} \big[\mathbb{P}_h V_{h+1}^{\pi^k,\rho}\big]\big(s_h^k, a_h^k\big) + 4\hat{\Gamma}_{k,h}\big(s_h^k, a_h^k\big),$$

where the first equality holds due to the selection of $\pi_h^k$, the first inequality holds due to the definition of $\hat{Q}_{k,h}^\rho$, the second inequality hold from Lemma D.4, the third inequality holds from Lemma H.2. Note that

$$\inf_{P_h(\cdot|s_h^k,a_h^k)\in\mathcal{U}_h^\rho(s_h^k,a_h^k;\boldsymbol{\mu}_h^0)} \big[\mathbb{P}_h \hat{V}_{k,h+1}^\rho\big]\big(s_h^k, a_h^k\big) - \inf_{P_h(\cdot|s_h^k,a_h^k)\in\mathcal{U}_h^\rho(s_h^k,a_h^k;\boldsymbol{\mu}_h^0)} \big[\mathbb{P}_h V_{h+1}^{\pi^k,\rho}\big]\big(s_h^k, a_h^k\big)$$
$$= \left\langle \phi(s_h^k, a_h^k), \left[\max_{\alpha_i \in [0,H]} \Big\{\mathbb{E}^{\mu_{h,i}^0}\big[\hat{V}_{k,h+1}^\rho(s)\big]_{\alpha_i} - \rho\alpha_i\Big\}\right]_{i\in[d]}\right\rangle$$
$$- \left\langle \phi(s_h^k, a_h^k), \left[\max_{\alpha_i \in [0,H]} \Big\{\mathbb{E}^{\mu_{h,i}^0}\big[V_{h+1}^{\pi^k,\rho}(s)\big]_{\alpha_i} - \rho\alpha_i\Big\}\right]_{i\in[d]}\right\rangle$$

$$\leq \left\langle \phi(s_h^k, a_h^k), \left[ \max_{\alpha_i \in [0,H]} \left\{ \mathbb{E}^{\mu_{h,i}^0}\left[\hat{V}_{k,h+1}^\rho(s)\right]_{\alpha_i} - \mathbb{E}^{\mu_{h,i}^0}\left[V_{h+1}^{\pi^k,\rho}(s)\right]_{\alpha_i} \right\} \right]_{i \in [d]} \right\rangle$$

$$\leq \left\langle \phi(s_h^k, a_h^k), \mathbb{E}^{\mu_h^0}\left[\hat{V}_{k,h+1}^\rho(s) - V_{h+1}^{\pi^k,\rho}(s)\right] \right\rangle$$

$$= \mathbb{P}_h^0\left[\hat{V}_{k,h+1}^\rho - V_{h+1}^{\pi^k,\rho}\right]\big]\left(s_h^k, a_h^k\right)$$

$$= \left[\mathbb{P}_h^0\left[\hat{V}_{k,h+1}^\rho - V_{h+1}^{\pi^k,\rho}\right]\right]\left(s_h^k, a_h^k\right) - \left[\hat{V}_{k,h+1}^\rho(s_{h+1}^k) - V_{h+1}^{\pi^k,\rho}(s_{h+1}^k)\right] + \left[\hat{V}_{k,h+1}^\rho(s_{h+1}^k) - V_{h+1}^{\pi^k,\rho}(s_{h+1}^k)\right],$$

where the second inequality holds from Lemma D.5. Then we have

$$\hat{V}_{k,h}^\rho\left(s_h^k\right) - V_h^{\pi^k,\rho}\left(s_h^k\right)$$

$$\leq \left[\hat{V}_{k,h+1}^\rho(s_{h+1}^k) - V_{h+1}^{\pi^k,\rho}(s_{h+1}^k)\right] + \left[\mathbb{P}_h^0\left[\hat{V}_{k,h+1}^\rho - V_{h+1}^{\pi^k,\rho}\right]\right]\left(s_h^k, a_h^k\right) - \left[\hat{V}_{k,h+1}^\rho(s_{h+1}^k) - V_{h+1}^{\pi^k,\rho}(s_{h+1}^k)\right]$$

$$+ 4\hat{\Gamma}_{k,h}\left(s_h^k, a_h^k\right), \tag{D.2}$$

Then by applying (D.2) iteratively and applying Azuma-Hoeffding inequality, with probability at least $1 - \delta/3$, we have

$$K \times \text{AveSubopt}(K) = \sum_{k=1}^K \left(V_1^{*,\rho}\left(s_1^k\right) - V_1^{\pi^k,\rho}\left(s_1^k\right)\right)$$

$$\leq \sum_{k=1}^K \left(\hat{V}_{k,1}^\rho\left(s_1^k\right) - V_1^{\pi^k,\rho}\left(s_1^k\right)\right)$$

$$\leq \sum_{k=1}^K \sum_{h=1}^H \left(\left[\mathbb{P}_h\left[\hat{V}_{k,h+1}^\rho - V_{h+1}^{\pi^k,\rho}\right]\right]\left(s_h^k, a_h^k\right) - \left[\hat{V}_{k,h+1}^\rho(s_{h+1}^k) - V_{h+1}^{\pi^k,\rho}(s_{h+1}^k)\right]\right)$$

$$+ \sum_{k=1}^K \sum_{h=1}^H 4\hat{\Gamma}_{k,h}\left(s_h^k, a_h^k\right)$$

$$\leq 2\sqrt{2H^3 K \log(6/\delta)} + 4\beta \sum_{k=1}^K \sum_{h=1}^H \sum_{i=1}^d \phi_i\left(s_h^k, a_h^k\right)\sqrt{\mathbf{1}_i^\top \mathbf{\Sigma}_{k,h}^{-1} \mathbf{1}_i},$$

where the first inequality holds from Lemma D.5, the second inequality holds from (D.2), the third inequality holds from Azuma-Hoeffding inequality and the definition of $\hat{\Gamma}_{k,h}\left(s_h^k, a_h^k\right)$. Finally, by taking probability union bound over $\mathcal{E}$ and $\bar{\mathcal{E}}$, with probability at least $1 - \delta$, we can get the result of Theorem 4.2,

$$\text{AveSubopt}(K) \leq 2\sqrt{2H^3 \log(6/\delta)/K} + 4\beta/K \sum_{k=1}^K \sum_{h=1}^H \sum_{i=1}^d \phi_{h,i}^k\sqrt{\mathbf{1}_i^\top \mathbf{\Sigma}_{k,h}^{-1} \mathbf{1}_i}.$$

This completes the proof. □

### D.3   Proof of Theorem 4.2

*Proof of Theorem 4.2.* Conditioned on the event $\mathcal{E}$ and $\bar{\mathcal{E}}$, we first do the decomposition as follows

$$K \times \text{AveSubopt}(K) = \sum_{k=1}^K \left(V_1^{*,\rho}\left(s_1^k\right) - V_1^{\pi^k,\rho}\left(s_1^k\right)\right)$$

$$= \sum_{k=1}^{\widetilde{K}} \left(V_1^{*,\rho}\left(s_1^k\right) - V_1^{\pi^k,\rho}\left(s_1^k\right)\right) + \sum_{k=\widetilde{K}+1}^K \left(V_1^{*,\rho}\left(s_1^k\right) - V_1^{\pi^k,\rho}\left(s_1^k\right)\right)$$

$$\leq H\widetilde{K} + \sum_{k=\widetilde{K}+1}^K \left(V_1^{*,\rho}\left(s_1^k\right) - V_1^{\pi^k,\rho}\left(s_1^k\right)\right).$$

Recall from (D.2) in the proof of Theorem 4.1, we have

$$\hat{V}_{k,h}^\rho\left(s_h^k\right) - V_h^{\pi^k,\rho}\left(s_h^k\right)$$

$$\leq \left[\hat{V}^{\rho}_{k,h+1}(s^k_{h+1}) - V^{\pi^k,\rho}_{h+1}(s^k_{h+1})\right] + \left[\mathbb{P}_h\left[\hat{V}^{\rho}_{k,h+1} - V^{\pi^k,\rho}_{h+1}\right]\right](s^k_h, a^k_h) - \left[\hat{V}^{\rho}_{k,h+1}(s^k_{h+1}) - V^{\pi^k,\rho}_{h+1}(s^k_{h+1})\right]$$
$$+ 4\hat{\Gamma}_{k,h}(s^k_h, a^k_h).$$

Then by applying (D.2) iteratively and applying Azuma-Hoeffding inequality, with probability at least $1 - \delta/4$, we have

$$K \times \mathrm{AveSubopt}(K) \leq H\widetilde{K} + \sum_{k=\widetilde{K}+1}^{K} \left(V^{*,\rho}_1(s^k_1) - V^{\pi^k,\rho}_1(s^k_1)\right)$$

$$\leq H\widetilde{K} + \sum_{k=\widetilde{K}+1}^{K} \left(\hat{V}^{\rho}_{k,1}(s^k_1) - V^{\pi^k,\rho}_1(s^k_1)\right)$$

$$\leq H\widetilde{K} + \sum_{k=\widetilde{K}+1}^{K} \sum_{h=1}^{H} \left(\left[\mathbb{P}^0_h\left[\hat{V}^{\rho}_{k,h+1} - V^{\pi^k,\rho}_{h+1}\right]\right](s^k_h, a^k_h)\right)$$

$$- \left[\hat{V}^{\rho}_{k,h+1}(s^k_{h+1}) - V^{\pi^k,\rho}_{h+1}(s^k_{h+1})\right]\right) + \sum_{k=\widetilde{K}+1}^{K} \sum_{h=1}^{H} 4\hat{\Gamma}_{k,h}(s^k_h, a^k_h)$$

$$\leq H\widetilde{K} + 2\sqrt{2H^3 K \log(8/\delta)} + 4\beta \sum_{k=\widetilde{K}+1}^{K} \sum_{h=1}^{H} \sum_{i=1}^{d} \phi_i(s^k_h, a^k_h) \sqrt{\mathbf{1}^\top_i \boldsymbol{\Sigma}^{-1}_{k,h} \mathbf{1}_i},$$

where the second inequality holds from Lemma D.5, the third inequality holds from (D.2) and the last inequality holds from Azuma-Hoeffding inequality and the definition of $\hat{\Gamma}_{k,h}(s^k_h, a^k_h)$. Based on (4.2) and Lemma D.7, with probability at least $1 - \delta/4$, we can further have

$$4\beta \sum_{k=\widetilde{K}+1}^{K} \sum_{h=1}^{H} \sum_{i=1}^{d} \phi_i(s^k_h, a^k_h) \sqrt{\mathbf{1}^\top_i \boldsymbol{\Sigma}^{-1}_{k,h} \mathbf{1}_i}$$

$$\leq 4c_1 \beta \min\left\{\frac{1}{\rho}, H\right\} \cdot \sum_{k=\widetilde{K}+1}^{K} \sum_{h=1}^{H} \sum_{i=1}^{d} \phi_i(s^k_h, a^k_h) \sqrt{\mathbf{1}^\top_i \boldsymbol{\Lambda}^{-1}_{k,h} \mathbf{1}_i}$$

$$\leq 4c_1 \beta \min\left\{\frac{1}{\rho}, H\right\} \cdot \sum_{k=\widetilde{K}+1}^{K} \sum_{h=1}^{H} \sum_{i=1}^{d} \phi_i(s^k_h, a^k_h) \sqrt{\lambda_{\max}(\boldsymbol{\Lambda}^{-1}_{k,h})}$$

$$\leq 4c_1 \beta \min\left\{\frac{1}{\rho}, H\right\} \cdot \sum_{k=\widetilde{K}+1}^{K} \sum_{h=1}^{H} \sqrt{\frac{1}{\lambda_{\min}(\boldsymbol{\Lambda}_{k,h})}}$$

$$\leq 4c_1 \beta \min\left\{\frac{1}{\rho}, H\right\} \cdot \sum_{k=\widetilde{K}+1}^{K} \sum_{h=1}^{H} \sqrt{\frac{2d}{k \cdot c}}$$

$$\leq 4c_1 \sqrt{2d} \beta \frac{H}{\sqrt{c}} \cdot \min\left\{\frac{1}{\rho}, H\right\} \cdot \int_{\widetilde{K}+1}^{K} \frac{1}{\sqrt{k-1}} dk$$

$$\leq 4c_1 \sqrt{2d} \beta \frac{H}{\sqrt{c}} \cdot \min\left\{\frac{1}{\rho}, H\right\} \cdot 2\sqrt{K}$$

$$\leq \widetilde{\mathcal{O}}\left(dH\sqrt{K} \cdot \min\left\{\frac{1}{\rho}, H\right\}\right),$$

where $c_1 > 0$ is an absolute constant. The first inequality holds from Lemma D.7, the third inequality holds because $\sum_{i=1}^{d} \phi_i(s, a) = 1$ and the fourth inequality holds due to (E.10) with $\widetilde{K} > 512/\eta^2 \log(dKH/\delta)$. Therefore, we can further bound the regret that

$$K \times \mathrm{AveSubopt}(K) \leq H\widetilde{K} + 2\sqrt{2H^3 K \log(8/\delta)} + 4\beta \sum_{k=\widetilde{K}+1}^{K} \sum_{h=1}^{H} \sum_{i=1}^{d} \phi_i(s^k_h, a^k_h) \sqrt{\mathbf{1}^\top_i \boldsymbol{\Sigma}^{-1}_{k,h} \mathbf{1}_i}$$

$$\leq \widetilde{\mathcal{O}}\Big(dH\sqrt{K}\cdot \min\Big\{\frac{1}{\rho}, H\Big\} + H^{\frac{3}{2}}\sqrt{K} + d^{15}H^{13}\Big).$$

Finally, by taking probability union bound over $\mathcal{E}$ and $\bar{\mathcal{E}}$, with probability at least $1-\delta$, we can bound the average suboptimality of We-DRIVE-U as follows

$$\text{AveSubopt}(K) \leq \widetilde{\mathcal{O}}\left(\frac{dH\cdot \min\big\{\frac{1}{\rho}, H\big\} + H^{\frac{3}{2}}}{\sqrt{K}} + \frac{d^{15}H^{13}}{K}\right). \tag{D.3}$$

We complete the proof by substituting $\eta = O(1/d)$ into (D.3). $\qquad\square$

# E  Proof of the Technical Lemmas

## E.1  Proof of Lemma D.2

Before the proof of Lemma D.2, we first present a lemma that defines the optimistic value function class and gives a upper bound for its covering number.

**Lemma E.1** (Function class covering number)**.** In Algorithm 1, for each episode $k \in [K]$ and $h \in [H]$, the optimistic value function $\hat{V}^{\rho}_{k,h}$ belongs to the following function class

$$\mathcal{V}_h = \Big\{V\Big|V(\cdot) = \max_a \max_{1\leq j \leq \ell}\min\Big\{r_h(\cdot, a) + \boldsymbol{\phi}(\cdot, a)^\top \mathbf{w}_j + \beta\sum_{i=1}^{d}\phi_i(\cdot, a)\sqrt{\mathbf{1}_i^\top \boldsymbol{\Gamma}_j \mathbf{1}_i}, H\Big\},$$

$$\|\mathbf{w}_j\| \leq L, \|\boldsymbol{\Gamma}_j\|_F \leq \lambda^{-1}\sqrt{d}\Big\},$$

where $\ell \leq dH\log(1 + K/\lambda)$ is the number of value function updates from Lemma F.1 and $L = 2H\sqrt{dK/\lambda}$ from Lemma F.2. Define $\mathcal{N}_\epsilon$ be the $\epsilon$-covering number of $\mathcal{V}_h$ with respect to the distance $\text{dist}(V_1, V_2) = \sup_s |V_1(s) - V_2(s)|$. Then the covering entropy can be bounded by

$$\log \mathcal{N}_\epsilon \leq d\ell \log(1 + 4L/\epsilon) + d^2\ell \log\big(1 + 8\sqrt{d}\beta^2/\lambda\epsilon^2\big).$$

*Proof of Lemma E.1.* For any two function $V_1, V_2 \in \mathcal{V}_h$, we can write $V_1, V_2$ as follows

$$V_1(\cdot) = \max_a \max_{1\leq j \leq \ell}\min\Big\{r_h(\cdot, a) + \boldsymbol{\phi}(\cdot, a)^\top \mathbf{w}_{1,j} + \beta\sum_{i=1}^{d}\phi_i(\cdot, a)\sqrt{\mathbf{1}_i^\top \boldsymbol{\Gamma}_{1,j} \mathbf{1}_i}, H\Big\},$$

$$V_2(\cdot) = \max_a \max_{1\leq j \leq \ell}\min\Big\{r_h(\cdot, a) + \boldsymbol{\phi}(\cdot, a)^\top \mathbf{w}_{2,j} + \beta\sum_{i=1}^{d}\phi_i(\cdot, a)\sqrt{\mathbf{1}_i^\top \boldsymbol{\Gamma}_{2,j} \mathbf{1}_i}, H\Big\},$$

where $\|\mathbf{w}_{1,j}\|, \|\mathbf{w}_{2,j}\| \leq L, \boldsymbol{\Gamma}_{1,j}, \boldsymbol{\Gamma}_{2,j} \preccurlyeq \lambda^{-1}\mathbf{I}$ and $\|\boldsymbol{\Gamma}_{1,j}\|_F, \|\boldsymbol{\Gamma}_{2,j}\|_F \leq \lambda^{-1}\sqrt{d}$. Then we have

$$\text{dist}(V_1, V_2) = \sup_s |V_1(s) - V_2(s)|$$

$$\leq \sup_{1\leq j \leq \ell, s\in\mathcal{S}, a\in\mathcal{A}}\Big|\boldsymbol{\phi}(s, a)^\top \mathbf{w}_{1,j} + \beta\sum_{i=1}^{d}\phi_i(s, a)\sqrt{\mathbf{1}_i^\top \boldsymbol{\Gamma}_{1,j} \mathbf{1}_i}$$

$$- \boldsymbol{\phi}(s, a)^\top \mathbf{w}_{2,j} - \beta\sum_{i=1}^{d}\phi_i(s, a)\sqrt{\mathbf{1}_i^\top \boldsymbol{\Gamma}_{2,j} \mathbf{1}_i}\Big|$$

$$\leq \beta\sup_{1\leq j \leq \ell, s\in\mathcal{S}, a\in\mathcal{A}}\Big|\sum_{i=1}^{d}\phi_i(s, a)\Big(\sqrt{\mathbf{1}_i^\top \boldsymbol{\Gamma}_{1,j} \mathbf{1}_i} - \sqrt{\mathbf{1}_i^\top \boldsymbol{\Gamma}_{2,j} \mathbf{1}_i}\Big)\Big|$$

$$+ \sup_{1\leq j \leq \ell, s\in\mathcal{S}, a\in\mathcal{A}}\big|\boldsymbol{\phi}(s, a)^\top(\mathbf{w}_{1,j} - \mathbf{w}_{2,j})\big|$$

$$\leq \beta\sup_{1\leq j \leq \ell, s\in\mathcal{S}, a\in\mathcal{A}}\Big|\sum_{i=1}^{d}\sqrt{\phi_i(s, a)\mathbf{1}_i^\top(\boldsymbol{\Gamma}_{1,j} - \boldsymbol{\Gamma}_{2,j})\phi_i(s, a)\mathbf{1}_i}\Big|$$

$$+ \sup_{1 \leq j \leq \ell, s \in \mathcal{S}, a \in \mathcal{A}} \left| \phi(s,a)^\top (\mathbf{w}_{1,j} - \mathbf{w}_{2,j}) \right|$$

$$\leq \beta \sup_{1 \leq j \leq \ell} \sqrt{\left\| \mathbf{\Gamma}_{1,j} - \mathbf{\Gamma}_{2,j} \right\|_F} + \sup_{1 \leq j \leq \ell} \left\| \mathbf{w}_{1,j} - \mathbf{w}_{2,j} \right\|_2 \qquad \text{(E.1)}$$

where the third inequality holds because $\left| \sqrt{x-y} \right| \geq \left| \sqrt{x} - \sqrt{y} \right|$, the fourth inequality holds because Cauchy-Schwarz inequality, $\|\phi(s,a)\|_2 \leq 1$ and $\sum_{i=1}^d \phi_i = 1$. Moreover, $\|\cdot\|_F$ is the Frobenius norm.

Now, we denote $\mathcal{C}_{\mathbf{w}}$ as a $\epsilon/2$-cover of the set $\left\{ \mathbf{w} \in \mathbb{R}^d | \|\mathbf{w}\|_2 \leq L \right\}$ and $\mathcal{C}_{\mathbf{\Gamma}}$ as a $\epsilon^2/4\beta^2$-cover of the set $\left\{ \mathbf{\Gamma} \in \mathbb{R}^{d \times d} \mid \|\mathbf{\Gamma}\|_F \leq \lambda^{-1}\sqrt{d} \right\}$ with respect to the Frobenius norm. Then according to Lemma H.6, we have

$$|\mathcal{C}_{\mathbf{w}}| \leq (1 + 4L/\epsilon)^d, |\mathcal{C}_{\mathbf{\Gamma}}| \leq \left( 1 + 8\sqrt{d}\beta^2/\lambda\epsilon^2 \right)^{d^2}.$$

Then for any function $V_1 \in \mathcal{V}_h$ with parameters $\mathbf{w}_{1,j}, \mathbf{\Gamma}_{1,j}, 1 \leq j \leq \ell$, we can find parameters $\mathbf{w}_{2,j} \in \mathcal{C}_{\mathbf{w}}, \mathbf{\Gamma}_{2,j} \in \mathcal{C}_{\mathbf{\Gamma}}, 1 \leq j \leq \ell$, such that $\|\mathbf{w}_{2,j} - \mathbf{w}_{1,j}\|_2 \leq \epsilon/2, \|\mathbf{\Gamma}_{2,j} - \mathbf{\Gamma}_{1,j}\|_F \leq \epsilon^2/4\beta^2$. Thus we have

$$\text{dist}(V_1, V_2) \leq \beta \sup_{1 \leq j \leq \ell} \sqrt{\left\| \mathbf{\Gamma}_{1,j} - \mathbf{\Gamma}_{2,j} \right\|_F} + \sup_{1 \leq j \leq \ell} \left\| \mathbf{w}_{1,j} - \mathbf{w}_{2,j} \right\|_2 \leq \epsilon,$$

where the inequality holds from (E.1). Therefore, the $\epsilon$-covering number of optimistic function class $\mathcal{V}_h$ is bounded by $\mathcal{N}_\epsilon \leq |\mathcal{C}_{\mathbf{w}}|^\ell \cdot |\mathcal{C}_{\mathbf{\Gamma}}|^\ell$, thus we have

$$\log \mathcal{N}_\epsilon \leq d\ell \log(1 + 4L/\epsilon) + d^2\ell \log \left( 1 + 8\sqrt{d}\beta^2/\lambda\epsilon^2 \right),$$

which completes the proof. $\qquad \square$

Now we are ready to prove Lemma D.2.

*Proof of Lemma D.2.* For any stage $h \in [H]$ and the optimistic value function $\hat{V}^\rho_{k,h+1}$, according to Lemma F.3, there exists a vector $\mathbf{w}_h^k$ such that $\mathbb{P}_h^0 \hat{V}^\rho_{k,h+1}(s,a)$ can be represented by $\phi(s,a)^\top \mathbf{w}_h^k$ and $\|\mathbf{w}_h^k\|_2 \leq H\sqrt{d}$. Therefore, the parameter estimation error can be decomposed as

$$\left\| \hat{\mathbf{w}}_{h,1}^k - \mathbf{w}_h^k \right\|_{\mathbf{\Lambda}_{k,h}}$$

$$\leq \left\| \mathbf{\Lambda}_{k,h}^{-1} \sum_{\tau=1}^{k-1} \phi(s_h^\tau, a_h^\tau) \hat{V}^\rho_{k,h+1}(s_{h+1}^\tau) - \mathbf{\Lambda}_{k,h}^{-1} \left( \lambda \mathbf{I} + \sum_{\tau=1}^{k-1} \phi(s_h^\tau, a_h^\tau) \phi(s_h^\tau, a_h^\tau)^\top \right) \mathbf{w}_h^k \right\|_{\mathbf{\Lambda}_{k,h}}$$

$$\leq \left\| \mathbf{\Lambda}_{k,h}^{-1} \sum_{\tau=1}^{k-1} \phi(s_h^\tau, a_h^\tau) \left( \hat{V}^\rho_{k,h+1}(s_{h+1}^\tau) - \mathbb{P}_h^0 \hat{V}^\rho_{k,h+1}(s_h^\tau, a_h^\tau) \right) - \lambda \mathbf{\Lambda}_{k,h}^{-1} \mathbf{w}_h^k \right\|_{\mathbf{\Lambda}_{k,h}}$$

$$\leq \underbrace{\left\| \lambda \mathbf{\Lambda}_{k,h}^{-1} \mathbf{w}_h^k \right\|_{\mathbf{\Lambda}_{k,h}}}_{I_1} + \underbrace{\left\| \mathbf{\Lambda}_{k,h}^{-1} \sum_{\tau=1}^{k-1} \phi(s_h^\tau, a_h^\tau) \left( \hat{V}^\rho_{k,h+1}(s_{h+1}^\tau) - \mathbb{P}_h^0 \hat{V}^\rho_{k,h+1}(s_h^\tau, a_h^\tau) \right) \right\|_{\mathbf{\Lambda}_{k,h}}}_{I_2}.$$

**Bound term $I_1$:**

$$I_1 = \left\| \lambda \mathbf{\Lambda}_{k,h}^{-1} \mathbf{w}_h^k \right\|_{\mathbf{\Lambda}_{k,h}} = \lambda \left\| \mathbf{w}_h^k \right\|_{\mathbf{\Lambda}_{k,h}^{-1}} \leq \sqrt{\lambda} \left\| \mathbf{w}_h^k \right\|_2 \leq H\sqrt{d\lambda},$$

where we have $\mathbf{\Lambda}_{k,h} \succcurlyeq \lambda \mathbf{I}$ and $\|\mathbf{w}_h^k\|_2 \leq H\sqrt{d}$.

**Bound term $I_2$:** we apply Lemma H.9 with the optimistic value function class $\mathcal{V}_h$ and $\epsilon = H\sqrt{\lambda}/K$, then for any fixed $h \in [H]$, with probability at least $1 - \delta/3H$, for all episode $k \in [K]$, we have

$$I_2 = \left\| \sum_{\tau=1}^{k-1} \phi(s_h^\tau, a_h^\tau) \left( \hat{V}^\rho_{k,h+1}(s_{h+1}^\tau) - \mathbb{P}_h^0 \hat{V}^\rho_{k,h+1}(s_h^\tau, a_h^\tau) \right) \right\|_{\mathbf{\Lambda}_{k,h}^{-1}}$$

$$\leq \sqrt{4H^2 \left[ \frac{d}{2} \log \left( \frac{k+\lambda}{\lambda} \right) + \log \frac{\mathcal{N}_\varepsilon}{\delta} \right] + \frac{8k^2\varepsilon^2}{\lambda}}$$

$$\leq \widetilde{\mathcal{O}}\big(\sqrt{d^3 H^3}\big),$$

where the first inequality holds because of Lemma H.9, the second inequality holds from Lemma E.1. Thus we have

$$\big\|\hat{\boldsymbol{w}}_{h,1}^k - \boldsymbol{w}_h^k\big\|_{\boldsymbol{\Lambda}_{k,h}} \leq I_1 + I_2 = \widetilde{\mathcal{O}}\big(H\sqrt{d\lambda} + \sqrt{d^3 H^3}\big) = \bar{\beta}.$$

Therefore, the estimation error can be bounded by

$$
\begin{aligned}
\big|\boldsymbol{\phi}(s,a)^\top \hat{\boldsymbol{w}}_{h,1}^k - \big[\mathbb{P}_h^0 \hat{V}_{k,h+1}^\rho\big](s,a)\big| &= \big|\boldsymbol{\phi}(s,a)^\top \hat{\boldsymbol{w}}_{h,1}^k - \boldsymbol{\phi}(s,a)^\top \boldsymbol{w}_h^k\big| \\
&\leq \big\|\hat{\boldsymbol{w}}_{h,1}^k - \boldsymbol{w}_h^k\big\|_{\boldsymbol{\Lambda}_{k,h}} \cdot \|\boldsymbol{\phi}(s,a)\|_{\boldsymbol{\Lambda}_{k,h}^{-1}} \\
&\leq \bar{\beta}\sqrt{\boldsymbol{\phi}(s,a)^\top \boldsymbol{\Lambda}_{k,h}^{-1} \boldsymbol{\phi}(s,a)},
\end{aligned}
$$

where the first inequality holds from Cauchy-Schwarz inequality. Similarly, for the pessimistic function class $\check{\mathcal{V}}_h$ (or squared value function class $\mathcal{V}_h^2$), we have the similar result as follows

$$\big|\boldsymbol{\phi}(s,a)^\top \check{\boldsymbol{w}}_{h,1}^k - \big[\mathbb{P}_h^0 \check{V}_{k,h+1}^\rho\big](s,a)\big| \leq \bar{\beta}\sqrt{\boldsymbol{\phi}(s,a)^\top \boldsymbol{\Lambda}_{k,h}^{-1} \boldsymbol{\phi}(s,a)},$$

$$\big|\boldsymbol{\phi}(s,a)^\top \widetilde{\boldsymbol{w}}_{h,2}^k - \big[\mathbb{P}_h^0 \big(\hat{V}_{k,h+1}^\rho\big)^2\big](s,a)\big| \leq \widetilde{\beta}\sqrt{\boldsymbol{\phi}(s,a)^\top \boldsymbol{\Lambda}_{k,h}^{-1} \boldsymbol{\phi}(s,a)},$$

where $\bar{\beta} = \widetilde{\mathcal{O}}\big(H\sqrt{d\lambda} + \sqrt{d^3 H^3}\big)$ and $\widetilde{\beta} = \widetilde{\mathcal{O}}\big(H^2\sqrt{d\lambda} + \sqrt{d^3 H^6}\big)$. By taking union bound over $h \in [H]$ and three function classes, we have that the event $\bar{\mathcal{E}}$ holds with probability at least $1 - \delta$. This completes the proof. $\qquad\square$

## E.2 Proof of Lemma D.3

*Proof of Lemma D.3.* First, recall from (3.6), we have

$$\big[\mathbb{V}_h \hat{V}_{k,h+1}^\rho\big](s,a) \approx \big[\bar{\mathbb{V}}_h \hat{V}_{k,h+1}^\rho\big](s,a) = \big[\boldsymbol{\phi}(s,a)^\top \widetilde{\boldsymbol{w}}_{h,2}^k\big]_{[0,H^2]} - \big[\boldsymbol{\phi}(s,a)^\top \hat{\boldsymbol{w}}_{h,1}^k\big]_{[0,H]}^2,$$

where $\hat{\boldsymbol{w}}_{h,1}^k$ and $\widetilde{\boldsymbol{w}}_{h,2}^k$ is the solution of the following ridge regression problems

$$\widetilde{\boldsymbol{w}}_{h,2}^k = \operatorname*{argmin}_{\boldsymbol{w}\in\mathbb{R}^d} \sum_{\tau=1}^{k-1} \Big(\boldsymbol{w}^\top \boldsymbol{\phi}\big(s_h^\tau, a_h^\tau\big) - \big(\hat{V}_{k,h+1}^\rho\big(s_{h+1}^\tau\big)\big)^2\Big)^2 + \lambda\|\boldsymbol{w}\|_2^2,$$

$$\hat{\boldsymbol{w}}_{h,1}^k = \operatorname*{argmin}_{\boldsymbol{w}\in\mathbb{R}^d} \sum_{\tau=1}^{k-1} \Big(\boldsymbol{w}^\top \boldsymbol{\phi}\big(s_h^\tau, a_h^\tau\big) - \hat{V}_{k,h+1}^\rho\big(s_{h+1}^\tau\big)\Big)^2 + \lambda\|\boldsymbol{w}\|_2^2.$$

Then we have

$$
\begin{aligned}
&\big|\big[\bar{\mathbb{V}}_h \hat{V}_{k,h+1}^\rho\big]\big(s_h^k, a_h^k\big) - \big[\mathbb{V}_h \hat{V}_{k,h+1}^\rho\big]\big(s_h^k, a_h^k\big)\big| \\
&\leq \Big|\big[\boldsymbol{\phi}\big(s_h^k, a_h^k\big)^\top \widetilde{\boldsymbol{w}}_{h,2}^k\big]_{[0,H^2]} - \big[\boldsymbol{\phi}\big(s_h^k, a_h^k\big)^\top \hat{\boldsymbol{w}}_{h,1}^k\big]_{[0,H]}^2 - \big[\mathbb{P}_h^0 \big(\hat{V}_{k,h+1}^\rho\big)^2\big]\big(s_h^k, a_h^k\big) + \big(\big[\mathbb{P}_h^0 \hat{V}_{k,h+1}^\rho\big]\big(s_h^k, a_h^k\big)\big)^2\Big| \\
&\leq \Big|\big[\boldsymbol{\phi}\big(s_h^k, a_h^k\big)^\top \widetilde{\boldsymbol{w}}_{h,2}^k\big]_{[0,H^2]} - \big[\mathbb{P}_h^0 \big(\hat{V}_{k,h+1}^\rho\big)^2\big]\big(s_h^k, a_h^k\big)\Big| + \Big|\big[\boldsymbol{\phi}\big(s_h^k, a_h^k\big)^\top \hat{\boldsymbol{w}}_{h,1}^k\big]_{[0,H]}^2 - \big(\big[\mathbb{P}_h^0 \hat{V}_{k,h+1}^\rho\big]\big(s_h^k, a_h^k\big)\big)^2\Big| \\
&\leq \Big|\big[\boldsymbol{\phi}\big(s_h^k, a_h^k\big)^\top \widetilde{\boldsymbol{w}}_{h,2}^k\big]_{[0,H^2]} - \big[\mathbb{P}_h^0 \big(\hat{V}_{k,h+1}^\rho\big)^2\big]\big(s_h^k, a_h^k\big)\Big| + 2H\Big|\big[\boldsymbol{\phi}\big(s_h^k, a_h^k\big)^\top \hat{\boldsymbol{w}}_{h,1}^k\big]_{[0,H]} - \big[\mathbb{P}_h^0 \hat{V}_{k,h+1}^\rho\big]\big(s_h^k, a_h^k\big)\Big| \\
&\leq \min\Big\{\widetilde{\beta}\big\|\boldsymbol{\phi}\big(s_h^k, a_h^k\big)\big\|_{\boldsymbol{\Lambda}_{k,h}^{-1}}, H^2\Big\} + \min\Big\{2H\bar{\beta}\big\|\boldsymbol{\phi}\big(s_h^k, a_h^k\big)\big\|_{\boldsymbol{\Lambda}_{k,h}^{-1}}, H^2\Big\} \\
&= E_{k,h},
\end{aligned}
$$

where the last inequality holds from Lemma D.2. For the second result, we have

$$
\begin{aligned}
&\big|\big[\mathbb{V}_h \hat{V}_{k,h+1}^\rho\big]\big(s_h^k, a_h^k\big) - \big[\mathbb{V}_h V_{h+1}^{*,\rho}\big]\big(s_h^k, a_h^k\big)\big| \\
&= \Big|\big[\mathbb{P}_h^0 \big(\hat{V}_{k,h+1}^\rho\big)^2\big]\big(s_h^k, a_h^k\big) - \big(\big[\mathbb{P}_h^0 \hat{V}_{k,h+1}^\rho\big]\big(s_h^k, a_h^k\big)\big)^2 - \big[\mathbb{P}_h^0 \big(V_{h+1}^{*,\rho}\big)^2\big]\big(s_h^k, a_h^k\big) + \big(\big[\mathbb{P}_h^0 V_{h+1}^{*,\rho}\big]\big(s_h^k, a_h^k\big)\big)^2\Big| \\
&\leq \Big|\big[\mathbb{P}_h^0 \big(\hat{V}_{k,h+1}^\rho\big)^2\big]\big(s_h^k, a_h^k\big) - \big[\mathbb{P}_h^0 \big(V_{h+1}^{*,\rho}\big)^2\big]\big(s_h^k, a_h^k\big)\Big| + \Big|\big(\big[\mathbb{P}_h^0 \hat{V}_{k,h+1}^\rho\big]\big(s_h^k, a_h^k\big)\big)^2 - \big(\big[\mathbb{P}_h^0 V_{h+1}^{*,\rho}\big]\big(s_h^k, a_h^k\big)\big)^2\Big| \\
&\leq 4H\Big|\big[\mathbb{P}_h^0 \hat{V}_{k,h+1}^\rho\big]\big(s_h^k, a_h^k\big) - \big[\mathbb{P}_h^0 V_{h+1}^{*,\rho}\big]\big(s_h^k, a_h^k\big)\Big|
\end{aligned}
$$

$$\leq 4H\left(\left[\mathbb{P}_h^0 \hat{V}_{k,h+1}^\rho\right]\left(s_h^k, a_h^k\right) - \left[\mathbb{P}_h^0 V_{h+1}^{*,\rho}\right]\left(s_h^k, a_h^k\right)\right)$$

$$\leq 4H\left(\left[\mathbb{P}_h^0 \hat{V}_{k,h+1}^\rho\right]\left(s_h^k, a_h^k\right) - \left[\mathbb{P}_h^0 \check{V}_{k,h+1}^\rho\right]\left(s_h^k, a_h^k\right)\right)$$

$$\leq \min\left\{4H\left(\boldsymbol{\phi}\left(s_h^k, a_h^k\right)^\top \hat{\boldsymbol{w}}_{h,1}^k - \boldsymbol{\phi}\left(s_h^k, a_h^k\right)^\top \check{\boldsymbol{w}}_{h,1}^k + 2\bar{\beta}\left\|\boldsymbol{\phi}\left(s_h^k, a_h^k\right)\right\|_{\boldsymbol{\Lambda}_{k,h}^{-1}}\right), H^2\right\}$$

$$= D_{k,h}.$$

707 where the second inequality holds because $0 \leq V_{h+1}^{*,\rho}, \hat{V}_{k,h+1}^\rho \leq H$, the third and fourth inequality
708 holds because of Lemma D.5, the fifth inequality holds due to Lemma D.2 and the last inequality
709 holds because the trivial result $0 \leq \left[\bar{\mathbb{V}}_h \hat{V}_{k,h+1}^\rho\right]\left(s_h^k, a_h^k\right), \left[\mathbb{V}_h V_{h+1}^{*,\rho}\right]\left(s_h^k, a_h^k\right) \leq H^2$. Thus we have

$$\left|\left[\bar{\mathbb{V}}_h \hat{V}_{k,h+1}^\rho\right]\left(s_h^k, a_h^k\right) - \left[\mathbb{V}_h V_{h+1}^{*,\rho}\right]\left(s_h^k, a_h^k\right)\right| \leq E_{k,h} + D_{k,h}.$$

710 Then we also have

$$\bar{\sigma}_{k,h}^2 \geq \left[\bar{\mathbb{V}}_h \hat{V}_{k,h+1}^\rho\right]\left(s_h^k, a_h^k\right) + E_{k,h} + D_{k,h} \geq \left[\mathbb{V}_h V_{h+1}^{*,\rho}\right]\left(s_h^k, a_h^k\right),$$

711 where we use the definition of $\bar{\sigma}_{k,h}$ in (3.6). This completes the proof. $\qquad\square$

### E.3 Proof of Lemma D.4

713 *Proof of Lemma D.4.* For all $(s,a) \in \mathcal{S}/\{s_f\} \times \mathcal{A}$, for stage $h \leq h' \leq H$ (we use $h$ to replace $h'$ in
714 this part for simplicity), we have

$$Q_h^{\pi,\rho}(s,a) = r_h(s,a) + \boldsymbol{\phi}(s,a)^\top \boldsymbol{\nu}_h^{\pi,\rho} = r_h(s,a) + \inf_{P_h(\cdot|s,a)\in\mathcal{U}_h^\rho(s,a;\boldsymbol{\mu}_h^0)} \left[\mathbb{P}_h V_{h+1}^{\pi,\rho}\right](s,a).$$

715 We first decompose the gap $\hat{\boldsymbol{\nu}}_h^{\rho,k} - \boldsymbol{\nu}_h^{\pi,\rho}$ into two terms

$$\hat{\boldsymbol{\nu}}_h^{\rho,k} - \boldsymbol{\nu}_h^{\pi,\rho} = \underbrace{\hat{\boldsymbol{\nu}}_h^{\rho,k} - \tilde{\boldsymbol{\nu}}_h^{\rho,k}}_{\text{I}} + \underbrace{\tilde{\boldsymbol{\nu}}_h^{\rho,k} - \boldsymbol{\nu}_h^{\pi,\rho}}_{\text{II}}, \tag{E.2}$$

716 where $\tilde{\boldsymbol{\nu}}_h^{\rho,k} = \left[\tilde{\nu}_{h,i}^{\rho,k}\right]_{i\in[d]}$, and $\tilde{\nu}_{h,i}^{\rho,i} = \max_{\alpha\in[0,H]}\left\{\mathbb{E}^{\mu_{h,i}^0}\left[\hat{V}_{k,h+1}^\rho(s)\right]_\alpha - \rho\alpha\right\}$. Then we will bound
717 these two terms separately.

**Bound term I in** (E.2): we have

$$\hat{\boldsymbol{\nu}}_h^{\rho,k} - \tilde{\boldsymbol{\nu}}_h^{\rho,k} \leq \left[\max_{\alpha\in[0,H]}\left\{\hat{z}_{h,i}^k(\alpha) - \mathbb{E}^{\mu_{h,i}^0}\left[\hat{V}_{k,h+1}^\rho(s)\right]_\alpha\right\}\right]_{i\in[d]}.$$

719 Denote $\alpha_i^k = \arg\max_{\alpha\in[0,H]}\left\{\hat{z}_{h,i}^k(\alpha) - \mathbb{E}^{\mu_{h,i}^0}\left[\hat{V}_{k,h+1}^\rho(s)\right]_\alpha\right\}$, $i = 1, \cdots, d$. Then we have

$$\hat{\boldsymbol{\nu}}_h^{\rho,k} - \tilde{\boldsymbol{\nu}}_h^{\rho,k} \leq \left[\left(\boldsymbol{\Sigma}_{k,h}^{-1}\sum_{\tau=1}^{k-1}\bar{\sigma}_{\tau,h}^{-2}\boldsymbol{\phi}\left(s_h^\tau, a_h^\tau\right)\left[\hat{V}_{k,h+1}^\rho\left(s_{h+1}^\tau\right)\right]_{\alpha_i^k}\right)_i - \left(\mathbb{E}^{\boldsymbol{\mu}_h^0}\left[\hat{V}_{k,h+1}^\rho(s)\right]_{\alpha_i^k}\right)_i\right]_{i\in[d]}$$

$$= \left[\left(-\lambda\boldsymbol{\Sigma}_{k,h}^{-1}\mathbb{E}^{\boldsymbol{\mu}_h^0}\left[\hat{V}_{k,h+1}^\rho(s)\right]_{\alpha_i^k}\right)_i + \left(\boldsymbol{\Sigma}_{k,h}^{-1}\sum_{\tau=1}^{k-1}\bar{\sigma}_{\tau,h}^{-2}\boldsymbol{\phi}_h^\tau\left[\left[\hat{V}_{k,h+1}^\rho(s_{h+1}^\tau)\right]_{\alpha_i^k}\right.\right.\right.$$

$$\left.\left.\left. - \left[\mathbb{P}_h^0\left[\hat{V}_{k,h+1}^\rho\right]_{\alpha_i^k}\right]\left(s_h^\tau, a_h^\tau\right)\right]\right)_i\right]_{i\in[d]}. \tag{E.3}$$

720 For the first term on the RHS of (E.3),

$$\left|\left\langle \boldsymbol{\phi}(s,a), \left[\left(-\lambda\boldsymbol{\Sigma}_{k,h}^{-1}\mathbb{E}^{\boldsymbol{\mu}_h^0}\left[\hat{V}_{k,h+1}^\rho(s)\right]_{\alpha_i^k}\right)_i\right]_{i\in[d]}\right\rangle\right|$$

$$= \left|\sum_{i=1}^d \phi_i(s,a)\mathbf{1}_i^\top(-\lambda)\boldsymbol{\Sigma}_{k,h}^{-1}\mathbb{E}^{\boldsymbol{\mu}_h^0}\left[\hat{V}_{k,h+1}^\rho(s)\right]_{\alpha_i^k}\right|$$

$$\leq \lambda\sum_{i=1}^d \sqrt{\phi_i(s,a)\mathbf{1}_i^\top\boldsymbol{\Sigma}_{k,h}^{-1}\phi_i(s,a)\mathbf{1}_i} \cdot \left\|\mathbb{E}^{\boldsymbol{\mu}_h^0}\left[\hat{V}_{k,h+1}^\rho(s)\right]_{\alpha_i^k}\right\|_{\boldsymbol{\Sigma}_{k,h}^{-1}}$$

$$\leq \sqrt{\lambda d} H \sum_{i=1}^{d} \sqrt{\phi_i(s,a)\mathbf{1}_i^\top \mathbf{\Sigma}_{k,h}^{-1}\phi_i(s,a)\mathbf{1}_i}, \tag{E.4}$$

where $\mathbf{1}_i$ is the vector with the $i$-th entry being 1 and else being 0. The first inequality holds due to the Cauchy-Schwarz inequality.

For the second term on the RHS of (E.3), given the event $\mathcal{E}_h$ defined in Definition D.1, we have

$$\left| \left\langle \phi(s,a), \left[ \left( \mathbf{\Sigma}_{k,h}^{-1} \sum_{\tau=1}^{k-1} \bar\sigma_{\tau,h}^{-2} \phi_h^\tau \left[ \left[ \hat{V}_{k,h+1}^\rho(s_{h+1}^\tau) \right]_{\alpha_i^k} - \left[ \mathbb{P}_h^0 \left[ \hat{V}_{k,h+1}^\rho \right]_{\alpha_i^k} \right](s_h^\tau, a_h^\tau) \right] \right)_i \right]_{i\in[d]} \right\rangle \right|$$

$$= \left| \sum_{i=1}^{d} \phi_i(s,a)\mathbf{1}_i^\top \mathbf{\Sigma}_{k,h}^{-1} \sum_{\tau=1}^{k-1} \bar\sigma_{\tau,h}^{-2} \phi_h^\tau \left[ \left[ \hat{V}_{k,h+1}^\rho(s_{h+1}^\tau) \right]_{\alpha_i^k} - \left[ \mathbb{P}_h^0 \left[ \hat{V}_{k,h+1}^\rho \right]_{\alpha_i^k} \right](s_h^\tau, a_h^\tau) \right] \right|$$

$$\leq \sum_{i=1}^{d} \sqrt{\phi_i(s,a)\mathbf{1}_i^\top \mathbf{\Sigma}_{k,h}^{-1}\phi_i(s,a)\mathbf{1}_i} \cdot \left\| \sum_{\tau=1}^{k-1} \bar\sigma_{\tau,h}^{-2} \phi_h^\tau \left[ \left[ \hat{V}_{k,h+1}^\rho(s_{h+1}^\tau) \right]_{\alpha_i^k} - \left[ \mathbb{P}_h^0 \left[ \hat{V}_{k,h+1}^\rho \right]_{\alpha_i^k} \right](s_h^\tau, a_h^\tau) \right] \right\|_{\mathbf{\Sigma}_{k,h}^{-1}}$$

$$\leq \gamma \sum_{i=1}^{d} \sqrt{\phi_i(s,a)\mathbf{1}_i^\top \mathbf{\Sigma}_{k,h}^{-1}\phi_i(s,a)\mathbf{1}_i}, \tag{E.5}$$

where the first inequality follows from Cauchy-Schwarz inequality and the second inequality holds from the event $\mathcal{E}_h$. Combining (E.3), (E.4) and (E.5), we have

$$\langle \phi(s,a), \hat{\boldsymbol{\nu}}_h^{\rho,k} - \tilde{\boldsymbol{\nu}}_h^{\rho,k} \rangle \leq \left( \sqrt{\lambda d} H + \gamma \right) \sum_{i=1}^{d} \phi_i(s,a)\sqrt{\mathbf{1}_i^\top \mathbf{\Sigma}_{k,h}^{-1}\mathbf{1}_i},$$

On the other hand, we can similarly do analysis for $\langle \phi(s,a), \tilde{\boldsymbol{\nu}}_h^{\rho,k} - \hat{\boldsymbol{\nu}}_h^{\rho,k} \rangle$. Then we have

$$\left| \langle \phi(s,a), \hat{\boldsymbol{\nu}}_h^{\rho,k} - \tilde{\boldsymbol{\nu}}_h^{\rho,k} \rangle \right| \leq \beta \sum_{i=1}^{d} \phi_i(s,a)\sqrt{\mathbf{1}_i^\top \mathbf{\Sigma}_{k,h}^{-1}\mathbf{1}_i}, \tag{E.6}$$

where $\beta = \left( \sqrt{\lambda d} H + \gamma \right) = \widetilde{\mathcal{O}}\left( \sqrt{\lambda d} H + \sqrt{d} \right)$.

**Bound term II in** (E.2)**:** we have

$$\langle \phi(s,a), \tilde{\boldsymbol{\nu}}_h^{\rho,k} - \boldsymbol{\nu}_h^{\pi,\rho} \rangle = \inf_{P_h(\cdot|s,a)\in\mathcal{U}_h^\rho(s,a;\boldsymbol{\mu}_h^0)} \left[ \mathbb{P}_h \hat{V}_{k,h+1}^\rho \right](s,a) - \inf_{P_h(\cdot|s,a)\in\mathcal{U}_h^\rho(s,a;\boldsymbol{\mu}_h^0)} \left[ \mathbb{P}_h V_{h+1}^{\pi,\rho} \right](s,a).$$

Finally we have

$$\left( r_h(s,a) + \phi(s,a)^\top \hat{\boldsymbol{\nu}}_h^{\rho,k} \right) - Q_h^{\pi,\rho}(s,a)$$
$$= \langle \phi(s,a), \hat{\boldsymbol{\nu}}_h^{\rho,k} - \tilde{\boldsymbol{\nu}}_h^{k,\rho} + \tilde{\boldsymbol{\nu}}_h^{k,\rho} - \boldsymbol{\nu}_h^{\pi,\rho} \rangle$$
$$= \inf_{P_h(\cdot|s,a)\in\mathcal{U}_h^\rho(s,a;\boldsymbol{\mu}_h^0)} \left[ \mathbb{P}_h \hat{V}_{k,h+1}^\rho \right](s,a) - \inf_{P_h(\cdot|s,a)\in\mathcal{U}_h^\rho(s,a;\boldsymbol{\mu}_h^0)} \left[ \mathbb{P}_h V_{h+1}^{\pi,\rho} \right](s,a) + \Delta_h^k(s,a),$$

where $|\Delta_h^k(s,a)| \leq \beta \sum_{i=1}^{d} \phi_i(s,a)\sqrt{\mathbf{1}_i^\top \mathbf{\Sigma}_{k,h}^{-1}\mathbf{1}_i}$. This completes the proof. $\square$

### E.4   Proof of Lemma D.5

*Proof of Lemma D.5.* We prove this lemma by induction. For last stage $H + 1$, it is trivial because for all $(s,a) \in \mathcal{S} \times \mathcal{A}$, we have $\hat{Q}_{k,H+1}^\rho(s,a) = Q_{H+1}^{*,\rho}(s,a) = \check{Q}_{k,H+1}^\rho(s,a) = 0$.

Assume that the lemma holds at stage $h'+1$, now consider the situation at stage $h'$ (we use $h$ to replace $h'$ in this part for simplicity). For all episode $k \in [K]$, we have

$$r_h(s,a) + \phi(s,a)^\top \hat{\boldsymbol{\nu}}_h^{\rho,k} + \hat{\Gamma}_{k,h}(s,a) - Q_h^{*,\rho}(s,a)$$
$$\geq \inf_{P_h(\cdot|s,a)\in\mathcal{U}_h^\rho(s,a;\boldsymbol{\mu}_h^0)} \left[ \mathbb{P}_h \hat{V}_{k,h+1}^\rho \right](s,a) - \inf_{P_h(\cdot|s,a)\in\mathcal{U}_h^\rho(s,a;\boldsymbol{\mu}_h^0)} \left[ \mathbb{P}_h V_{h+1}^{*,\rho} \right](s,a) + \Delta_h^k(s,a) + \hat{\Gamma}_{k,h}(s,a)$$
$$\geq \inf_{P_h(\cdot|s,a)\in\mathcal{U}_h^\rho(s,a;\boldsymbol{\mu}_h^0)} \left[ \mathbb{P}_h \left( \hat{V}_{k,h+1}^\rho - V_{h+1}^{*,\rho} \right) \right](s,a)$$

$$\geq 0,$$

where the first inequality holds from Lemma D.4, the second inequality holds because $|\Delta_h^k(s,a)| \leq \hat{\Gamma}_{k,h}(s,a)$, the third inequality holds from induction assumption. Thus we have

$$Q_h^{*,\rho}(s,a) \leq \min\left\{\min_{i\in[k]} r_h(s,a) + \phi(s,a)^\top \hat{\boldsymbol{\nu}}_h^{\rho,i} + \hat{\Gamma}_{i,h}(s,a), H - h + 1\right\} \leq \hat{Q}_{k,h}^\rho(s,a).$$

Thus for value function $V$, we have

$$\hat{V}_{k,h}^\rho(s) = \max_a \hat{Q}_{k,h}^\rho(s,a) \geq \max_a Q_h^{*,\rho}(s,a) = V_h^{*,\rho}(s).$$

For the pessimistic value function $\check{Q}_{k,h}^\rho(s,a)$, we can do the similar analysis. Finally, by induction, we finish the proof. $\qquad\square$

### E.5 Proof of Lemma D.6

*Proof of Lemma D.6.* We use backward induction to prove this lemma. For the base case, the stage $H$, it is trivial to obtain (D.1) because $\hat{V}_{k,H+1}^\rho = 0$. Assume (D.1) hold for the stage $h+1$, then we consider the stage $h$.

For all episode $k \in [K]$, we first do the following decomposition

$$\left\|\sum_{\tau=1}^{k-1} \bar{\sigma}_{\tau,h}^{-2} \phi_h^\tau \left[\left[\hat{V}_{k,h+1}^\rho(s_{h+1}^\tau)\right]_{\alpha'} - \left[\mathbb{P}_h^0\left[\hat{V}_{k,h+1}^\rho\right]_{\alpha'}\right](s_h^\tau, a_h^\tau)\right]\right\|_{\mathbf{\Sigma}_{k,h}^{-1}}$$

$$\leq \underbrace{\left\|\sum_{\tau=1}^{k-1} \bar{\sigma}_{\tau,h}^{-2} \phi_h^\tau \left[\left[V_{h+1}^{*,\rho}(s_{h+1}^\tau)\right]_{\alpha'} - \left[\mathbb{P}_h^0\left[V_{h+1}^{*,\rho}\right]_{\alpha'}\right](s_h^\tau, a_h^\tau)\right]\right\|_{\mathbf{\Sigma}_{k,h}^{-1}}}_{J_1}$$

$$+ \underbrace{\left\|\sum_{\tau=1}^{k-1} \bar{\sigma}_{\tau,h}^{-2} \phi_h^\tau \left[\Delta_{\alpha'}\hat{V}_{k,h+1}^\rho(s_{h+1}^\tau) - \left[\mathbb{P}_h^0\left(\Delta_{\alpha'}\hat{V}_{k,h+1}^\rho\right)\right](s_h^\tau, a_h^\tau)\right]\right\|_{\mathbf{\Sigma}_{k,h}^{-1}}}_{J_2}, \qquad (\text{E.7})$$

where $\Delta_{\alpha'}\hat{V}_{k,h+1}^\rho(s_{h+1}^\tau) = \left[\hat{V}_{k,h+1}^\rho(s_{h+1}^\tau)\right]_{\alpha'} - \left[V_{h+1}^{*,\rho}(s_{h+1}^\tau)\right]_{\alpha'}$.

**Bound term $J_1$ in (E.7):** For term $J_1$, we apply Lemma H.8 with $\mathbf{x}_i = \bar{\sigma}_{i,h}^{-1}\phi(s_h^i, a_h^i)$ and $\eta_i = \bar{\sigma}_{i,h}^{-1}\left(\left[V_{h+1}^{*,\rho}(s_{h+1}^i)\right]_{\alpha'} - \left[\mathbb{P}_h^0\left[V_{h+1}^{*,\rho}\right]_{\alpha'}\right](s_h^i, a_h^i)\right)$. Note that based on Lemma D.3, we have $\bar{\sigma}_{k,h}^2 \geq \left[\mathbb{V}_h V_{h+1}^{*,\rho}\right](s_h^k, a_h^k) \geq \left[\mathbb{V}_h\left[V_{h+1}^{*,\rho}\right]_{\alpha'}\right](s_h^k, a_h^k)$. Then for $\mathbf{x}_i$ and $\eta_i$, we have

$$\|\mathbf{x}_i\|_2 = \left\|\phi(s_h^i, a_h^i)\right\|_2 / \bar{\sigma}_{i,h} \leq 1,$$

$$\mathbb{E}[\eta_i|\mathcal{F}_i] = 0, |\eta_i| \leq \left|\bar{\sigma}_{i,h}^{-1}\left(\left[V_{h+1}^{*,\rho}(s_{h+1}^i)\right]_{\alpha'} - \left[\mathbb{P}_h^0\left[V_{h+1}^{*,\rho}\right]_{\alpha'}\right](s_h^i, a_h^i)\right)\right| \leq 2H,$$

$$\mathbb{E}[\eta_i^2|\mathcal{F}_i] = \mathbb{E}\left[\bar{\sigma}_{i,h}^{-2}\left(\left[V_{h+1}^{*,\rho}(s_{h+1}^i)\right]_{\alpha'} - \left[\mathbb{P}_h^0\left[V_{h+1}^{*,\rho}\right]_{\alpha'}\right](s_h^i, a_h^i)\right)^2\right] \leq 1 = \sigma^2,$$

$$\max_{1\leq i\leq k}\left\{|\eta_i| \cdot \min\left\{1, \|\mathbf{x}_i\|_{\mathbf{\Sigma}_{i,h}^{-1}}\right\}\right\} \leq \max_{1\leq i\leq k}\left\{2H\bar{\sigma}_{i,h}^{-1}\|\mathbf{x}_i\|_{\mathbf{\Sigma}_{i,h}^{-1}}\right\} \leq \sqrt{d},$$

where we use the definition of $\bar{\sigma}_{i,h}$ in (3.7). Then for all $k \in [K]$, with probability at least $1 - \delta/2H$, we have

$$J_1 = \left\|\sum_{i=1}^{k-1}\mathbf{x}_i\eta_i\right\|_{\mathbf{\Sigma}_{k,h}^{-1}} \leq \widetilde{\mathcal{O}}\left(\sigma\sqrt{d} + \max_{1\leq i\leq k}|\eta_i|\min\left\{1, \|\mathbf{x}_i\|_{\mathbf{\Sigma}_{i,h}^{-1}}\right\}\right) = \widetilde{O}(\sqrt{d}).$$

**Bound term $J_2$ in (E.7):** To bound term $J_2$, we need to use $\epsilon$-covering for function class $\widetilde{\mathcal{V}}_{h+1} - \left[V_{h+1}^{*,\rho}\right]_{\alpha'}$ where $\widetilde{\mathcal{V}}_{h+1} = \left\{[V]_\alpha | V \in \mathcal{V}_{h+1}, \alpha \in [0, H]\right\}$ is the truncated optimistic value function class. For any two function $\widetilde{V}_1, \widetilde{V}_2 \in \widetilde{\mathcal{V}}_{h+1}$, we can write that

$$\widetilde{V}_1 = [V_1]_{\alpha_1}, \widetilde{V}_2 = [V_2]_{\alpha_2},$$

where $V_1, V_2 \in \mathcal{V}_{h+1}$, $\alpha_1, \alpha_2 \in [0, H]$. Then we have

$$\begin{aligned}
\mathrm{dist}(\widetilde{V}_1, \widetilde{V}_2) &= \sup_s \left| \widetilde{V}_1(s) - \widetilde{V}_2(s) \right| \\
&= \sup_s \left| [V_1]_{\alpha_1}(s) - [V_2]_{\alpha_2}(s) \right| \\
&\leq \sup_s \left| [V_1]_{\alpha_1}(s) - [V_1]_{\alpha_2}(s) \right| + \sup_s \left| [V_1]_{\alpha_2}(s) - [V_2]_{\alpha_2}(s) \right| \\
&\leq |\alpha_1 - \alpha_2| + \mathrm{dist}(V_1, V_2).
\end{aligned}$$

This indicates that the $\epsilon$-covering number $\widetilde{\mathcal{N}}_\epsilon$ for function class $\widetilde{\mathcal{V}}_{h+1}$ can be bounded by

$$\widetilde{\mathcal{N}}_\epsilon \leq \mathcal{N}_{1, \frac{\epsilon}{2}} \cdot \mathcal{N}_{2, \frac{\epsilon}{2}},$$

where $\mathcal{N}_{1, \frac{\epsilon}{2}}$ is the $\frac{\epsilon}{2}$-covering number for optimistic value function class $\mathcal{V}_{h+1}$ and $\mathcal{N}_{2, \frac{\epsilon}{2}}$ is the $\frac{\epsilon}{2}$-covering number for closed interval $[0, H]$. Then based on Lemma E.1 and Lemma H.7, we have

$$\log \widetilde{\mathcal{N}}_\epsilon \leq d\ell \log(1 + 8L/\epsilon) + d^2\ell \log\left(1 + 32\sqrt{d}\beta^2/\lambda\epsilon^2\right) + \log(6H/\epsilon),$$

where $\ell = dH \log(1 + K/\lambda)$ and $L = 2H\sqrt{dK/\lambda}$. Here we set $\epsilon = \sqrt{\lambda}/4H^2 d^3 K$, then the covering entropy can be bounded by

$$\log \widetilde{\mathcal{N}}_\epsilon \leq \widetilde{\mathcal{O}}(d^3 H).$$

For simplicity, we denote $\Delta_{\alpha'} \hat{V}^\rho_{k, h+1}$ here as $\Delta V$, then for $\Delta V$, there exsit a function $\widetilde{V}$ in the $\epsilon$-net satisfies that

$$\mathrm{dist}\big(\Delta V, \widetilde{V}\big) \leq \epsilon.$$

Then the difference of the variance of $\Delta V$ and $\widetilde{V}$ can be bounded by

$$\begin{aligned}
&\left[ \mathbb{V}_h \widetilde{V} \right]\left(s_h^k, a_h^k\right) - \left[ \mathbb{V}_h \Delta V \right]\left(s_h^k, a_h^k\right) \\
&= \left[ \mathbb{P}_h \widetilde{V}^2 \right]\left(s_h^k, a_h^k\right) - \left[ \mathbb{P}_h (\Delta V)^2 \right]\left(s_h^k, a_h^k\right) - \left( \left[ \mathbb{P}_h \widetilde{V} \right]\left(s_h^k, a_h^k\right) \right)^2 + \left( \left[ \mathbb{P}_h (\Delta V) \right]\left(s_h^k, a_h^k\right) \right)^2 \\
&\leq 2 \sup_s \left| \Delta V(s) - \widetilde{V}(s) \right| \cdot \sup_s \left| \Delta V(s) + \widetilde{V}(s) \right| \\
&\leq 4H \mathrm{dist}\big(\Delta V, \widetilde{V}\big) \\
&\leq \frac{1}{2d^3 H}.
\end{aligned}$$

This indicates that

$$\begin{aligned}
\left[ \mathbb{V}_h \widetilde{V} \right]\left(s_h^k, a_h^k\right) &\leq \left[ \mathbb{V}_h \Delta V \right]\left(s_h^k, a_h^k\right) + \frac{1}{2d^3 H} \\
&\leq \left[ \mathbb{P}_h \left( [\hat{V}^\rho_{k, h+1}]_{\alpha'} - [V^{*, \rho}_{h+1}]_{\alpha'} \right)^2 \right]\left(s_h^k, a_h^k\right) + \frac{1}{2d^3 H} \\
&\leq \left[ \mathbb{P}_h \left( \hat{V}^\rho_{k, h+1} - V^{*, \rho}_{h+1} \right)^2 \right]\left(s_h^k, a_h^k\right) + \frac{1}{2d^3 H} \\
&\leq 2H \left[ \mathbb{P}_h \left( \hat{V}^\rho_{k, h+1} - \check{V}^\rho_{k, h+1} \right) \right]\left(s_h^k, a_h^k\right) + \frac{1}{2d^3 H} \\
&\leq 2H \left( \mathbb{P}_h \hat{V}^\rho_{k, h+1}\left(s_h^k, a_h^k\right) - \mathbb{P}_h \check{V}^\rho_{k, h+1}\left(s_h^k, a_h^k\right) \right) + \frac{1}{2d^3 H} \\
&\leq D_{k, h} + \frac{1}{2d^3 H} \\
&\leq \bar{\sigma}^2_{k, h}/d^3 H, \quad\quad\quad\quad\quad\quad\quad\quad\quad\quad\quad\quad\quad\quad (\mathrm{E}.8)
\end{aligned}$$

where the fourth inequality holds due to Lemma D.5 with induction assumption $\mathcal{E}_{h+1}$, the sixth inequality holds due to the definition of $D_{k, h}$ and the last inequality holds because of the definition of $\sigma_{k, h}$. Then we apply we apply Lemma H.8 with $\mathbf{x}_i = \bar{\sigma}_{i, h}^{-1} \phi\big(s_h^i, a_h^i\big)$ and $\eta_i = \bar{\sigma}_{i, h}^{-1} \big( \widetilde{V}(s_{h+1}^i) - \mathbb{P}_h^0 \widetilde{V}\big(s_h^i, a_h^i\big) \big)$. For $\mathbf{x}_i$ and $\eta_i$, we have

$$\|\mathbf{x}_i\|_2 \leq \left\| \phi\big(s_h^i, a_h^i\big) \right\|_2 / \bar{\sigma}_{i, h} \leq 1,$$

$$\mathbb{E}[\eta_i|\mathcal{F}_i] = 0, |\eta_i| \leq \left|\bar{\sigma}_{i,h}^{-1}\big(\widetilde{V}(s_{h+1}^i) - \mathbb{P}_h^0\widetilde{V}(s_h^i, a_h^i)\big)\right| \leq 2H,$$

$$\mathbb{E}[\eta_i^2|\mathcal{F}_i] = \bar{\sigma}_{i,h}^{-2}\big[\mathbb{V}_h\widetilde{V}\big](s_h^i, a_h^i) \leq 1/d^3H,$$

$$\max_{1 \leq i \leq k}\left\{|\eta_i| \cdot \min\left\{1, \|\mathbf{x}_i\|_{\mathbf{\Sigma}_{i,h}^{-1}}\right\}\right\} \leq \max_{1 \leq i \leq k}\left\{2H\bar{\sigma}_{i,h}^{-1}\|\mathbf{x}_i\|_{\mathbf{\Sigma}_{i,h}^{-1}}\right\} \leq 1/d^3H,$$

where we use the construction of $\bar{\sigma}_{i,h}$ in (3.7) and (E.8). After taking union probability bound over $\epsilon$-covering for function class $\widetilde{\mathcal{V}}_{h+1} - \big[V_{h+1}^{*,\rho}\big]_{\alpha'}$, we have

$$\left\|\sum_{\tau=1}^{k-1}\bar{\sigma}_{\tau,h}^{-2}\phi_h^\tau\big[\widetilde{V}(s_{h+1}^\tau) - [\mathbb{P}_h^0\widetilde{V}](s_h^\tau, a_h^\tau)\big]\right\|_{\mathbf{\Sigma}_{k,h}^{-1}} \leq \widetilde{\mathcal{O}}(\sqrt{d}).$$

For simplicity, we denote that $\bar{V} = \Delta V - \widetilde{V} = \Delta_{\alpha'}\hat{V}_{k,h+1}^\rho - \widetilde{V}$ and have $\sup_s |\bar{V}(s)| \leq \epsilon$. Then we obtain

$$\begin{aligned}
J_2 &= \left\|\sum_{\tau=1}^{k-1}\bar{\sigma}_{\tau,h}^{-2}\phi_h^\tau\big[\Delta_{\alpha'}\hat{V}_{k,h+1}^\rho(s_{h+1}^\tau) - [\mathbb{P}_h^0(\Delta_{\alpha'}\hat{V}_{k,h+1}^\rho)](s_h^\tau, a_h^\tau)\big]\right\|_{\mathbf{\Sigma}_{k,h}^{-1}} \\
&\leq 2\left\|\sum_{\tau=1}^{k-1}\bar{\sigma}_{\tau,h}^{-2}\phi_h^\tau\big[\widetilde{V}(s_{h+1}^\tau) - [\mathbb{P}_h^0\widetilde{V}](s_h^\tau, a_h^\tau)\big]\right\|_{\mathbf{\Sigma}_{k,h}^{-1}} \\
&\quad + 2\left\|\sum_{\tau=1}^{k-1}\bar{\sigma}_{\tau,h}^{-2}\phi_h^\tau\big[\bar{V}(s_{h+1}^\tau) - [\mathbb{P}_h^0\bar{V}](s_h^\tau, a_h^\tau)\big]\right\|_{\mathbf{\Sigma}_{k,h}^{-1}} \\
&\leq \widetilde{\mathcal{O}}(\sqrt{d}) + 4\epsilon k/\sqrt{\lambda} \\
&\leq \widetilde{\mathcal{O}}(\sqrt{d}),
\end{aligned}$$

where we use that $\epsilon = \sqrt{\lambda}/4H^2d^3K$. Finally, we have

$$\left\|\sum_{\tau=1}^{k-1}\bar{\sigma}_{\tau,h}^{-2}\phi_h^\tau\big[[\hat{V}_{k,h+1}^\rho(s_{h+1}^\tau)]_{\alpha'} - [\mathbb{P}_h^0[\hat{V}_{k,h+1}^\rho]_{\alpha'}](s_h^\tau, a_h^\tau)\big]\right\|_{\mathbf{\Sigma}_{k,h}^{-1}} = J_1 + J_2 \leq \gamma,$$

where $\gamma = \widetilde{\mathcal{O}}(\sqrt{d})$. Thus, by induction we complete the proof. $\qquad\square$

## E.6 Proof of Lemma D.7

*Proof of Lemma D.7.* Conditioned on the event $\mathcal{E}$ and $\bar{\mathcal{E}}$, to bound the weight $\bar{\sigma}_{k,h}^2$, recall from the definition (3.7), we have

$$\bar{\sigma}_{k,h} = \max\left\{\sigma_{k,h}, 1, \sqrt{2d^3H^2}\|\phi(s_h^k, a_h^k)\|_{\mathbf{\Sigma}_{k,h}^{-1}}^{\frac{1}{2}}\right\},$$

According to (3.6), we have

$$\sigma_{k,h}^2 = \big[\bar{\mathbb{V}}_h\hat{V}_{k,h+1}^\rho\big](s_h^k, a_h^k) + E_{k,h} + d^3H \cdot D_{k,h} + \frac{1}{2},$$

where $E_{k,h}, D_{k,h}$ are defined as follows

$$E_{k,h} = \min\left\{\widetilde{\beta}\|\phi(s_h^k, a_h^k)\|_{\mathbf{\Lambda}_{k,h}^{-1}}, H^2\right\} + \min\left\{2H\bar{\beta}\|\phi(s_h^k, a_h^k)\|_{\mathbf{\Lambda}_{k,h}^{-1}}, H^2\right\},$$

$$D_{k,h} = \min\left\{4H\big(\phi(s_h^k, a_h^k)^\top\hat{\boldsymbol{w}}_{h,1}^k - \phi(s_h^k, a_h^k)^\top\breve{\boldsymbol{w}}_{h,1}^k + 2\bar{\beta}\|\phi(s_h^k, a_h^k)\|_{\mathbf{\Lambda}_{k,h}^{-1}}\big), H^2\right\},$$

where $\bar{\beta} = \widetilde{\mathcal{O}}(d^{\frac{3}{2}}H^{\frac{3}{2}})$, $\widetilde{\beta} = \widetilde{\mathcal{O}}(d^{\frac{3}{2}}H^3)$ when we set $\lambda = 1/H^2$. Note that

$$\sqrt{2d^3H^2}\|\phi(s_h^k, a_h^k)\|_{\mathbf{\Lambda}_{k,h}^{-1}}^{\frac{1}{2}} \leq \sqrt{2d^3H^2}\|\phi(s_h^k, a_h^k)\|_2^{\frac{1}{2}}/\lambda^{\frac{1}{4}} \leq \sqrt{2d^3H^3}.$$

Also note that

$$\sigma_{k,h}^2 = \big[\bar{\mathbb{V}}_h\hat{V}_{k,h+1}^\rho\big](s_h^k, a_h^k) + E_{k,h} + d^3H \cdot D_{k,h} + \frac{1}{2}$$

$$\leq H^2 + 2H^2 + d^3 H \cdot H^2 + \frac{1}{2}$$
$$\leq 2d^3 H^3.$$

Then we obtain the trivial upper bound $\bar{\alpha}$ for $\bar{\sigma}_{k,h}$

$$\bar{\sigma}_{k,h} \leq 2\sqrt{d^3 H^3} = \bar{\alpha}, \tag{E.9}$$

Based on Lemma D.3, we have

$$\left[\bar{\mathbb{V}}_h \hat{V}^\rho_{k,h+1}\right]\left(s^k_h, a^k_h\right) \leq E_{k,h} + D_{k,h} + \left[\mathbb{V}_h V^{*,\rho}_{h+1}\right]\left(s^k_h, a^k_h\right).$$

Then we have

$$\sigma^2_{k,h} \leq \left[\mathbb{V}_h V^{*,\rho}_{h+1}\right]\left(s^k_h, a^k_h\right) + 2E_{k,h} + 2d^3 H \cdot D_{k,h} + \frac{1}{2}.$$

Next, we carefully bound $\sigma^2_{k,h}$ and $\bar{\sigma}^2_{k,h}$. To this end, we bound term $E_{k,h}, D_{k,h}$ when $k$ is large enough. The intuition is that, when the episode $k$ is large enough, all the error terms should be small under the assumption (E.10).

**Bound term $E_{k,h}$:**   Note that based on (4.2), with the same analysis as the proof of Corollary 5.3 in [23], with probability at least $1 - \delta$, we have

$$\lambda_{\min}(\boldsymbol{\Lambda}_{k,h}) \geq \max\left\{ c(k-1)/d + \lambda - \sqrt{32k \log(dKH/\delta)}, \lambda \right\}.$$

Then when we choose $k > 512d^2 \log(dKH/\delta)/c^2$ and note that $\lambda = 1/H^2$, we have

$$c(k-1)/d + \lambda - \sqrt{32k \log(dKH/\delta)} \geq \frac{c}{2d}k,$$

which indicates that

$$\lambda_{\min}(\boldsymbol{\Lambda}_{k,h}) \geq \frac{c}{2d}k. \tag{E.10}$$

Then when $k > 512d^2 \log(dKH/\delta)/c^2$, we can calculate that

$$\left\|\phi\left(s^k_h, a^k_h\right)\right\|_{\boldsymbol{\Lambda}^{-1}_{k,h}} = \left\|\boldsymbol{\Lambda}^{-\frac{1}{2}}_{k,h}\phi\left(s^k_h, a^k_h\right)\right\|_2 \leq \sqrt{\lambda_{\max}(\boldsymbol{\Lambda}^{-1}_{k,h})} \leq \sqrt{\frac{2d}{kc}}, \tag{E.11}$$

where in the first inequality we use the fact that $\|\phi(s,a)\|_2 \leq 1$ for all $(s,a) \in \mathcal{S} \times \mathcal{A}$. Then when $k$ is large enough and also at least $k > 512d^2 \log(dKH/\delta)/c^2$, we can have that

$$E_{k,h} \leq \widetilde{\mathcal{O}}\left(\frac{1}{\sqrt{kc}}d^2 H^3\right).$$

This indicates that there exists an absolute constant $c_E > 0$ such that

$$E_{k,h} \leq c_E \frac{d^2 H^3}{\sqrt{k}}.$$

**Bound term $D_{k,h}$:** note that $\hat{\boldsymbol{w}}^k_{h,1}$ and $\check{\boldsymbol{w}}^k_{h,1}$ have the closed-form expression as follows

$$\hat{\boldsymbol{w}}^k_{h,1} = \boldsymbol{\Lambda}^{-1}_{k,h} \sum_{\tau=1}^{k-1} \phi\left(s^\tau_h, a^\tau_h\right)\hat{V}^\rho_{k,h+1}\left(s^\tau_{h+1}\right),$$

$$\check{\boldsymbol{w}}^k_{h,1} = \boldsymbol{\Lambda}^{-1}_{k,h} \sum_{\tau=1}^{k-1} \phi\left(s^\tau_h, a^\tau_h\right)\check{V}^\rho_{k,h+1}\left(s^\tau_{h+1}\right).$$

Thus we can calculate that

$$\phi\left(s^k_h, a^k_h\right)^\top \hat{\boldsymbol{w}}^k_{h,1} - \phi\left(s^k_h, a^k_h\right)^\top \check{\boldsymbol{w}}^k_{h,1}$$
$$= \phi\left(s^k_h, a^k_h\right)^\top \boldsymbol{\Lambda}^{-1}_{k,h} \sum_{\tau=1}^{k-1} \phi\left(s^\tau_h, a^\tau_h\right)\left(\hat{V}^\rho_{k,h+1}\left(s^\tau_{h+1}\right) - \check{V}^\rho_{k,h+1}\left(s^\tau_{h+1}\right)\right),$$

$$\leq \left\|\phi(s_h^k, a_h^k)\right\|_{\mathbf{\Lambda}_{k,h}^{-1}} \cdot \left\|\sum_{\tau=1}^{k-1} \phi(s_h^\tau, a_h^\tau)\left(\hat{V}_{k,h+1}^\rho(s_{h+1}^\tau) - \check{V}_{k,h+1}^\rho(s_{h+1}^\tau)\right)\right\|_{\mathbf{\Lambda}_{k,h}^{-1}}$$

$$\leq \left\|\phi(s_h^k, a_h^k)\right\|_{\mathbf{\Lambda}_{k,h}^{-1}} \cdot \sum_{\tau=1}^{k-1} \left\|\phi(s_h^\tau, a_h^\tau)\right\|_{\mathbf{\Lambda}_{k,h}^{-1}} \cdot \left(\hat{V}_{k,h+1}^\rho(\tilde{s}_{h+1}^k) - \check{V}_{k,h+1}^\rho(\tilde{s}_{h+1}^k)\right)$$

$$\leq \sqrt{\frac{2d}{kc}} \sum_{\tau=1}^{k-1} \sqrt{\frac{2d}{kc}} \cdot \left(\hat{V}_{k,h+1}^\rho(\tilde{s}_{h+1}^k) - \check{V}_{k,h+1}^\rho(\tilde{s}_{h+1}^k)\right),$$

$$\leq 2d/c \cdot \left(\hat{V}_{k,h+1}^\rho(\tilde{s}_{h+1}^k) - \check{V}_{k,h+1}^\rho(\tilde{s}_{h+1}^k)\right), \tag{E.12}$$

where $\tilde{s}_{h+1}^k = \operatorname{argmax}_{s \in \mathcal{S}} \left\{\hat{V}_{k,h+1}^\rho(s) - \check{V}_{k,h+1}^\rho(s)\right\}$, the first inequality holds because of Cauchy-Schwarz inequality, the third inequality holds due to (E.11). Next, we bound $\hat{V}_{k,h+1}^\rho(\tilde{s}_{h+1}^k) - \check{V}_{k,h+1}^\rho(\tilde{s}_{h+1}^k)$. The intuition is that when $k$ is large, both $\hat{V}_{k,h+1}^\rho$ and $\check{V}_{k,h+1}^\rho$ should be close to the robust optimal value function. Thus, $\hat{V}_{k,h+1}^\rho$ should be close to $\check{V}_{k,h+1}^\rho$, and the closeness could be quantified by the bonus terms, which is of order $\widetilde{\mathcal{O}}(d/\sqrt{k})$ under the assumption (E.10). In particular, we have

$$\hat{V}_{k,h+1}^\rho(\tilde{s}_{h+1}^k) - \check{V}_{k,h+1}^\rho(\tilde{s}_{h+1}^k)$$
$$= \underbrace{\hat{V}_{k,h+1}^\rho(\tilde{s}_{h+1}^k) - V_{h+1}^{*,\rho}(\tilde{s}_{h+1}^k)}_{\text{I}} + \underbrace{V_{h+1}^{*,\rho}(\tilde{s}_{h+1}^k) - \check{V}_{k,h+1}^\rho(\tilde{s}_{h+1}^k)}_{\text{II}}. \tag{E.13}$$

**Bound term I in** (E.13): note that

$$\hat{V}_{k,h}^\rho(s) - V_h^{*,\rho}(s)$$
$$= \hat{Q}_{k,h}^\rho\big(s, \pi_h^k(s)\big) - Q_h^{*,\rho}\big(s, \pi_h^*(s)\big)$$
$$\leq \hat{Q}_{k,h}^\rho\big(s, \pi_h^k(s)\big) - Q_h^{*,\rho}\big(s, \pi_h^k(s)\big)$$
$$\leq \inf_{P_h(\cdot|s,a) \in \mathcal{U}_h^\rho(s,a;\boldsymbol{\mu}_h^0)} \big[\mathbb{P}_h \hat{V}_{k,h+1}^\rho\big]\big(s, \pi_h^k(s)\big) - \inf_{P_h(\cdot|s,a) \in \mathcal{U}_h^\rho(s,a;\boldsymbol{\mu}_h^0)} \big[\mathbb{P}_h V_{h+1}^{*,\rho}\big]\big(s, \pi_h^k(s)\big)$$
$$\quad + \Delta_h^k\big(s, \pi_h^k(s)\big) + \hat{\Gamma}_{k,h}\big(s, \pi_h^k(s)\big)$$
$$\leq \big[\hat{\mathbb{P}}_h\big(\hat{V}_{k,h+1}^\rho - V_{h+1}^{*,\rho}\big)\big]\big(s, \pi_h^k(s)\big) + 2\hat{\Gamma}_{k,h}\big(s, \pi_h^k(s)\big),$$

where the second inequality holds due to the definition of $\hat{Q}_{k,h}^\rho$, robust Bellman equation and Lemma D.4, $\hat{P}_h(\cdot|s,a) = \operatorname{arginf}_{P_h(\cdot|s,a) \in \mathcal{U}_h^\rho(s,a;\boldsymbol{\mu}_{h,i}^0)} \big[\mathbb{P}_h V_{h+1}^{*,\rho}\big](s,a), \forall (s,a) \in \mathcal{S} \times \mathcal{A}$. By recursively applying it, then we have

$$\hat{V}_{k,h}^\rho(s) - V_h^{*,\rho}(s) \leq \big[\hat{\mathbb{P}}_h\big(\hat{V}_{k,h+1}^\rho - V_{h+1}^{*,\rho}\big)\big]\big(s, \pi_h^k(s)\big) + 2\hat{\Gamma}_{k,h}\big(s, \pi_h^k(s)\big)$$

$$\leq 2 \sum_{h'=h}^H \mathbb{E}^{\pi_{h'}^k, \hat{P}}\big[\hat{\Gamma}_{k,h'}(s,a)|s_{h'} = s\big].$$

Note that by (E.9) $\bar{\sigma}_{k,h}^2 \leq \bar{\alpha}^2$, we have $\mathbf{\Sigma}_{k,h} \succcurlyeq \bar{\alpha}^{-2}\mathbf{\Lambda}_{k,h}$. Similar to the analysis of (E.11), when $k > 512/c^2 \log(dKH/\delta)$, we have

$$\hat{\Gamma}_{k,h}(s,a) = \beta \sum_{i=1}^d \phi_i(s,a)\sqrt{\mathbf{1}_i^\top \mathbf{\Sigma}_{k,h}^{-1} \mathbf{1}_i}$$

$$\leq \beta\bar{\alpha} \sum_{i=1}^d \phi_i(s_h^k, a_h^k)\sqrt{\mathbf{1}_i^\top \mathbf{\Lambda}_{k,h}^{-1} \mathbf{1}_i}$$

$$\leq \beta\bar{\alpha}\sqrt{\lambda_{\max}(\mathbf{\Lambda}_{k,h}^{-1})}$$

$$\leq \sqrt{\frac{2d}{kc}} \beta\bar{\alpha}.$$

811 Then we have

$$\hat{V}^{\rho}_{k,h}(s) - V^{*,\rho}_h(s) \leq 2H\sqrt{\frac{2d}{kc}}\beta\bar{\alpha} \leq \frac{4\beta\sqrt{d}\bar{\alpha}H}{\sqrt{kc}}.$$

812 Therefore, we can bound $I$ as follows

$$\mathrm{I} = \hat{V}^{\rho}_{k,h+1}(\tilde{s}^k_{h+1}) - V^{*,\rho}_{h+1}(\tilde{s}^k_{h+1}) \leq \frac{4\beta\sqrt{d}\bar{\alpha}H}{\sqrt{kc}}.$$

813 **Bound term II in** (E.13)**:** Similar to the analysis above, we can derive the similar result as follows

$$\mathrm{II} = V^{*,\rho}_{h+1}(\tilde{s}^k_{h+1}) - \check{V}^{\rho}_{k,h+1}(\tilde{s}^k_{h+1}) \leq \frac{4\bar{\beta}\sqrt{d}\bar{\alpha}H}{\sqrt{kc}}.$$

814 Now we can bound that

$$\phi\big(s^k_h, a^k_h\big)^\top \hat{\boldsymbol{w}}^k_{h,1} - \phi\big(s^k_h, a^k_h\big)^\top \check{\boldsymbol{w}}^k_{h,1} \leq \frac{2d}{c} \cdot \frac{4(\bar{\beta}+\beta)\sqrt{d}\bar{\alpha}H}{\sqrt{kc}} \leq \frac{16\bar{\beta}d^{3/2}\bar{\alpha}H}{\sqrt{kc^3}}.$$

815 Then when $k$ is large enough, we can have that

$$D_{k,h} \leq \widetilde{\mathcal{O}}\Big(\frac{\bar{\alpha}}{\sqrt{k}}d^3 H^{\frac{7}{2}}\Big).$$

816 This indicates that there exists an absolute constant $c_D > 0$ such that

$$D_{k,h} \leq c_D \frac{\bar{\alpha}}{\sqrt{k}}d^3 H^{\frac{7}{2}}.$$

817 When $k$ is large enough, we have

$$\sigma^2_{k,h} \leq [\mathbb{V}_h V^{*,\rho}_{h+1}]\big(s^k_h, a^k_h\big) + (2E_{k,h} + 2d^3 H \cdot D_{k,h}) + \frac{1}{2}$$
$$\leq [\mathbb{V}_h V^{*,\rho}_{h+1}]\big(s^k_h, a^k_h\big) + 2c_E \frac{\bar{\alpha}}{\sqrt{k}}d^2 H^3 + 2c_D \frac{\bar{\alpha}}{\sqrt{k}}d^6 H^{\frac{9}{2}} + \frac{1}{2}.$$

818 When we choose $\widetilde{K} = \widetilde{c} \cdot \bar{\alpha}^2 d^{12} H^9$ where $\widetilde{c} = \widetilde{\mathcal{O}}(1)$. When $k > \widetilde{K}$, then we have

$$\bar{\sigma}^2_{k,h} = \max\Big\{\sigma^2_{k,h}, 1, 2d^3 H^2\big\|\phi\big(s^k_h, a^k_h\big)\big\|_{\boldsymbol{\Sigma}^{-1}_{k,h}}\Big\}$$
$$\leq \max\big\{\big[\mathbb{V}_h V^{*,\rho}_{h+1}\big]\big(s^k_h, a^k_h\big) + 1, 1\big\}$$
$$\leq 2\big[\mathbb{V}_h V^{*,\rho}_{h+1}\big(s^k_h, a^k_h\big)\big]_{[1,H^2]}.$$

819 Based on Lemma H.10, we have

$$\big[\mathbb{V}_h V^{*,\rho}_{h+1}\big](s,a) \leq \Big(\frac{1-(1-\rho)^{H-h+1}}{\rho}\Big)^2 \leq \Big(\frac{1-(1-\rho)^H}{\rho}\Big)^2 = \boldsymbol{\Theta}\Big(\min\Big\{\frac{1}{\rho^2}, H^2\Big\}\Big).$$

820 Then when $k > \widetilde{K}$, we have

$$\bar{\sigma}^2_{k,h} \leq \mathcal{O}\Big(\min\Big\{\frac{1}{\rho^2}, H^2\Big\}\Big).$$

821 Additionally, note that $\bar{\alpha}^2 = \mathcal{O}\big(d^3 H^3\big)$, we have

$$\widetilde{K} = \widetilde{\mathcal{O}}\big(d^{15} H^{12}\big).$$

822 This completes the proof. □

## F Supporting Lemmas

**Lemma F.1** (Number of value function updates). The number of episodes where the algorithm updates the value function in Algorithm 1 is upper bounded by $dH \log(1 + K/\lambda)$.

*Proof of Lemma F.1.* This proof is the same as [10, Lemma F.1] because of the same rare-switching condition (Line 5 in Algorithm 1). $\quad\square$

**Lemma F.2.** For any $(k, h) \in [K] \times [H]$, the weight $\hat{\boldsymbol{\nu}}_h^{\rho,k}$ satisfies

$$\left\|\hat{\boldsymbol{\nu}}_h^{\rho,k}\right\|_2 \le 2H\sqrt{dk/\lambda}.$$

*Proof of Lemma F.2.* Denote $\alpha_i = \operatorname{argmax}_{\alpha \in [0,H]} \left\{\hat{z}_{h,i}^k(\alpha) - \rho\alpha\right\}, i \in [d]$. Then we have

$$
\begin{aligned}
\left\|\hat{\boldsymbol{\nu}}_h^{\rho,k}\right\|_2 &= \left\|\left[\max_{\alpha \in [0,H]}\left\{\hat{z}_{h,i}^k(\alpha) - \rho\alpha\right\}\right]_{i \in [d]}\right\|_2 \\
&\le \rho\sqrt{d}\alpha + \left\|\left[\left(\boldsymbol{\Sigma}_{k,h}^{-1}\sum_{\tau=1}^{k-1}\bar{\sigma}_{\tau,h}^{-2}\boldsymbol{\phi}(s_h^\tau, a_h^\tau)\left[\hat{V}_{k,h+1}^\rho(s_{h+1}^\tau)\right]_{\alpha_i}\right)_i\right]_{i \in [d]}\right\|_2 \\
&\le H\sqrt{d} + H \cdot \left\|\boldsymbol{\Sigma}_{k,h}^{-1}\sum_{\tau=1}^{k-1}\bar{\sigma}_{\tau,h}^{-2}\boldsymbol{\phi}(s_h^\tau, a_h^\tau)\right\|_2 \\
&\le H\sqrt{d} + H\sqrt{k/\lambda} \cdot \left(\sum_{\tau=1}^{k-1}\left(\bar{\sigma}_{\tau,h}^{-1}\boldsymbol{\phi}(s_h^\tau, a_h^\tau)\right)^\top \boldsymbol{\Sigma}_{k,h}^{-1}\left(\bar{\sigma}_{\tau,h}^{-1}\boldsymbol{\phi}(s_h^\tau, a_h^\tau)\right)\right)^{\frac{1}{2}} \\
&\le H\sqrt{d} + H\sqrt{dk/\lambda} \\
&\le 2H\sqrt{dk/\lambda},
\end{aligned}
$$

where the first inequality holds due to the triangle inequality, the second inequality holds from the fact that $\rho \le 1, 0 \le \alpha \le H$ and $0 \le \left[\hat{V}_{k,h+1}^\rho(s_{h+1}^\tau)\right]_{\alpha_i} \le H$, the third inequality holds because of Lemma H.5 and the fourth inequality holds because $\boldsymbol{\Sigma}_{k,h} \succcurlyeq \lambda\mathbf{I}$ and Lemma H.4. This completes the proof. $\quad\square$

**Lemma F.3.** Under a linear MDP, for any stage $h \in [H]$ and any bounded function $V: \mathcal{S} \to [0, H]$, there always exists a vector $\boldsymbol{z} \in \mathbb{R}^d$ such that for all $(s, a) \in \mathcal{S} \times \mathcal{A}$, we have

$$\left[\mathbb{P}_h^0 V\right](s, a) = \boldsymbol{z}^\top \boldsymbol{\phi}(s, a),$$

where $\boldsymbol{z}$ satisfies that $\|\boldsymbol{z}\|_2 \le H\sqrt{d}$.

*Proof of Lemma F.3.* Based on Assumption 2.1, we have

$$
\begin{aligned}
\left[\mathbb{P}_h^0 V\right](s, a) &= \int \mathbb{P}_h^0(s'|s, a)V(s')ds' \\
&= \int \boldsymbol{\phi}(s, a)^\top V(s')d\boldsymbol{\mu}_h^0(s') \\
&= \boldsymbol{\phi}(s, a)^\top \int V(s')d\boldsymbol{\mu}_h^0(s') \\
&= \boldsymbol{\phi}(s, a)^\top \boldsymbol{z},
\end{aligned}
$$

where $\boldsymbol{z} = \int V(s')d\boldsymbol{\mu}_h^0(s')$. Thus we have

$$\|\boldsymbol{z}\|_2 = \left\|\int V(s')d\boldsymbol{\mu}_h^0(s')\right\|_2 \le \max_{s'} V(s') \cdot \left\|\boldsymbol{\mu}_h^0(\mathcal{S})\right\|_2 \le H\sqrt{d}.$$

This completes the proof. $\quad\square$

# G  Proof of the Minimax Lower Bound

841 In this section, we prove the minimax lower bound. To this end, we first introduce the construction of
842 hard instances in Appendix G.1, and then we prove Theorem 5.1 in Appendix G.3.

## G.1  Construction of Hard Instances

844 We construct a family of $d$-rectangular linear DRMDPs based on the hard-to-learn linear MDP
845 introduced in [62]. Let $\delta = 1/H$, $\Delta = \sqrt{\delta/K}/(4\sqrt{2})$. Each $d$-rectangular linear DRMDP in
846 this family is parameterized by a Boolean vector $\boldsymbol{\xi} = \{\boldsymbol{\xi}_h\}_{h \in [H-1]}$, where $\boldsymbol{\xi}_h \in \{-\Delta, \Delta\}^d$. For
847 a given $\boldsymbol{\xi}$ and uncertainty level $\rho \in (0, 3/4]$, the corresponding $d$-rectangular linear DRMDP $M_{\boldsymbol{\xi}}^\rho$
848 has the following structure. The state space $\mathcal{S} = \{x_1, x_2, \cdots, x_H, x_{H+1}\}$ and the action space
849 $\mathcal{A} = \{-1, 1\}^d$. The first state is always $x_1$. The feature mapping $\phi : \mathcal{S} \times \mathcal{A} \to \mathbb{R}^{2d+2}$ is defined to
850 depend on the state $x_h$ through $\boldsymbol{\xi}_h$ as follows:

$$\phi(x_1, a) = \begin{pmatrix} \frac{1}{2d} - \frac{\delta}{d} - \xi_{11}a_1 \\ \frac{1}{2d} - \frac{\delta}{d} - \xi_{12}a_2 \\ \vdots \\ \frac{1}{2d} - \frac{\delta}{d} - \xi_{1d}a_d \\ \frac{1}{2} \\ \frac{\delta}{d} + \xi_{11}a_1 \\ \frac{\delta}{d} + \xi_{12}a_2 \\ \vdots \\ \frac{\delta}{d} + \xi_{1d}a_d \\ 0 \end{pmatrix}, \phi(x_2, a) = \begin{pmatrix} \frac{1}{2d} - \frac{\delta}{d} - \xi_{21}a_1 \\ \frac{1}{2d} - \frac{\delta}{d} - \xi_{22}a_2 \\ \vdots \\ \frac{1}{2d} - \frac{\delta}{d} - \xi_{2d}a_d \\ \frac{1}{2} \\ \frac{\delta}{d} + \xi_{21}a_1 \\ \frac{\delta}{d} + \xi_{22}a_2 \\ \vdots \\ \frac{\delta}{d} + \xi_{2d}a_d \\ 0 \end{pmatrix}, \cdots,$$

$$\phi(x_{H-1}, a) = \begin{pmatrix} \frac{1}{2d} - \frac{\delta}{d} - \xi_{H-1,1}a_1 \\ \frac{1}{2d} - \frac{\delta}{d} - \xi_{H-1,2}a_2 \\ \vdots \\ \frac{1}{2d} - \frac{\delta}{d} - \xi_{H-1,d}a_d \\ \frac{1}{2} \\ \frac{\delta}{d} + \xi_{H-1,1}a_1 \\ \frac{\delta}{d} + \xi_{H-1,2}a_2 \\ \vdots \\ \frac{\delta}{d} + \xi_{H-1,d}a_d \\ 0 \end{pmatrix}, \phi(x_H, a) = \begin{pmatrix} 0 \\ 0 \\ \vdots \\ 0 \\ 0 \\ 0 \\ 0 \\ \vdots \\ 0 \\ 1 \end{pmatrix}, \phi(x_{H+1}, a) = \begin{pmatrix} 0 \\ 0 \\ \vdots \\ 0 \\ 0 \\ \frac{1}{d} \\ \frac{1}{d} \\ \vdots \\ \frac{1}{d} \\ 0 \end{pmatrix}.$$

851 We assume that

$$K \geq 9d^2 H/32 \text{ and } H \geq 6, \tag{G.1}$$

852 such that $\frac{1}{2d} - \frac{1}{dH} - \delta \geq 0$. Then it can be easily checked that for any $s \in \mathcal{S}$, we have $\phi_i(s, a) \geq 0$
853 and $\sum_{i=1}^{2d+2} \phi_i(s, a) = 1$. The factor distribution $\boldsymbol{\mu}_1 : \mathcal{S} \to \mathbb{R}^{2d+2}$ is defined as follows.

$$\boldsymbol{\mu}_1(\cdot) = (\underbrace{\delta_{x_2}(\cdot), \cdots, \delta_{x_2}(\cdot)}_{d \text{ terms}}, \delta_{x_2}(\cdot), \underbrace{\delta_{x_{H+1}}(\cdot), \cdots, \delta_{x_{H+1}}(\cdot)}_{d \text{ terms}}, \delta_{x_H}(\cdot))^\top.$$

854 Similarly, for $h = 2, \ldots, H$, we have

$$\boldsymbol{\mu}_2(\cdot) = (\delta_{x_3}(\cdot), \cdots, \delta_{x_3}(\cdot), \delta_{x_3}(\cdot), \delta_{x_{H+1}}(\cdot), \cdots, \delta_{x_{H+1}}(\cdot), \delta_{x_H}(\cdot))^\top,$$

$$\cdots$$

$$\boldsymbol{\mu}_{H-1}(\cdot) = \boldsymbol{\mu}_H(\cdot) = (\delta_{x_H}(\cdot), \cdots, \delta_{x_H}(\cdot), \delta_{x_H}(\cdot), \delta_{x_{H+1}}(\cdot), \cdots, \delta_{x_{H+1}}(\cdot), \delta_{x_H}(\cdot))^\top,$$

855 Note that for each episode $k$, the initial state $s_1^k$ is always $x_1$. In the nominal environment, at step $h$,
856 the state $s_h^k$ is either $x_h$ or $x_{H+1}$. State $x_H$ and $x_{H+1}$ are absorbing states. Figure 4(a) illustrates the
857 nominal MDP.

858 Now we construct the reward parameters $\{\boldsymbol{\theta}_h\}_{h \in [H]}$ as follows.

$$\boldsymbol{\theta}_h = (1, 1, \cdots, 1, -1, 1, 1, \cdots, 1, 0)^\top, \ \forall h \in [H].$$

859 We have $\forall h \in [H]$,

$$r_h(x_H, a) = \boldsymbol{\phi}(x_H, \boldsymbol{a})^\top \boldsymbol{\theta}_h = 0,$$
$$r_h(x_h, a) = \boldsymbol{\phi}(x_h, \boldsymbol{a})^\top \boldsymbol{\theta}_h = 0,$$
$$r_h(x_{H+1}, a) = \boldsymbol{\phi}(x_{H+1}, \boldsymbol{a})^\top \boldsymbol{\theta}_h = 1.$$

860 Thus, only the transition starting from $x_{H+1}$ generates a reward of 1, and transitions starting from any
861 other state generate 0 reward. Next, we consider the model perturbation. An observation is that $x_H$ is
862 the worst state since it is an absorbing state with zero reward. By the definition of the $d$-rectangular
863 uncertainty set, the worst case kernel is the linear combination of worst case factor distributions.
864 Further, by the definition of the factor uncertainty set, the worst case factor distribution is the one that
865 leads to the highest probability $\rho$ to the worst state $x_H$. Thus, the worst factor distributions are

$$\check{\boldsymbol{\mu}}_1 = ((1-\rho)\delta_{x_2} + \rho\delta_{x_H}, (1-\rho)\delta_{x_2} + \rho\delta_{x_H}, \cdots, (1-\rho)\delta_{x_2} + \rho\delta_{x_H}, (1-\rho)\delta_{x_2} + \rho\delta_{x_H},$$

$$(1-\rho)\delta_{x_{H+1}} + \rho\delta_{x_H}, (1-\rho)\delta_{x_{H+1}} + \rho\delta_{x_H}, \cdots, (1-\rho)\delta_{x_{H+1}} + \rho\delta_{x_H}, \delta_{x_H})^\top,$$

$$\check{\boldsymbol{\mu}}_2 = ((1-\rho)\delta_{x_3} + \rho\delta_{x_H}, (1-\rho)\delta_{x_3} + \rho\delta_{x_H}, \cdots, (1-\rho)\delta_{x_3} + \rho\delta_{x_H}, (1-\rho)\delta_{x_3} + \rho\delta_{x_H},$$

$$(1-\rho)\delta_{x_{H+1}} + \rho\delta_{x_H}, (1-\rho)\delta_{x_{H+1}} + \rho\delta_{x_H}, \cdots, (1-\rho)\delta_{x_{H+1}} + \rho\delta_{x_H}, \delta_{x_H})^\top,$$

$$\cdots$$

$$\check{\boldsymbol{\mu}}_{H-1} = ((1-\rho)\delta_{x_{H-1}} + \rho\delta_{x_H}, (1-\rho)\delta_{x_{H-1}} + \rho\delta_{x_H}, \cdots, (1-\rho)\delta_{x_{H-1}} + \rho\delta_{x_H}, (1-\rho)\delta_{x_{H-1}} + \rho\delta_{x_H},$$

$$(1-\rho)\delta_{x_{H+1}} + \rho\delta_{x_H}, (1-\rho)\delta_{x_{H+1}} + \rho\delta_{x_H}, \cdots, (1-\rho)\delta_{x_{H+1}} + \rho\delta_{x_H}, \delta_{x_H})^\top,$$

$$\check{\boldsymbol{\mu}}_{H-1} = \boldsymbol{\mu}_H.$$

866 Figure 4(b) illustrates the worst case MDP.

## G.2 Reduction from $d$-Rectangular DRMDP to Linear Bandits

868 Note that by construction, at steps $h = 1, \cdots, H - 2$, the probability of transitioning to the worst
869 case state $x_H$ is independent of the action $a$. Moreover, since $x_{H+1}$ is the only rewarding state,
870 so the optimal action at step $h$ is the one the leads to the largest probability to $x_{H+1}$, i.e., $\boldsymbol{a}_h^\star =$
871 $\operatorname{argmax}_{\boldsymbol{a} \in \mathcal{A}} \langle \boldsymbol{\xi}_h, \boldsymbol{a} \rangle$. Further, in the nominal environment, state $x_h$ can only be reached through states
872 $x_1, x_2, \cdots, x_{h-1}$. As discussed by [62], knowing the state $x_h$ is equivalent to knowing the entire
873 history starting from the initial state at current episode. Consequently, policies dictating what actions
874 to take upon reaching a state at the beginning of an episode are equivalent to policies relying on the
875 "within episode" history (we refer to the discussion in E.1 of [62] for more details). In the following
876 lemma, we shows that the average suboptimality of the $d$-rectangular DRMDP can be lower bounded
877 by the regret of $H/2$ bandit instances.

878 **Lemma G.1.** With the choice of $d, K, H$ in (G.1), we have $3d\Delta \leq \delta$. Fix $\boldsymbol{\xi} = \{\boldsymbol{\xi}_h\}_{h \in [H-1]}$. Fix a
879 possibly history dependent policy $\pi$ and define $\bar{\boldsymbol{a}}_h^\pi = \mathbb{E}_{\boldsymbol{\xi}}[\boldsymbol{a}_h | s_h = x_h]$ as the expected action taken
880 by the policy when it visits state $x_h$ in stage $h$. Then, there exist a constant $c > 0$ such that

$$V_1^{\star,\rho}(x_1) - V_1^{\pi,\rho}(x_1) \geq c \min\left\{\frac{1}{\rho}, H\right\} \sum_{h=1}^{H/2} \left(\max_{a \in \mathcal{A}} \langle \boldsymbol{\mu}_h, \boldsymbol{a} \rangle - \langle \boldsymbol{\mu}_h, \bar{\boldsymbol{a}}_h^\pi \rangle\right).$$

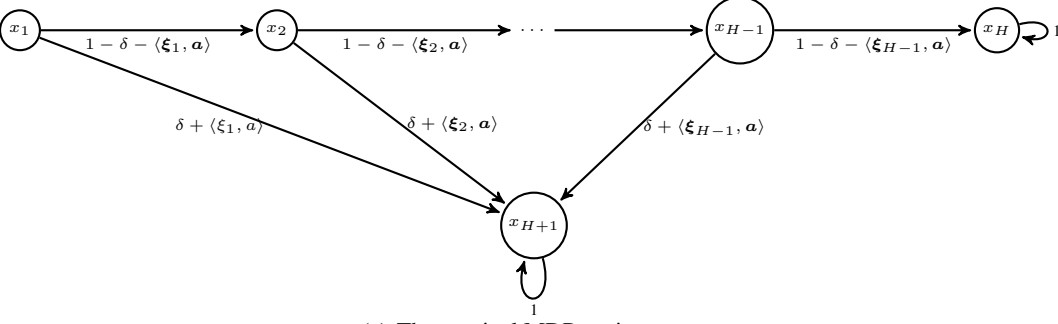

(a) The nominal MDP environment.

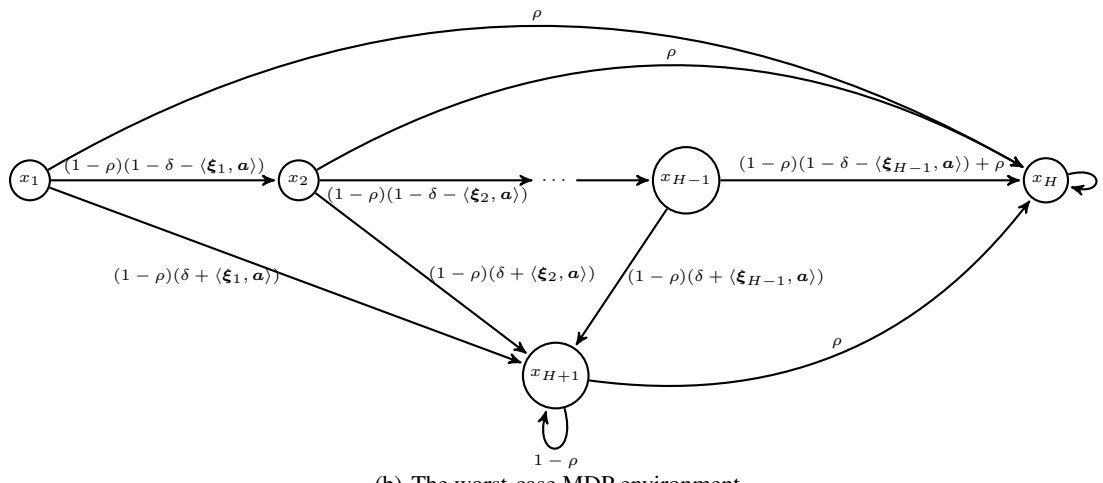

(b) The worst-case MDP environment.

Figure 4: Constructions of the nominal MDP and the worst-case MDP environments.

*Proof of Lemma G.1.* For the fixed policy $\pi$, we first get the ground truth robust value $V_1^{\pi,\rho}(x_1)$ by induction. Starting from the last step $H$, we have

$$V_H^{\pi,\rho}(x_H) = 0, \quad V_H^{\pi,\rho}(x_{H+1}) = 1.$$

For step $H - 1$, we have

$$V_{H-1}^{\pi,\rho}(x_H) = 0, \quad V_{H-1}^{\pi,\rho}(x_{H-1}) = (1-\rho)(\delta + \langle \boldsymbol{\xi}_{H-1}, \boldsymbol{a}_{H-1}^{\pi}\rangle) \cdot 1, \quad V_{H-1}^{\pi,\rho}(x_{H+1}) = 1 + (1-\rho) \cdot 1.$$

For step $H - 2$, we have $V_{H-2}^{\pi,\rho}(x_H) = 0$ and

$$
\begin{aligned}
V_{H-2}^{\pi,\rho}(x_{H+1}) &= 1 + (1-\rho) \cdot V_{H-1}(x_{H+1}) = 1 + (1-\rho) + (1-\rho)^2, \\
V_{H-2}^{\pi,\rho}(x_{H-2}) &= (1-\rho)(\delta + \langle \boldsymbol{\xi}_{H-2}, \bar{\boldsymbol{a}}_{H-2}^{\pi}\rangle) \cdot V_{H-1}^{\pi,\rho}(x_{H+1}) \\
&\quad + (1-\rho)(1 - \delta - \langle \boldsymbol{\xi}_{H-2}, \bar{\boldsymbol{a}}_{H-2}^{\pi}\rangle) \cdot V_{H-1}^{\pi,\rho}(x_{H-1}) \\
&= \left[(1-\rho) + (1-\rho)^2\right](\delta + \langle \boldsymbol{\xi}_{H-2}, \bar{\boldsymbol{a}}_{H-2}^{\pi}\rangle) \\
&\quad + (1-\rho)^2(1 - \delta - \langle \boldsymbol{\xi}_{H-2}, \bar{\boldsymbol{a}}_{H-2}^{\pi}\rangle)(\delta + \langle \boldsymbol{\xi}_{H-1}, \bar{\boldsymbol{a}}_{H-1}^{\pi}\rangle).
\end{aligned}
$$

For step $H - 3$, we have $V_{H-3}^{\pi,\rho}(x_H) = 0$ and

$$
\begin{aligned}
V_{H-3}^{\pi,\rho}(x_{H+1}) &= 1 + (1-\rho) \cdot V_{H-2}(x_{H+1}) = 1 + (1-\rho) + (1-\rho)^2 + (1-\rho)^3, \\
V_{H-3}^{\pi,\rho}(x_{H-3}) &= (1-\rho)(\delta + \langle \boldsymbol{\xi}_{H-3}, \bar{\boldsymbol{a}}_{H-3}^{\pi}\rangle) \cdot V_{H-2}^{\pi,\rho}(x_{H+1}) \\
&\quad + (1-\rho)(1 - \delta - \langle \boldsymbol{\xi}_{H-3}, \bar{\boldsymbol{a}}_{H-3}^{\pi}\rangle) \cdot V_{H-2}^{\pi,\rho}(x_{H-2}) \\
&= \left[(1-\rho) + (1-\rho)^2 + (1-\rho)^3\right](\delta + \langle \boldsymbol{\xi}_{H-3}, \bar{\boldsymbol{a}}_{H-3}^{\pi}\rangle) \\
&\quad + \left[(1-\rho)^2 + (1-\rho)^3\right](1 - \delta - \langle \boldsymbol{\xi}_{H-3}, \bar{\boldsymbol{a}}_{H-3}^{\pi}\rangle)(\delta + \langle \boldsymbol{\xi}_{H-2}, \bar{\boldsymbol{a}}_{H-2}^{\pi}\rangle)
\end{aligned}
$$

$$+ (1-\rho)^3 (1 - \delta - \langle \boldsymbol{\xi}_{H-3}, \bar{\boldsymbol{a}}_{H-3}^\pi \rangle)(1 - \delta - \langle \boldsymbol{\xi}_{H-2}, \bar{\boldsymbol{a}}_{H-2}^\pi \rangle)(\delta + \langle \boldsymbol{\xi}_{H-1}, \bar{\boldsymbol{a}}_{H-1}^\pi \rangle).$$

Keep performing the backward induction until step $h = 1$, we have

$$
\begin{aligned}
V_1^{\pi,\rho}(x_1) \\
= V_{H-(H-1)}^{\pi,\rho}(x_1) \\
= \big[(1-\rho) + \cdots + (1-\rho)^{H-1}\big](\delta + \langle \boldsymbol{\xi}_1, \bar{\boldsymbol{a}}_1^\pi \rangle) + \\
\quad \big[(1-\rho)^2 + \cdots + (1-\rho)^{H-1}\big](1 - \delta - \langle \boldsymbol{\xi}_1, \bar{\boldsymbol{a}}_1^\pi \rangle)(\delta + \langle \boldsymbol{\xi}_2, \bar{\boldsymbol{a}}_2^\pi \rangle) + \\
\quad \big[(1-\rho)^3 + \cdots + (1-\rho)^{H-1}\big](1 - \delta - \langle \boldsymbol{\xi}_1, \bar{\boldsymbol{a}}_1^\pi \rangle)(1 - \delta - \langle \boldsymbol{\xi}_2, \bar{\boldsymbol{a}}_2^\pi \rangle)(\delta + \langle \boldsymbol{\xi}_3, \bar{\boldsymbol{a}}_3^\pi \rangle) + \\
\quad + \cdots + \\
\quad (1-\rho)^{H-1}(1 - \delta - \langle \boldsymbol{\xi}_1, \bar{\boldsymbol{a}}_1^\pi \rangle)(1 - \delta - \langle \boldsymbol{\xi}_2, \bar{\boldsymbol{a}}_2^\pi \rangle) \cdots (1 - \delta - \langle \boldsymbol{\xi}_{H-2}, \bar{\boldsymbol{a}}_{H-2}^\pi \rangle)(\delta + \langle \boldsymbol{\xi}_{H-1}, \bar{\boldsymbol{a}}_{H-1}^\pi \rangle) \\
= \sum_{h=1}^{H-1} \Big( \sum_{i=h}^{H-1} (1-\rho)^i \Big)(o_h + \delta) \prod_{j=1}^{h-1} (1 - o_j - \delta),
\end{aligned}
\tag{G.2}
$$

where $o_h = \langle \boldsymbol{\xi}_h, \bar{\boldsymbol{a}}_h^\pi \rangle, \forall h \in [H]$. Recall that the optimal robust action at step $h$ is $\boldsymbol{a}_h^\star = \arg\max_{\boldsymbol{a} \in \mathcal{A}} \langle \boldsymbol{\xi}_h, \boldsymbol{a} \rangle$, and hence $\max_{\boldsymbol{a} \in \mathcal{A}} \langle \boldsymbol{\xi}_h, \boldsymbol{a} \rangle = \Delta d$. Thus, we have

$$V_1^{\star,\rho}(x_1) = \sum_{h=1}^{H-1} \Big( \sum_{i=h}^{H-1} (1-\rho)^i \Big)(d\Delta + \delta) \prod_{j=1}^{h-1} (1 - d\Delta - \delta). \tag{G.3}$$

For $k \in [H-1]$, we define

$$S_k = \sum_{h=k}^{H-1} \Big( \sum_{i=h-k+1}^{H-k} (1-\rho)^i \Big) \prod_{j=k}^{h-1} (1 - o_j - \delta)(o_h + \delta), \tag{G.4}$$

$$T_k = \sum_{h=k}^{H-1} \Big( \sum_{i=h-k+1}^{H-k} (1-\rho)^i \Big) \prod_{j=k}^{h-1} (1 - d\Delta - \delta)(d\Delta + \delta). \tag{G.5}$$

Then by (G.2), (G.3), (G.4) and (G.5), we know $V_1^{\star,\rho}(x_1) - V_1^{\pi,\rho}(x_1) = T_1 - S_1$. Next, we aim to lower bound $T_1 - S_1$. Inspired by the backward induction process, we have

$$S_k = \Big( \sum_{i=1}^{H-k} (1-\rho)^i \Big)(o_k + \delta) + S_{k+1}(1 - o_k - \delta),$$

$$T_k = \Big( \sum_{i=1}^{H-k} (1-\rho)^i \Big)(d\Delta + \delta) + T_{k+1}(1 - d\Delta - \delta).$$

Then, we have

$$
\begin{aligned}
T_k - S_k &= \Big( \sum_{i=1}^{H-k} (1-\rho)^i \Big)(d\Delta - o_k) - S_{k+1}(1 - o_k - \delta) + T_{k+1}(1 - d\Delta - \delta) \\
&= \Big( \sum_{i=1}^{H-k} (1-\rho)^i - T_{k+1} \Big)(d\Delta - o_k) + (1 - o_k - \delta)(T_{k+1} - S_{k+1}).
\end{aligned}
\tag{G.6}
$$

Define $T_H = S_H = 0$, then by the recursive formula (G.6), we have

$$T_1 - S_1 = \sum_{h=1}^{H-1} (d\Delta - o_h) \underbrace{\Big( \sum_{i=1}^{H-h} (1-\rho)^i - T_{h+1} \Big)}_{\mathrm{I}} \prod_{j=1}^{h-1} (1 - o_j - \delta). \tag{G.7}$$

To further bound (G.7), we first study the term I. Next we derive a close form expression of $T_k$. In specific, we have

$$T_k = \sum_{h=k}^{H-1} \Big( \sum_{i=h-k+1}^{H-k} (1-\rho)^i \Big) \prod_{j=k}^{h-1} (1 - d\Delta - \delta)(d\Delta + \delta)$$

$$= \Big( \sum_{i=1}^{H-k} (1-\rho)^i \Big)(d\Delta + \delta) + \Big( \sum_{i=2}^{H-k} (1-\rho)^i \Big)(1 - d\Delta - \delta)(d\Delta + \delta)$$

$$+ \Big( \sum_{i=3}^{H-k} (1 - d\Delta - \delta)^2 (d\Delta + \delta) \Big) + \cdots + (1-\rho)^{H-k}(1 - d\Delta - \delta)^{H-k-1}(d\Delta + \delta).$$

$$(G.8)$$

Multiply $T_k$ by $(1 - d\Delta - \delta)$, we have

$$(1 - d\Delta - \delta)T_k$$
$$= \Big( \sum_{i=1}^{H-k} (1-\rho)^i \Big)(d\Delta + \delta)(1 - d\Delta - \delta) + \Big( \sum_{i=2}^{H-k} (1-\rho)^i \Big)(1 - d\Delta - \delta)^2 (d\Delta + \delta)$$

$$+ \Big( \sum_{i=3}^{H-k} (1 - d\Delta - \delta)^2 (d\Delta + \delta) \Big) + \cdots + (1-\rho)^{H-k}(1 - d\Delta - \delta)^{H-k}(d\Delta + \delta). \quad (G.9)$$

Then we have

$$(G.8) - (G.9)$$
$$= (d\Delta + \delta)T_k$$
$$= \Big( \sum_{i=1}^{H-k} (1-\rho)^i \Big)(d\Delta + \delta) - (1-\rho)(1 - d\Delta - \delta)(d\Delta + \delta) - (1-\rho)^2(1 - d\Delta - \delta)^2(d\Delta + \delta)$$

$$- \cdots - (1-\rho)^{H-k}(1 - d\Delta - \delta)^{H-k}(d\Delta + \delta). \quad (G.10)$$

Divide both side of equation (G.10) by $(d\Delta+\delta)$ and then apply the formula for the sum of a geometric series, we know $T_k$ has the following closed form expression

$$T_k = \Big( \sum_{i=1}^{H-k} (1-\rho)^i \Big) - \frac{(1-\rho)(1 - d\Delta - \delta)(1 - (1-\rho)^{H-k}(1 - d\Delta - \delta)^{H-k})}{1 - (1-\rho)(1 - d\Delta - \delta)}.$$

Then, for any $h \le H/2$, we have the following bound on the term I of (G.7),

$$\sum_{i=1}^{H-h} (1-\rho)^i - T_{h+1}$$

$$= \sum_{i=1}^{H-h} (1-\rho)^i - \sum_{i=1}^{H-h-1} (1-\rho)^i + \frac{(1-\rho)(1 - d\Delta - \delta)(1 - (1-\rho)^{H-h-1}(1 - d\Delta - \delta)^{H-h-1})}{1 - (1-\rho)(1 - d\Delta - \delta)}$$

$$= (1-\rho)^{H-h} + \frac{(1-\rho)(1 - d\Delta - \delta)(1 - (1-\rho)^{H-h-1}(1 - d\Delta - \delta)^{H-h-1})}{1 - (1-\rho)(1 - d\Delta - \delta)}$$

$$= (1-\rho)^{H-h} + (1-\rho)(1 - d\Delta - \delta) + \cdots + (1-\rho)^{H-h-1}(1 - d\Delta - \delta)^{H-h-1}$$

$$\ge (1 - d\Delta - \delta)^H \big( (1-\rho) + \cdots + (1-\rho)^{H-h-1} + (1-\rho)^{H-h} \big) \quad (G.11)$$

$$\ge \Big( 1 - \frac{2}{H} \Big)^H \big( (1-\rho) + \cdots + (1-\rho)^{H-h-1} + (1-\rho)^{H-h} \big)$$

$$\ge \frac{1}{12} \sum_{i=1}^{H-h} (1-\rho)^i, \quad (G.12)$$

where (G.11) holds due to $3d\Delta \le \delta = 1/H$ and (G.12) holds due to $H \ge 6$. Next, we carefully bound the LHS of (G.12) with respect to $\rho$. For any $h \le H/2$ and $\rho \in (0, 3/4]$, we have

$$\frac{1}{12} \sum_{i=1}^{H-h} (1-\rho)^i \ge \frac{1}{12} \frac{(1-\rho)(1 - (1-\rho)^{H/2})}{\rho} \ge \frac{1}{50} \frac{1 - (1-\rho)^{H/2}}{\rho}.$$

Given the fact that

$$\frac{1 - (1-\rho)^{H/2}}{\rho} = \Theta\Big( \min \Big(H, \frac{1}{\rho}\Big)\Big),$$

there exist a constant $c > 0$, such that

$$\frac{1 - (1-\rho)^{H/2}}{\rho} \geq c \cdot \min\left(H, \frac{1}{\rho}\right).$$

Then we have

$$\sum_{i=1}^{H-h} (1-\rho)^i - T_{h+1} \geq c' \cdot \min\left(H, \frac{1}{\rho}\right), \tag{G.13}$$

where $c' = c/50$. Moreover, with the choice of parameter $3d\Delta \leq \delta, \delta = 1/H$, and $H \geq 6$, we have

$$\prod_{j=1}^{h-1} (1 - o_j - \delta) \geq (1 - 4\delta/3)^H \geq 1/3. \tag{G.14}$$

Therefore, by (G.7), (G.13) and (G.14), we have

$$V_1^{\star,\rho}(x_1) - V_1^{\pi,\rho}(x_1) = T_1 - S_1$$

$$\geq c'' \cdot \min\{H, 1/\rho\} \cdot \sum_{h=1}^{H/2} (d\Delta - o_h)$$

$$= c'' \cdot \min\{H, 1/\rho\} \cdot \sum_{h=1}^{H/2} \left( \max_{a \in \mathcal{A}} \langle \boldsymbol{\mu}_h, \boldsymbol{a} \rangle - \langle \boldsymbol{\mu}_h, \bar{\boldsymbol{a}}_h^\pi \rangle \right),$$

where $c'' = c'/3$. This completes the proof. $\qquad\square$

## G.3   Proof of Theorem 5.1

Next, we present an existing result on lower bounding the regret of linear bandits induced by Lemma G.1. This result is useful in deriving the lower bound in Theorem 5.1.

**Lemma G.2.** [62, Lemma 25] Fix a positive real $0 \leq \delta \leq 1/3$, and positive integers $K, d$ and assume that $K \geq d^2/(2\delta)$. Let $\Delta = \sqrt{\delta/K}/(4\sqrt{2})$ and consider the linear bandit problems $\mathcal{L}_{\boldsymbol{\mu}}$ parameterized with a parameter vector $\boldsymbol{\mu} \in \{-\Delta, \Delta\}^d$ and action set $\mathcal{A} = \{-1, 1\}^d$ so that the reward distribution for taking action $\boldsymbol{a} \in \mathcal{A}$ is a Bernoulli distribution $\text{Bernoulli}(\delta + \langle \boldsymbol{\mu}, \boldsymbol{a} \rangle)$. Then for any bandit algorithm $\mathcal{B}$, there exists a $\boldsymbol{\mu}^\star \in \{-\Delta, \Delta\}^d$ such that the expected pseudo-regret of $\mathcal{B}$ over first $K$ steps on bandit $\mathcal{L}_{\boldsymbol{\mu}^\star}$ is lower bounded as follows:

$$\mathbb{E}_{\boldsymbol{\mu}^\star} Regret(K) \geq \frac{d\sqrt{K\delta}}{8\sqrt{2}}.$$

Note that the expectation is with respect to a distribution that depends both on $\mathcal{B}$ and $\mu^\star$, but since $\mathcal{B}$ is fixed, this dependence is hidden.

Now we are ready to prove the lower bound in Theorem 5.1.

*Proof of Theorem 5.1.* By Lemma G.1, we have

$$\mathbb{E}_{\boldsymbol{\xi}} \text{AveSubopt}(M_{\boldsymbol{\xi}}, K) = \frac{1}{K} \mathbb{E}_{\boldsymbol{\xi}} \Big[ \sum_{k=1}^{K} [V_1^{\star,\rho}(x_1) - V_1^{\pi,\rho}(x_1)] \Big]$$

$$\geq c \cdot \frac{\min\{H, 1/\rho\}}{K} \sum_{h=1}^{H/2} \mathbb{E}_{\boldsymbol{\xi}} \Big[ \sum_{k=1}^{K} \left( \max_{a \in \mathcal{A}} \langle \boldsymbol{\xi}_h, \boldsymbol{a} \rangle - \langle \boldsymbol{\xi}_h, \bar{\boldsymbol{a}}_h^{\pi_k} \rangle \right) \Big].$$

Note that the learning process is conducted on the nominal environment, which is exactly the MDP in [62], thus the rest proof of Theorem 4.2 follows the argument in the proof of Theorem 8 in [62]. In particular, define $\boldsymbol{\xi}^{-h} = (\boldsymbol{\xi}_1, \cdots, \boldsymbol{\xi}_{h-1}, \boldsymbol{\xi}_{h+1}, \cdots, \boldsymbol{\xi}_H)$, then every MDP policy $\pi$ induces a bandit algorithm $\mathcal{B}_{\pi, h, \boldsymbol{\xi}^{-h}}$ for the linear bandit of Lemma G.2. Moreover, our choice of parameters in

926 (G.1) satisfy the requirement of Lemma G.2. Denote the regret of this bandit problem on $\mathcal{L}_{\boldsymbol{\xi}}$ as
927 BanditRegret$(\mathcal{B}_{\pi,h,\boldsymbol{\xi}^{-h}}, \boldsymbol{\xi}_h)$, then we have

$$
\begin{aligned}
\sup_{\boldsymbol{\xi}} \mathbb{E}_{\boldsymbol{\xi}} \text{AveSubopt}(M_{\boldsymbol{\xi}}, K) &\geq \sup_{\boldsymbol{\xi}} c \cdot \frac{\min\{H, 1/\rho\}}{K} \sum_{h=1}^{H/2} \text{BanditRegret}(\mathcal{B}_{\pi,h,\boldsymbol{\xi}^{-h}}, \boldsymbol{\xi}_h) \\
&\geq \sup_{\boldsymbol{\xi}} c \cdot \frac{\min\{H, 1/\rho\}}{K} \sum_{h=1}^{H/2} \inf_{\tilde{\boldsymbol{\xi}}^{-h}} \text{BanditRegret}(\mathcal{B}_{\pi,h,\tilde{\boldsymbol{\xi}}^{-h}}, \boldsymbol{\xi}_h) \\
&= c \cdot \frac{\min\{H, 1/\rho\}}{K} \sum_{h=1}^{H/2} \sup_{\boldsymbol{\xi}} \inf_{\tilde{\boldsymbol{\xi}}^{-h}} \text{BanditRegret}(\mathcal{B}_{\pi,h,\tilde{\boldsymbol{\xi}}^{-h}}, \boldsymbol{\xi}_h) \\
&\geq c \cdot \frac{\min\{H, 1/\rho\} d H \sqrt{K\delta}}{16\sqrt{2} \cdot K} \\
&= \frac{c}{16\sqrt{2}} \cdot \frac{d\sqrt{H} \cdot \min\{H, 1/\rho\}}{\sqrt{K}}.
\end{aligned}
$$

928 This completes the proof. $\qquad\square$

# H Auxiliary Lemmas

930 In this section, we present some standard technical results in the literature that our proofs are built on.

931 **Proposition H.1.** (Strong duality for TV [36, Lemma 4]). Given any probability measure $\mu^0$ over
932 $\mathcal{S}$, a fixed uncertainty level $\rho$, the uncertainty set $\mathcal{U}^\rho(\mu^0) = \{\mu : \mu \in \Delta(\mathcal{S}), D_{TV}(\mu||\mu^0) \leq \rho\}$, and
933 any function $V : \mathcal{S} \to [0, H]$, we obtain

$$
\inf_{\mu \in \mathcal{U}^\rho(\mu^0)} \mathbb{E}_{s \sim \mu} V(s) = \max_{\alpha \in [V_{\min}, V_{\max}]} \left\{ \mathbb{E}_{s \sim \mu^0}[V(s)]_\alpha - \rho\big(\alpha - \min_{s'}[V(s')]_\alpha\big) \right\}, \quad \text{(H.1)}
$$

934 where $[V(s)]_\alpha = \min\{V(s), \alpha\}$, $V_{\min} = \min_s V(s)$ and $V_{\max} = \max_s V(s)$. Notably, the range of
935 $\alpha$ can be relaxed to $[0, H]$ without impacting the optimization.

936 **Lemma H.2.** [1, Lemma 12] Let $\mathbf{A}$, $\mathbf{B}$ and $\mathbf{C}$ be positive semi-definite matrices such that $\mathbf{A} =$
937 $\mathbf{B} + \mathbf{C}$. Then we have that

$$
\sup_{\mathbf{x} \neq 0} \frac{\mathbf{x}^\top \mathbf{A} \mathbf{x}}{\mathbf{x}^\top \mathbf{B} \mathbf{x}} \leq \frac{\det(\mathbf{A})}{\det(\mathbf{B})}.
$$

938 **Lemma H.3.** [1, Confidence Ellipsoid, Theorem 2] Let $\{\mathcal{G}_k\}_{k=1}^\infty$ be a filtration, and $\{\mathbf{x}_k, \eta_k\}_{k \geq 1}$
939 be a stochastic process such that $\mathbf{x}_k \in \mathbb{R}^d$ is $\mathcal{G}_k$-measurable and $\eta_k \in \mathbb{R}$ is $\mathcal{G}_{k+1}$-measurable. Let $L$,
940 $\sigma, \boldsymbol{\Sigma}, \epsilon > 0, \boldsymbol{\mu}^* \in \mathbb{R}^d$. For $k \geq 1$, let $y_k = \langle \boldsymbol{\mu}^*, \mathbf{x}_k \rangle + \eta_k$ and suppose that $\eta_k, \mathbf{x}_k$ also satisfy

$$
\mathbb{E}[\eta_k \mid \mathcal{G}_k] = 0, |\eta_k| \leq R, \|\mathbf{x}_k\|_2 \leq L.
$$

941 For $k \geq 1$, let $\mathbf{Z}_k = \lambda \mathbf{I} + \sum_{i=1}^k \mathbf{x}_i \mathbf{x}_i^\top$, $\boldsymbol{b}_k = \sum_{i=1}^k y_i \mathbf{x}_i$, $\boldsymbol{\mu}_k = \mathbf{Z}_k^{-1} \boldsymbol{b}_k$, and

$$
\beta_k = R \sqrt{d \log\left(1 + \frac{kL^2}{d\lambda}\right) + 2\log\frac{1}{\delta}}.
$$

942 Then, for any $0 < \delta < 1$, we have with probability at least $1 - \delta$ that,

$$
\forall k \geq 1, \left\| \sum_{i=1}^k \mathbf{x}_i \eta_i \right\|_{\mathbf{Z}_k^{-1}} \leq \beta_k, \|\boldsymbol{\mu}_k - \boldsymbol{\mu}^*\|_{\mathbf{Z}_k} \leq \beta_k + \sqrt{\lambda} \|\boldsymbol{\mu}^*\|_2.
$$

943 **Lemma H.4.** [19, Lemma D.1] Let $\boldsymbol{\Lambda}_t = \lambda \mathbf{I} + \sum_{i=1}^t \boldsymbol{\phi}_i \boldsymbol{\phi}_i^\top$, where $\boldsymbol{\phi}_i \in \mathbb{R}^d$ and $\lambda > 0$. Then we
944 have

$$
\sum_{i=1}^t \boldsymbol{\phi}_i^\top (\boldsymbol{\Lambda}_t)^{-1} \boldsymbol{\phi}_i \leq d.
$$

**Lemma H.5.** [14, Lemma D.5] Let $\mathbf{A} \in \mathbb{R}^{d \times d}$ be a positive definite matrix where its largest eigenvalue $\lambda_{\max}(\mathbf{A}) \leq \lambda$. Let $\mathbf{x}_1, ..., \mathbf{x}_k$ be $k$ vectors in $\mathbb{R}^d$. Then it holds that

$$\left\| \mathbf{A} \sum_{i=1}^{k} \mathbf{x}_i \right\| \leq \sqrt{\lambda k} \left( \sum_{i=1}^{k} \|\mathbf{x}_i\|_{\mathbf{A}}^2 \right)^{1/2}.$$

**Lemma H.6.** [39, Covering number of Euclidean ball] For any $\varepsilon > 0$, $\mathcal{N}_\varepsilon$, the $\varepsilon$-covering number of the Euclidean ball of radius $B > 0$ in $\mathbb{R}^d$ satisfies

$$\mathcal{N}_\varepsilon \leq \left( 1 + \frac{2B}{\varepsilon} \right)^d \leq \left( \frac{3B}{\varepsilon} \right)^d.$$

**Lemma H.7.** [39, Covering number of an interval] Denote the $\epsilon$-covering number of the closed interval $[a, b]$ for some real number $b > a$ with respect to the distance metric $d(\alpha_1, \alpha_2) = |\alpha_1 - \alpha_2|$ as $\mathcal{N}_\epsilon([a, b])$. Then we have $\mathcal{N}_\epsilon([a, b]) \leq 3(b - a)/\epsilon$.

**Lemma H.8.** [61, Theorem 4.3] Let $\{\mathcal{G}_k\}_{k=1}^{\infty}$ be a filtration, and $\{\mathbf{x}_k, \eta_k\}_{k \geq 1}$ be a stochastic process such that $\mathbf{x}_k \in \mathbb{R}^d$ is $\mathcal{G}_k$-measurable and $\eta_k \in \mathbb{R}$ is $\mathcal{G}_{k+1}$-measurable. Let $L, \sigma > 0, \mu^* \in \mathbb{R}^d$. For $k \geq 1$, let $y_k = \langle \mu^*, \mathbf{x}_k \rangle + \eta_k$ and suppose that $\eta_k, \mathbf{x}_k$ also satisfy

$$\mathbb{E}[\eta_k \mid \mathcal{G}_k] = 0, \mathbb{E}[\eta_k^2 \mid \mathcal{G}_k] \leq \sigma^2, |\eta_k| \leq R, \|\mathbf{x}_k\|_2 \leq L.$$

For $k \geq 1$, let $\beta_k = \widetilde{O}\big(\sigma\sqrt{d} + \max_{1 \leq i \leq k} |\eta_i| \min\{1, \|\mathbf{x}_i\|_{\mathbf{Z}_{i-1}^{-1}}\}\big)$ and $\mathbf{Z}_k = \lambda \mathbf{I} + \sum_{i=1}^{k} \mathbf{x}_i \mathbf{x}_i^\top$, $\mathbf{b}_k = \sum_{i=1}^{k} y_i \mathbf{x}_i$, $\mu_k = \mathbf{Z}_k^{-1} \mathbf{b}_k$. Then, for any $0 < \delta < 1$, with probability at least $1 - \delta$, for all $k \in [K]$, we have

$$\left\| \sum_{i=1}^{k} \mathbf{x}_i \eta_i \right\|_{\mathbf{Z}_k^{-1}} \leq \beta_k, \|\mu_k - \mu^*\|_{\mathbf{Z}_k} \leq \beta_k + \sqrt{\lambda}\|\mu^*\|_2.$$

**Lemma H.9.** [19, Lemma D.4] Let $\{s_i\}_{i=1}^{\infty}$ be a stochastic process on state space $\mathcal{S}$ with corresponding filtration $\{\mathcal{F}_i\}_{i=1}^{\infty}$. Let $\{\phi_i\}_{i=1}^{\infty}$ be an $\mathbb{R}^d$-valued stochastic process where $\phi_i \in \mathcal{F}_{i-1}$, and $\|\phi_i\| \leq 1$. Let $\Lambda_k = \lambda \mathbf{I} + \sum_{i=1}^{k} \phi_i \phi_i^\top$. Then for any $\delta > 0$, with probability at least $1 - \delta$, for all $k \geq 0$, and any $V \in \mathcal{V}$ with $\sup_{s \in \mathcal{S}} |V(s)| \leq H$, we have

$$\left\| \sum_{i=1}^{k} \phi_i \{V(s_i) - \mathbb{E}[V(s_i) \mid \mathcal{F}_{i-1}]\} \right\|_{\Lambda_k^{-1}}^2 \leq 4H^2 \left[ \frac{d}{2} \log\left( \frac{k + \lambda}{\lambda} \right) + \log \frac{\mathcal{N}_\varepsilon}{\delta} \right] + \frac{8k^2 \varepsilon^2}{\lambda},$$

where $\mathcal{N}_\varepsilon$ is the $\varepsilon$-covering number of $\mathcal{V}$ with respect to the distance $\text{dist}(V, V') = \sup_{s \in \mathcal{S}} |V(s) - V'(s)|$.

**Lemma H.10.** [24, Lemma 5.1 (Range Shrinkage)] For any $(\rho, \pi, h) \in (0, 1] \times \Pi \times [H]$, we have $\max_{s \in \mathcal{S}} V_h^{\pi, \rho}(s) - \min_{s \in \mathcal{S}} V_h^{\pi, \rho}(s) \leq (1 - (1 - \rho)^{H-h+1})/\rho$.

