# OpenReview forum: "Near-Optimal Reinforcement Learning for Linear Distributionally Robust Markov Decision Processes"
_NeurIPS.cc/2025/Workshop/Reliable_ML — NeurIPS 2025 - Reliable ML Workshop_

### Official Review · Reviewer_dUGA · 2025-09-20
**This is an interesting paper that fits well this workshop**

**Rating:** 7
**Confidence:** 2

**Review:**

Summary:
This paper studies online d-rectangular linear DRMDPs and proposes We-DRIVE-U, a novel variance-aware algorithm with rare-switching updates. The main result of this paper is a new algorithm that achieves a better average suboptimality with an information theoretical lower bound close to the obtained upper bound. This paper establishes a solid theory for DRMDP with a complete list of proofs. Furthermore, this paper is well written. I think this paper is a good fit for this workshop.

---

### Official Review · Reviewer_QjLn · 2025-09-20
**Review of "Near-Optimal Reinforcement Learning for Linear Distributionally Robust Markov Decision Processes"**

**Rating:** 8
**Confidence:** 3

**Review:**

Summary\
This paper is purely theoretical. The authors introduce We-DRIVE-U, which is an algorithm for online robust RL in linear DRMDPs with TV uncertainty. They prove a near-optimal suboptimality bound that improved from the previous best rate of about d²H²/√K down to dH/√K. Also, they give the first matching lower bound showing guarantees on policy switch and oracle-call complexity. Experiments exist only in the appendix and are very limited.

Pros
* Strong theoretical contribution by proving a tighter upper bound + first lower bound for online robust RL with function approximation.
* Rare-switching update rule, which reduces the policy switches and oracle calls.
* Clear and better positioning against prior SOTA works (i.e., DR-LSVI-UCB and offline robust RLs).

Cons
* Assumes the data covers all the important situations. But in many real tasks or scenarios, that’s not true.
* No experiments (only tiny toy tests in appendix section). So, it is difficult to evaluate how it works in practice.
* It handles only one specific kind of robustness (total variation). It is unclear if the ideas would still work for other common ones, like KL Divergence or Wasserstein.
* The math hides some very big numbers (like d^15 or H^13). So, the gains might only show up when there is a lot of data, even if small toy tests look good based on the result.